# `IAPO`: Information-Aware Policy Optimization for Token-Efficient Reasoning

Yinhan He [* 1]   Yaochen Zhu [* 1]   Mingjia Shi [1]   Wendy Zheng [1]   Lin Su [2]   Xiaoqing Wang [2]   Qi Guo [2]   Jundong Li [1]

## Abstract

Large language models increasingly rely on long chains of thought to improve accuracy, yet such gains come with substantial inference-time costs. We revisit token-efficient post-training and argue that existing sequence-level reward-shaping methods offer limited control over how reasoning effort is allocated across tokens. To bridge the gap, we propose `IAPO`, an information-theoretic post-training framework that assigns token-wise advantages based on each token's conditional mutual information (MI) with the final answer. This yields an explicit, principled mechanism for identifying informative reasoning steps and suppressing low-utility exploration. We provide a theoretical analysis showing that our `IAPO` can induce monotonic reductions in reasoning verbosity without harming correctness. Empirically, `IAPO` consistently improves reasoning accuracy while reducing reasoning length by up to 47%, outperforming existing token-efficient RL methods across various reasoning datasets. Our results demonstrate that information-aware advantage shaping is a powerful and general direction for token-efficient post-training. The code is available at https://github.com/YinhanHe123/IAPO.

## 1. Introduction

Reinforcement learning (RL)-based post-training has become one of the most effective tools for strengthening the reasoning capabilities of large language models (LLMs) (Kumar et al., 2025). Recently, group relative policy optimization (GRPO) (Shao et al., 2024) and its variants have proven especially powerful: by contrasting multiple sampled solutions and explicitly rewarding reasoning chains that lead to better answers, these methods incentivize deeper,

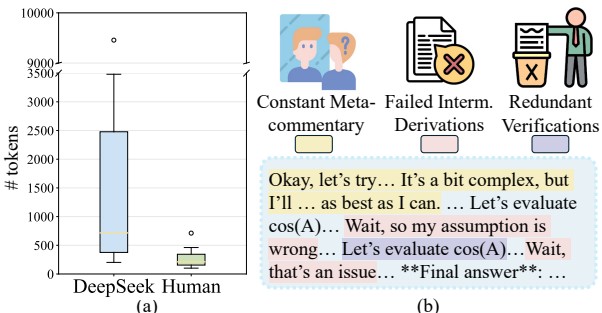

*Figure 1.* Reasoning verbosity of RL post-trained LLMs. (a) Comparison of reasoning length between LLM (DeepSeekR1-Distilled-Qwen-1.5B (DeepSeek-AI, 2025)) and a human volunteer on math problems (Lightman et al., 2024). (b) Illustration of why the reasoning generated by the LLM are unnecessarily verbose.

more structured multi-step reasoning, thus delivering substantial gains across diverse reasoning benchmarks.

However, GRPO also comes with an often-overlooked cost. Post-trained LLMs tend to produce *more* reasoning than necessary. While additional reasoning steps enable more computational power of the base LLM, the generated content could be redundant, circular, or uninformative. As shown in Fig. 1, DeepSeekR1-Distilled-Qwen-1.5B (DeepSeek-AI, 2025) and a Ph.D.-level volunteer both achieve perfect accuracy on 20 MATH-500 problems (Lightman et al., 2024), yet the model generates substantially longer reasoning (1,658 vs. 264 tokens on average). Such verbosity inflates inference latency and computational cost, which scale quadratically with sequence length, while offering diminishing returns in correctness. This mismatch motivates the central question: *Can we retain the reasoning strength achieved by RL post-training while reducing unnecessary reasoning tokens?*

Several recent approaches have attempted to answer this question by shaping advantages in GRPO (Shao et al., 2024) to favor brevity (Tan et al., 2025; Shrivastava et al., 2025; Lee & Tong, 2025; Yu et al., 2025). However, the semantics of the generated reasoning are overall overlooked. Generally, existing methods fall into two broad categories. Length-based methods (Shrivastava et al., 2025; Yu et al., 2025) favor shorter outputs by assigning uniformly higher advantages to all tokens in concise outputs, regardless of whether they are actually informative. Position-based methods (Dai et al., 2025; Lee & Tong, 2025; Yu et al., 2025) penalize

---

[*]Equal contribution   [1]University of Virginia, Charlottesville, VA, USA  [2]LinkedIn Inc., Sunnyvale, CA, USA. Correspondence to: Jundong Li <jl6qk@virginia.edu>.

*Proceedings of the 43rd International Conference on Machine Learning*, Seoul, South Korea. PMLR 306, 2026. Copyright 2026 by the author(s).

later tokens via position-dependent advantage decay, even if some later tokens can be crucial for producing correct answers. Both paradigms share a fundamental limitation: *they are content-agnostic*, i.e., they determine a token's advantage by completion length or token position, instead of the information it contributes to the correctness of the final answer. Consequently, current approaches struggle to distinguish essential reasoning from redundant tokens.

To overcome this limitation, we propose **I****nformation-** **A****ware** **P****olicy** **O****ptimization** (`IAPO`), a novel post-training framework that brings information-theoretic awareness into intra-completion token-level advantages for policy optimization. Specifically, `IAPO` consists of two modules. The **information-aware advantage shaping module** assigns token-level advantages by quantifying each token's contribution to answer correctness through its conditional mutual information (MI) with the final answer, conditioned on the preceding partial completion. This module provides a principled mechanism for identifying informative reasoning steps and reducing low-utility generation. The **efficient conditional MI estimation module** efficiently estimates token-level conditional MI values utilizing an early-exit–based conditional MI estimator, together with KV-cache preloading and chunk-wise forwarding techniques, to significantly reduce the computational overhead of MI estimation, making our proposed `IAPO` computationally tractable at scale.

Empirically, `IAPO` delivers substantial gains in token efficiency while preserving strong reasoning performance. Applied to Qwen2.5-7B-Instruct (Yang et al., 2024; Team, 2024), `IAPO` achieves a **47%** reduction in reasoning length relative to the base model and a 43% improvement over the best existing token-efficient RL baselines on GSM8K (Cobbe et al., 2021), without sacrificing accuracy.

Our contributions are as follows. (1) **Problem Formulation.** We formalize token-efficient post-training as maximizing the ratio of expected task accuracy to completion length, capturing the objective of achieving high reasoning performance with minimal token consumption. (2) **Theoretical Framework.** We propose `IAPO`, a token-level advantage-shaping framework that assigns advantages based on conditional MI between tokens and final answers. We provide a theoretical analysis demonstrating how `IAPO` reduces the expected completion length while maintaining model accuracy. (3) **Efficient Module Design.** We introduce an early-exit conditional MI estimator along with KV-cache preloading and chunk-wise forwarding techniques, making token-level MI computation tractable for modern LLMs at scale. (4) **Extensive Empirical Validation.** Across multiple reasoning datasets and model scales, `IAPO` consistently achieves state-of-the-art token efficiency compared with existing token-efficient RL post-training baselines for LLMs.

**Conflict of Interest Disclosure.** Lin Su, Xiaoqing Wang, and Qi Guo are employed by LinkedIn Inc.; the remaining authors are affiliated with the University of Virginia. This work does not evaluate or rely on any product, model, or service developed by LinkedIn, and the authors declare no financial conflicts of interest.

## 2. Preliminaries and Problem Definition

**Preliminaries.** Let $q \sim Q$ denote an input query (e.g., a math problem), and let $o$ be a completion sampled from an LLM policy $\pi_\theta$. Post-training aims to optimize a $\pi_\theta$ which maximize the expected rewards $r$ (a value quantifying the quality of $o$), i.e., $\max_\theta \mathbb{E}_{q \sim Q, o \sim \pi_\theta(o|q)}[r(o)]$.

For each query $q$, GRPO (Shao et al., 2024) samples a group of $G$ completions $\{o_i\}_{i=1}^G$ from a frozen policy $\pi_{\theta_{\text{old}}}$. Each completion $o_i$ receives a reward $r_i \in \mathbb{R}$ from a reward model. GRPO computes advantages as group-wise normalized rewards $\tilde{A}_i = \frac{r_i - \text{mean}(\mathbf{r})}{\text{std}(\mathbf{r})}$, where $\mathbf{r} = \{r_i\}_{i=1}^G$. The normalized advantage is then uniformly assigned to all token positions $t$s in $o_i$ as token-wise advantages $\tilde{A}_{i,t} = \tilde{A}_i, \forall t$. The GRPO objective maximizes a clipped policy-gradient surrogate (Grondman et al., 2012) with KL regularization:

$$J_{\text{GRPO}}(\theta) = \mathbb{E}_{q,\{o_i\}} \left[ \frac{1}{G} \sum_{i=1}^G \frac{1}{|o_i|} \sum_{t=1}^{|o_i|} \min \left( \rho_{i,t}(\theta) \, \tilde{A}_{i,t}, \right. \right.$$

$$\left. \left. \text{clip}(\rho_{i,t}(\theta), 1 - \varepsilon, 1 + \varepsilon) \, \tilde{A}_{i,t} \right) - \beta \, D_{\text{KL}}(\pi_\theta \, \| \, \pi_{\text{ref}}) \right],$$

where $\rho_{i,t}(\theta) = \frac{\pi_\theta(o_{i,t}|q,o_{i,<t})}{\pi_{\theta_{\text{old}}}(o_{i,t}|q,o_{i,<t})}$. While GRPO (Shao et al., 2024) effectively improves reasoning accuracy, its uniform token-level advantage fails to distinguish informative tokens from redundant ones, often leading to verbose completions.

**Problem 1** (Token-Efficient Post-Training). *Given a query–answer distribution $(Q, Y)$, we aim to learn a policy $\pi_\theta$ that maximizes reasoning accuracy per generated token for samples $(q, y)$ drawn from $(Q, Y)$:*

$$\max_\theta \quad \frac{\mathbb{E}_{q \sim Q, \, o \sim \pi_\theta} \left[ \mathbb{I}\{o \text{ is correct}\} \right]}{\mathbb{E}_{q \sim Q, \, o \sim \pi_\theta} \left[ |o| \right]} \tag{1}$$

$$s.t. \quad \mathbb{E}_{q \sim Q, \, o \sim \pi_\theta} \left[ \mathbb{I}\{o \text{ is correct}\} \right] \geq \tau,$$

Here, $\tau$ denotes a minimum effectiveness threshold that prevents degenerated policies, e.g., trivially short but incorrect completions, from being mistakenly regarded as token-efficient. This formulation captures the central objective of token-efficient reasoning: preserving task-relevant information while eliminating redundant token generation.

## 3. Proposed Methodology

We first present an overview of the `IAPO` framework, followed by an illustration of its two critical modules: the

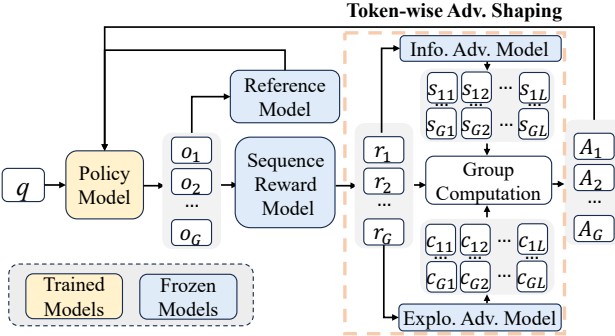

*Figure 2.* Illustration of the information-aware advantage shaping module, where $s_{i,j}$ and $c_{i,j}$ are token-wise advantages of the informativeness level and exploration adjustment of the $j$th token in the $i$th completion $o_i$ of the completions group $\{o_i\}_{i=1}^{G}$.

information-aware advantage shaping module and the efficient conditional MI estimation module.

### 3.1. Overview

We address the token-efficient post-training problem defined in Problem 1 by explicitly encouraging models to generate *informative* tokens while suppressing redundant ones. We propose IAPO framework (Fig. 2), composed of two modules: The **information-aware advantage shaping module** assigns token-level advantages based on each token's informativeness (measured via conditional MI w.r.t. the final answer). To prevent training collapse and encourage beneficial exploration, this module incorporates an exploration adjustment metric. However, computing token informativeness faces two fundamental challenges: (1) *inaccessible conditional MI* and (2) *significant computational overhead* for MI estimation. We address these challenges with the proposed **efficient conditional MI estimation module**, which tackles the first challenge via an *early-exit–based conditional MI estimator* and mitigates the second through our *KV-cache preloading* and *chunk-wise forwarding* techniques.

### 3.2. Information-Aware Advantage Shaping Module

#### 3.2.1. INFORMATIVENESS LEVEL OF A TOKEN

To quantify how much information a token $o_{i,t}$ from a completion $o_i$ contributes toward producing the final answer $y_i$, we measure its informativeness using the conditional mutual information (MI) $I(y_i; o_{i,t} \mid q, o_{i,<t})$. We adopt this metric for two reasons: (1) **Semantic alignment:** From an information-theoretic perspective (Ash, 2012; Brillouin, 2013), conditional MI measures the reduction in uncertainty of the final answer after observing $o_{i,t}$, conditioned on the preceding context (i.e., $o_{i,<t}$). This directly reflects how semantically informative $o_{i,t}$ is for determining $y_i$. (2) **Selective decomposition:** If an entire completion $o_i$ has MI $I(y_i; o_i \mid q)$ with the answer, the chain rule of MI gives $I(y_i; o_i \mid q) = \sum_t I(y; o_{i,t} \mid q, o_{i,<t})$. Thus, token-level

conditional MI provides a principled decomposition of the total information contribution across the sequence. For any token budget $k$, selecting the $k$ tokens with the highest conditional MI yields the subsequence that best preserves the mutual-information signal of the original completion. Formally, $\hat{o}_i^k = \arg\max_{\tilde{o}_i^k \subset o_i, |\tilde{o}_i^k| = k} I(y_i; \tilde{o}_i^k \mid q)$. We therefore incorporate $s_{i,t}$ as the token-level informativeness score in the advantage shaping of IAPO, where

$$s_{i,t} = I(y_i; o_t \mid q_i, o_{<t}). \tag{2}$$

#### 3.2.2. TOKEN EXPLORATION ADJUSTMENTS

While a token's informativeness level quantifies how much it contributes to producing the correct answer, relying solely on this metric for RL-based LLM post-training raises two potential concerns. First, aggressively optimizing for informativeness alone may lead to premature trajectory collapse, in which the model converges to overly concise reasoning patterns that sacrifice its reasoning accuracy. Second, the token-wise informativeness metric solely evaluates tokens' contributions within the current reasoning trajectory, neglecting that the model also needs to explore alternative, potentially more effective reasoning paths.

To address the challenges, we introduce an exploration adjustment term that introduce token-level advantages to explore other potentially effective reasoning paths, thus further maintaining the reasoning accuracy. Specifically, we define the token-level exploration advantage as:

$$c_{i,t} = \begin{cases} \pi_{\theta_{\text{old}}}(o_{i,t}|q, o_{i,<t}), & \text{if } o_i \text{ is correct,} \\ -\pi_{\theta_{\text{old}}}(o_{i,t}|q, o_{i,<t}), & \text{if } o_i \text{ is incorrect,} \end{cases} \tag{3}$$

where $\pi_{\theta_{\text{old}}}(o_{i,t}|q, o_{i,<t})$ denotes the probability assigned by the frozen policy $\pi_{\theta_{\text{old}}}$ of the LLM to token $o_{i,t}$ before the current policy update step. Intuitively, $\pi_{\theta_{\text{old}}}(o_{i,t} \mid q, o_{i,<t})$ reflects how expected or uncertain the model is about generating token $o_t$. Assigning a positive advantage of $\pi_{\theta_{\text{old}}}(o_{i,t} \mid q, o_{i,<t})$ for correct completions amplifies tokens that the model is already confident about, which *reduces the entropy* of the policy around high-confidence states and therefore *supresses exploration*. Conversely, assigning a negative advantage of $-\pi_{\theta_{\text{old}}}(o_{i,t} \mid q, o_{i,<t})$ for incorrect completions inverts this signal: tokens the model is confident about now reduce the advantage, pushing the model toward less probable alternatives. This *increases the entropy* of the policy and therefore *encourages exploration*. Please see the formal theoretical analysis in Section 4.2.

#### 3.2.3. TOKEN-WISE ADVANTAGE ASSIGNMENT

In contrast to the vanilla GRPO (Shao et al., 2024) that assigns a uniform advantage to all tokens within a completion, IAPO assigns the advantage of a token using three components: *sequence-level reward*, *token-level informativeness*

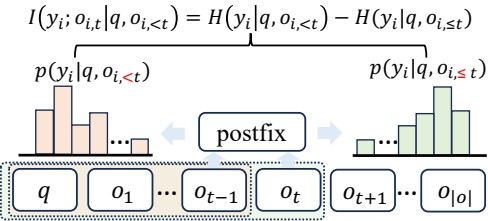

*Figure 3.* Illustration of early-exit-based conditional MI estimator. We highlight "$o_{<t}$" and "$o_{\leq t}$" to emphasis the inclusion and exclusion of the current examined token $o_t$ in the partial completions.

*level*, and *token-level exploration adjustment*. Specifically, for completion $o_i$ with reward $r_i$, IAPO assigns advantage

$$\tilde{A}_{i,t} = \underbrace{\text{norm}(r_i, \mathbf{r})}_{\text{seq-level reward}} + \alpha \underbrace{\text{norm}(s_{i,t}, \mathbf{s}_i)}_{\text{token-level info.}} + \beta \underbrace{\text{norm}(c_{i,t}, \mathbf{c}_i)}_{\text{token-level explo.}}$$
(4)

to token $o_{i,t}$, where $\text{norm}(x, \mathbf{v}) = [x - \text{mean}(\mathbf{v})]/\text{std}(\mathbf{v})$, $\mathbf{r} = \{r_i\}_{i=1}^G$, $\mathbf{s}_i = \{s_{i,t}\}_{t=1}^{|o_i|}$, and $\mathbf{c}_i = \{c_{i,t}\}_{t=1}^{|o_i|}$. Substituting $\tilde{A}_{i,t}$ into the GRPO objective yields the IAPO update. This design provably reduces expected completion length (see Section 4) while achieving satisfying accuracy. Next, we analyze the fundamental challenges of conditional MI computation and design a tailored module to tackle them.

### 3.3. Efficient Conditional MI Estimation Module

#### 3.3.1. EARLY-EXIT-BASED CONDITIONAL MI ESTIMATOR

The informativeness metric proposed in Section 3.2.1, defined as the conditional MI $I(y_i; o_{i,t} \mid q, o_{i,<t})$, is not directly observable during training. This is because, computing this quantity requires access to the actual posterior distribution $p(y_i \mid q, o_{i,\leq t})$, which cannot be obtained directly from the model's autoregressive generation process. To address this challenge, we leverage the information-theoretic relationship between conditional MI and entropy:

$$I(y_i; o_{i,t} \mid q, o_{i,<t}) = H(y_i \mid q, o_{i,<t}) - H(y_i \mid q, o_{i,\leq t}).$$

This formulation expresses the informativeness of the token $o_{i,t}$ as the reduction in uncertainty about the answer $y_i$ after observing $o_{i,t}$, given the preceding context. To estimate this value efficiently, we propose an early-exit–based approach that leverages the model's ability to generate answers from intermediate reasoning states. For each prefix $(q, o_{i,\leq t})$, we append a *lightweight answer-generation postfix prompt* (e.g., "`</think><answer>`") that instructs the model to immediately produce the final answer without further reasoning. Specifically, we obtain two answer distributions. First, before token $o_{i,t}$, we feed the prefix $(q, o_{i,<t})$ concatenated with the postfix prompt to the LLM and extract the probability distribution over answers from the logits at the final token position, yielding $H(y_i \mid q, o_{i,<t})$. Second, after generating

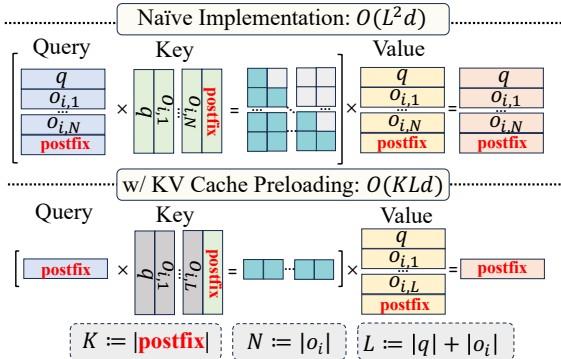

*Figure 4.* Illustration of the naive implementation and KV cache preloading technique in conditional MI estimation. We highlight the key, query, and values of the prompt postfix in red. We show the time complexities in the right, where $K$ is the length of the prompt postfix, $N$ is $|o_i|$, $L$ is $|q| + |o_i|$, and $d$ is the embedding dimension of the LLM. The technique is significantly faster than the naive implementation since the condition $K \ll L$ holds.

the token $o_t$, we similarly feed $(q, o_{i,\leq t})$ with the postfix prompt to obtain $H(y_i \mid q, o_{i,\leq t})$. The conditional MI is then approximated as $H(y_i \mid q, o_{i,<t}) - H(y_i \mid q, o_{i,\leq t})$.

This early-exit estimator provides a principled approximation by measuring how much the model's uncertainty about the final answer decreases after generating token $o_{i,t}$. Tokens that substantially reduce answer entropy receive higher informativeness level scores, while redundant tokens that provide little new information receive lower scores.

#### 3.3.2. TRAINING ACCELERATION TECHNIQUES

Naively estimating the conditional MI for all tokens in a completion is computationally prohibitive: Each token requires two separate forward passes through the LLM, i.e., one before and one after generating the token, to compute the entropy difference $H(y_i \mid q, o_{i,<t}) - H(y_i \mid q, o_{i,\leq t})$. For a completion of length $L$, the cumulative time complexity becomes $O(\sum_{l=1}^L l^2 d) = O(L^3 d)$, where $d$ is the model's embedding dimension. This cubic scaling makes the approach computationally intractable for long reasoning.

**KV-Cache Preloading.** We propose *KV-cache preloading* for fast conditional MI estimation. Specifically, we first perform *a single forward pass* over the full completion $(q, o_i)$ and store the resulting key–value (KV) caches at all transformer layers. These cached states fully encode the autoregressive prefixes $(q, o_{i,\leq t})$ for every token position $t$. Crucially, when estimating the entropy at an intermediate position, the model is *not* re-invoked on the textual prefix $(q, o_{i,<t})$. Instead, the forward pass takes as input only the preloaded KV cache corresponding to the prefix and the short postfix prompt (e.g., "`</think><answer>`") used to elicit an answer distribution. Therefore, all attention computations over the shared prefix tokens are reused exactly,

and the cost of each entropy evaluation depends *only* on the postfix length, rather than the full prefix length. This design eliminates redundant prefix processing entirely of conditional MI estimation from requiring full forward passes per token to lightweight cache continuations.

**Chunk-wise LLM Forwarding.** While KV-cache preloading removes redundant prefix computation, invoking the LLM sequentially for each token position would still incur nontrivial overhead. To further improve efficiency, we introduce chunk-wise LLM forwarding. This technique partitions each completion into contiguous token chunks and estimates conditional MIs for all tokens within a chunk simultaneously using a single batched forward pass. For all token positions within each chunk, we reuse the same KV cache states for the shared prefix and batch the corresponding postfix continuations. Chunk-wise forwarding amortizes the costs of model invocations and memory accesses across multiple tokens, substantially reducing per-token overhead.

*Remark on Time Complexity Improvements.* The proposed KV-cache preloading technique substantially reduces computational overhead. By caching intermediate states, we eliminate redundant prefix computations and reduce the complexity to $O((K^3 + L^2)d)$, where $K$ denotes the length of the postfix prompt ("`</think><answer>`") and $L$ is $|o| + |q|$. Since $K \ll L$ in practice, this represents a substantial improvement over the naive $O(L^3 d)$ approach.

Our chunk-wise LLM forwarding further amortizes costs by batching token evaluations. By partitioning the sequence into $C$ chunks and processing multiple positions simultaneously, we achieve an effective complexity of $O(K^3 d \cdot L/C + K^2 L^2 d/(2C) + L^2 d)$. This design reduces the per-token overhead by a factor of $C$ while maintaining exact computation of informativeness level scores, making `IAPO` practical for training with long reasoning sequences.

# 4. Theoretical Analysis

Here, we show that our proposed `IAPO` reduces completion lengths and enables well-adjusted RL exploration, thereby maintaining the reasoning accuracy of the LLMs.

## 4.1. Completion Lengths Reduction

We show that, under a small policy step, the difference in expected completion length between `IAPO` and vanilla GRPO (Shao et al., 2024) depends on the covariance between the completion length $L(o)$ and a key quantity $S(o)$. Here, $S(o)$ captures the total informativeness-weighted policy gradient signal accumulated across all tokens in $o$.

**Theorem 4.1.** *Given an LLM $\pi_0$, let $L_{GRPO}$ and $L_{IAPO}$ denote the expected completion lengths under $\pi_{GRPO}$ and $\pi_{IAPO}$, which are one-step updated policy models given by GRPO and `IAPO` upon the original policy $\pi_0$, respectively.*

*For sufficiently small step size $\eta$, we have*

$$L_{IAPO} - L_{GRPO} \propto \text{Cov}_{o \sim p_{GRPO}}(L(o), S(o)), \quad (5)$$

*where $S(o)$ is the informativeness-weighted accumulated token-level gradient induced by the `IAPO` advantage.*

Significantly, $S(o)$ is directly related to the average token-wise informativeness (see Appendix B for details): when a completion contains a high proportion of informative tokens, $S(o)$ is large; conversely, if it includes many low-informativeness tokens (e.g., redundant verifications), $S(o)$ is small. Thus, $S(o)$ can be characterized by the average token-wise informativeness. Therefore, we can justify that the completion length $L(o)$ and the term $S(o)$ are negatively correlated, as evidenced by the negative correlation between the average token-wise informativeness and $L(o)$.

**Corollary 4.2.** *Given a query $q$ and an LLM, if the average informativeness per token decreases monotonically as a function of completion length, then for sufficiently small policy update step size, we have $L_{IAPO} < L_{GRPO}$.*

This assumption is mild and natural because, as completions grow longer, they tend to accumulate more low-informativeness tokens, leading $S(o)$ to decrease. This establishes the negative covariance in Theorem 4.1, ensuring that `IAPO` reduces the expected completion length.

## 4.2. Exploration Adjustment

`IAPO` additionally incorporates a token-level term $c_{i,t} = \pi_{\theta_{old}}(o_{i,t} \mid q, o_{i,<t})$ to regulate exploration. To analyze how it affects exploration, we define the degree of exploration of an LLM $\pi_0$ under an input sequence $s := (q, o_{i,<t})$ as the entropy of its prediction distribution, i.e., $H(\pi_0(\cdot \mid s))$.

For the policy $\pi_{IAPO}$ obtained by one policy-gradient update from $\pi_0$, it holds that (Cui et al., 2025)

$$\begin{aligned} &H(\pi_{IAPO}(\cdot \mid s)) - H(\pi_0(\cdot \mid s)) \\ &\approx -\eta \, \text{Cov}_{o_t \sim \pi_0(\cdot|s)}(\log \pi_0(o_t \mid s), A(s, o_t)). \end{aligned} \quad (6)$$

where $\eta$ is the policy update step size, $A(s, o_t)$ is the advantage assigned to token $o_t$ given the context $s$. When the completion prefixed by $s$ yields a correct answer, we set $A(s, o_t) = \pi_0(o_t \mid s)$. In this case, we have the covariance

$$\text{Cov}_{o_t \sim \pi_0(\cdot|s)}(\log \pi_0(o_t \mid s), \pi_0(o_t \mid s)) > 0,$$

which, by Eq. (6), implies $H(\pi_{IAPO}(\cdot \mid s)) < H(\pi_0(\cdot \mid s))$. Thus, exploration is suppressed, encouraging the policy to remain close to the current correct reasoning trajectory.

Conversely, when the completion is incorrect, we set $A(s, o_t) = -\pi_0(o_t \mid s)$, which reverses the sign of the covariance term. As a result, Eq. (6) yields $H(\pi_{IAPO}(\cdot \mid s)) > H(\pi_0(\cdot \mid s))$, thereby increasing the policy entropy and promoting exploration of alternative potentially effective reasoning trajectories that may lead to correct answers.

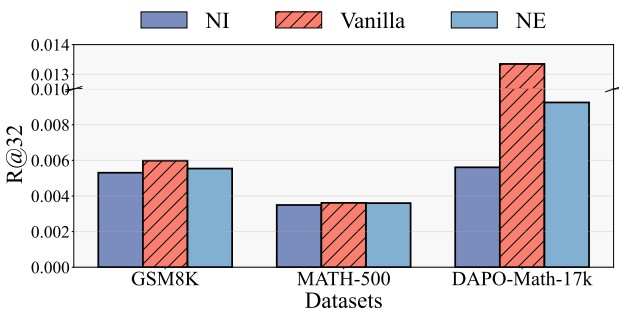

*Figure 5.* Ablation Study for IAPO.

# 5. Empirical Study

We first introduce the experiment setup. Then, we present the evaluation results to answer the following research questions: **RQ1**: How well can IAPO improve the reasoning efficiency compared with the state-of-the-art token-efficient LLM post-training baselines? **RQ2**: To what extent does each component of IAPO contribute to the overall reasoning token-efficiency? **RQ3**: How do hyperparameters such as the coefficient between the $s_{i,t}$ and $c_{i,t}$ affect IAPO's performance? **RQ4**: Is there a case study to show how IAPO is more token-efficient compared to other baselines?

## 5.1. Experiment Settings

Here, we introduce our experiment settings. For more details such as hardware information, please refer to Appendix C.

**Datasets.** We evaluate on three representative mathematics reasoning datasets commonly used for LLM post-training. GSM8K (Cobbe et al., 2021) contains grade-school–level arithmetic word problems requiring multi-step numerical reasoning. MATH-500 (Lightman et al., 2024) is a competition-level mathematics dataset. DAPO-Math-17k (Yu et al., 2025) is a large-scale, diverse math dataset featuring longer solutions than GSM8K and MATH-500.

**Baselines.** We adopt state-of-the-art LLM post-training methods for token efficiency as baselines. Specifically, (1) DAPO (Yu et al., 2025) utilizes an overlong reward shaping technique that assigns zero advantage to overlong sequences. (2) GFPO (Shrivastava et al., 2025) samples more completions during group sampling and only assigns rewards to the shortest completions. (3) GTPO (Tan et al., 2025) rewards tokens with high entropy in correct completions while penalizing tokens with low entropy in incorrect completions. (4) S-GRPO (Lee & Tong, 2025) assigns zero advantage to tokens in a completion whose index is larger than a threshold.

**Evaluation Metrics.** We follow well-established work (Yu et al., 2025) and adopt Pass@$k$ as the metric for evaluating model effectiveness, which is the percentage of evaluation samples that the LLM answers correctly with $k$ trials. Correspondingly, we utilize Length@$k$, the average length of

the $k$ completions across all test queries. Finally, we design a token-efficiency metric, Ratio@$k$, which is the ratio of Pass@$k$ to Length@$k$, indicating how effective each token is at producing the correct answer.

**Implementation Details.** We optimize the LLMs using AdamW (Loshchilov & Hutter, 2019) with a learning rate of $1 \times 10^{-6}$ and a decay factor of $0.5$. The completion group size $G$ (see Section 2) is set to 8, and the coefficient for KL-divergence $D_{\text{KL}}(\pi_\theta \| \pi_{\text{ref}})$ is fixed at $0.001$. Please see additional implementation details in Appendix C.

## 5.2. Effectiveness & Efficiency of IAPO

We answer **RQ1** by evaluating the effectiveness and efficiency of our proposed IAPO, as shown in Table 1. We observe that: (1) *Efficiency.* When achieving comparable Pass@$k$ performance, IAPO consistently generates the fewest tokens across datasets and model scales, outperforming all baselines in terms of reasoning length. (2) *Effectiveness.* IAPO attains the highest reasoning effectiveness (Pass@$k$) across almost all experimental settings. In particular, we achieve a Pass@32 gain of more than 0.06 over the runner-up in configurations such as the Qwen2.5-1.5B-Instruct (Team, 2024) model and MATH-500 (Lightman et al., 2024) dataset. (3) *Effectiveness–Efficiency Trade-off.* When evaluated using the token-efficiency metric Ratio@$k$, IAPO achieves the best (or runner-up) across all settings.

***Remark.*** We also evaluate IAPO on non-mathematical reasoning tasks (e.g., **commonsense reasoning**) to assess its generalizability; see Appendix C.2.1. Our results show that IAPO also achieves the best token efficiency on these tasks. Additionally, beyond measuring completion length as an efficiency indicator, we also measure **wall-clock inference time** and find that post-training with IAPO reduces inference time by more than 11% (see Appendix C.2.3).

## 5.3. Ablation Study

Here, we address **RQ2** by measuring the contribution of different components of our proposed IAPO using two variants: (1) **IAPO-NI**: removes the token-wise advantage assignment based on token informativeness level (the conditional MI term). (2) **IAPO-NE**: replaces the early-exit-based informativeness level estimator with next-token entropy reduction term, i.e., $H(o_{i,t} \mid q, o_{i,<t}) - H(o_{i,t+1} \mid q, o_{i,\leq t})$, where the entropies are those of the LLM's probability distributions when generating $o_{i,t}$ and $o_{i,t+1}$ given $\{q, o_{i,<t}\}$ and $\{q, o_{i,\leq t}\}$, respectively. We report Ratio@32 for these variants using Qwen2.5-0.5B-Instruct (Team, 2024; Yang et al., 2024) in Fig. 5. We observe that: (1) IAPO's token efficiency degrades when removing the informativeness level term or substituting it with next-token prediction entropies. (2) **IAPO-NE** has higher token efficiency than **IAPO-NI**, showing that incorporating conditional MI is promising.

*Table 1.* Comparison of IAPO with baselines across three mathematical reasoning datasets and three LLMs in various scales. P@$k$, L@$k$, and R@$k$ denote Pass@$k$, Length@$k$, and the ratio Pass@$k$/Length@$k$, respectively. The optimal R@$k$s are in **bold**, the runner-ups are underlined. Results are shown for $k \in \{16, 32\}$; for additional results with $k \in \{2, 4, 8\}$, please see Appendix C.2.2.

| Method | P@16 | P@32 | L@16 | L@32 | R@16 | R@32 |
|---|---|---|---|---|---|---|
| **Dataset GSM8K (Cobbe et al., 2021)** | | | | | | |
| Qwen2.5-0.5B-Instruct | 0.4261±0.0099 | 0.5722±0.0081 | 148.04±3.51 | 149.64±6.01 | $2.88 \times 10^{-3}$ | $3.82 \times 10^{-3}$ |
| + DAPO (Yu et al., 2025) | 0.8605±0.0006 | 0.9050±0.0004 | 172.95±0.27 | 172.53±0.09 | $4.98 \times 10^{-3}$ | $5.25 \times 10^{-3}$ |
| + GFPO (Shrivastava et al., 2025) | 0.8590±0.0056 | 0.9052±0.0043 | 216.26±0.31 | 215.99±0.12 | $3.97 \times 10^{-3}$ | $4.19 \times 10^{-3}$ |
| + GTPO (Tan et al., 2025) | 0.8519±0.0041 | 0.9085±0.0028 | 243.17±0.26 | 243.21±0.32 | $3.50 \times 10^{-3}$ | $3.74 \times 10^{-3}$ |
| + S-GRPO (Lee & Tong, 2025) | 0.8489±0.0077 | 0.9017±0.0009 | 159.70±0.13 | 159.49±0.20 | $\underline{5.32 \times 10^{-3}}$ | $\underline{5.65 \times 10^{-3}}$ |
| + IAPO (ours) | 0.8519±0.0084 | 0.8979±0.0068 | 150.37±0.22 | 150.15±0.24 | $\mathbf{5.67 \times 10^{-3}}$ | $\mathbf{5.98 \times 10^{-3}}$ |
| Qwen2.5-1.5B-Instruct | 0.8160±0.0090 | 0.8941±0.0029 | 152.66±1.53 | 152.59±3.76 | $5.35 \times 10^{-3}$ | $\underline{5.86 \times 10^{-3}}$ |
| + DAPO (Yu et al., 2025) | 0.9479±0.0059 | 0.9664±0.0050 | 169.31±0.37 | 169.80±0.26 | $\underline{5.60 \times 10^{-3}}$ | $5.69 \times 10^{-3}$ |
| + GFPO (Shrivastava et al., 2025) | 0.9510±0.0029 | 0.9725±0.0022 | 203.85±0.65 | 204.05±0.58 | $4.67 \times 10^{-3}$ | $4.77 \times 10^{-3}$ |
| + GTPO (Tan et al., 2025) | 0.9497±0.0028 | 0.9674±0.0012 | 262.63±0.62 | 262.49±0.78 | $3.62 \times 10^{-3}$ | $3.69 \times 10^{-3}$ |
| + S-GRPO (Lee & Tong, 2025) | 0.9558±0.0032 | 0.9735±0.0027 | 181.07±0.08 | 180.85±0.16 | $5.28 \times 10^{-3}$ | $5.38 \times 10^{-3}$ |
| + IAPO (ours) | 0.9512±0.0034 | 0.9707±0.0028 | 163.32±0.46 | 163.51±0.56 | $\mathbf{5.82 \times 10^{-3}}$ | $\mathbf{5.94 \times 10^{-3}}$ |
| Qwen2.5-7B-Instruct | 0.9793±0.0009 | 0.9851±0.0016 | 157.08±1.76 | 156.43±1.05 | $6.23 \times 10^{-3}$ | $6.30 \times 10^{-3}$ |
| + DAPO (Yu et al., 2025) | 0.9778±0.0026 | 0.9816±0.0026 | 160.36±0.73 | 159.83±0.21 | $6.10 \times 10^{-3}$ | $6.14 \times 10^{-3}$ |
| + GFPO (Shrivastava et al., 2025) | 0.9798±0.0016 | 0.9853±0.0009 | 160.36±0.99 | 160.50±0.57 | $6.11 \times 10^{-3}$ | $6.14 \times 10^{-3}$ |
| + GTPO (Tan et al., 2025) | 0.9765±0.0006 | 0.9826±0.0006 | 192.99±0.10 | 192.85±0.10 | $5.06 \times 10^{-3}$ | $5.10 \times 10^{-3}$ |
| + S-GRPO (Lee & Tong, 2025) | 0.9790±0.0020 | 0.9843±0.0004 | 147.67±0.70 | 147.40±0.42 | $\underline{6.63 \times 10^{-3}}$ | $\underline{6.68 \times 10^{-3}}$ |
| + IAPO (ours) | 0.9735±0.0012 | 0.9805±0.0009 | 84.03±0.12 | 83.97±0.04 | $\mathbf{1.16 \times 10^{-2}}$ | $\mathbf{1.17 \times 10^{-2}}$ |
| **Dataset MATH-500 (Lightman et al., 2024)** | | | | | | |
| Qwen2.5-0.5B-Instruct | 0.3199±0.0172 | 0.4478±0.0048 | 253.22±7.36 | 257.55±6.55 | $1.26 \times 10^{-3}$ | $1.74 \times 10^{-3}$ |
| + DAPO (Yu et al., 2025) | 0.4714±0.0172 | 0.5488±0.0048 | 162.41±1.57 | 162.30±1.31 | $2.90 \times 10^{-3}$ | $3.38 \times 10^{-3}$ |
| + GFPO (Shrivastava et al., 2025) | 0.5118±0.0048 | 0.5690±0.0095 | 161.56±1.66 | 161.55±1.90 | $\underline{3.17 \times 10^{-3}}$ | $3.52 \times 10^{-3}$ |
| + GTPO (Tan et al., 2025) | 0.4714±0.0252 | 0.5354±0.0082 | 472.01±1.23 | 472.58±0.20 | $\underline{9.99 \times 10^{-4}}$ | $1.13 \times 10^{-3}$ |
| + S-GRPO (Lee & Tong, 2025) | 0.4680±0.0423 | 0.5253±0.0378 | 103.37±0.39 | 103.31±0.12 | $\mathbf{4.53 \times 10^{-3}}$ | $\mathbf{5.08 \times 10^{-3}}$ |
| + IAPO (ours) | 0.5118±0.0172 | 0.5960±0.0218 | 164.42±0.09 | 164.96±0.41 | $3.11 \times 10^{-3}$ | $\underline{3.61 \times 10^{-3}}$ |
| Qwen2.5-1.5B-Instruct | 0.5152±0.0143 | 0.6330±0.0390 | 296.65±3.09 | 296.89±3.28 | $1.74 \times 10^{-3}$ | $2.13 \times 10^{-3}$ |
| + DAPO (Yu et al., 2025) | 0.6869±0.0082 | 0.7508±0.0190 | 327.34±1.24 | 326.89±1.11 | $2.10 \times 10^{-3}$ | $2.30 \times 10^{-3}$ |
| + GFPO (Shrivastava et al., 2025) | 0.6465±0.0165 | 0.7003±0.0095 | 374.95±2.54 | 374.47±0.71 | $1.72 \times 10^{-3}$ | $1.87 \times 10^{-3}$ |
| + GTPO (Tan et al., 2025) | 0.5084±0.0172 | 0.5421±0.0126 | 496.05±0.69 | 495.83±0.33 | $1.02 \times 10^{-3}$ | $1.09 \times 10^{-3}$ |
| + S-GRPO (Lee & Tong, 2025) | 0.7273±0.0143 | 0.7576±0.0218 | 276.75±1.02 | 278.08±0.37 | $\underline{2.63 \times 10^{-3}}$ | $\underline{2.72 \times 10^{-3}}$ |
| + IAPO (ours) | 0.7172±0.0165 | 0.7576±0.0165 | 265.12±1.35 | 264.28±0.38 | $\mathbf{2.71 \times 10^{-3}}$ | $\mathbf{2.87 \times 10^{-3}}$ |
| Qwen2.5-7B-Instruct | 0.6734±0.0126 | 0.7003±0.0190 | 341.46±2.43 | 340.73±1.95 | $1.97 \times 10^{-3}$ | $2.06 \times 10^{-3}$ |
| + DAPO (Yu et al., 2025) | 0.7811±0.0048 | 0.8148±0.0048 | 331.61±1.34 | 331.22±0.54 | $2.36 \times 10^{-3}$ | $\underline{2.46 \times 10^{-3}}$ |
| + GFPO (Shrivastava et al., 2025) | 0.7340±0.0048 | 0.7811±0.0126 | 344.24±0.39 | 344.59±0.25 | $2.13 \times 10^{-3}$ | $\underline{2.27 \times 10^{-3}}$ |
| + GTPO (Tan et al., 2025) | 0.6128±0.0126 | 0.6498±0.0252 | 453.00±0.47 | 453.14±0.56 | $1.35 \times 10^{-3}$ | $1.43 \times 10^{-3}$ |
| + S-GRPO (Lee & Tong, 2025) | 0.8013±0.0126 | 0.8215±0.0172 | 335.58±0.44 | 335.48±0.37 | $\underline{2.39 \times 10^{-3}}$ | $2.45 \times 10^{-3}$ |
| + IAPO (ours) | 0.7744±0.0126 | 0.8047±0.0048 | 303.04±1.30 | 302.24±0.75 | $\mathbf{2.56 \times 10^{-3}}$ | $\mathbf{2.66 \times 10^{-3}}$ |
| **Dataset DAPO-Math-17k (Yu et al., 2025)** | | | | | | |
| Qwen2.5-0.5B-Instruct | 0.0867±0.0249 | 0.1667±0.0094 | 374.85±8.83 | 378.65±8.64 | $2.31 \times 10^{-4}$ | $4.40 \times 10^{-4}$ |
| + DAPO (Yu et al., 2025) | 0.2467±0.0094 | 0.2733±0.0094 | 32.59±0.18 | 32.22±0.12 | $7.57 \times 10^{-3}$ | $8.48 \times 10^{-3}$ |
| + GFPO (Shrivastava et al., 2025) | 0.2067±0.0411 | 0.2667±0.0499 | 16.76±0.19 | 16.79±0.13 | $\mathbf{1.23 \times 10^{-2}}$ | $\mathbf{1.59 \times 10^{-2}}$ |
| + GTPO (Tan et al., 2025) | 0.1400±0.0163 | 0.1733±0.0094 | 15.87±0.02 | 15.86±0.01 | $8.82 \times 10^{-3}$ | $1.09 \times 10^{-2}$ |
| + S-GRPO (Lee & Tong, 2025) | 0.1733±0.0189 | 0.2667±0.0094 | 17.57±0.10 | 17.64±0.03 | $9.86 \times 10^{-3}$ | $\underline{1.51 \times 10^{-2}}$ |
| + IAPO (ours) | 0.1933±0.0094 | 0.2333±0.0094 | 17.52±0.24 | 17.48±0.09 | $\underline{1.10 \times 10^{-2}}$ | $1.33 \times 10^{-2}$ |
| Qwen2.5-1.5B-Instruct | 0.1800±0.0163 | 0.2333±0.0249 | 387.59±9.38 | 386.86±6.84 | $4.64 \times 10^{-4}$ | $6.03 \times 10^{-4}$ |
| + DAPO (Yu et al., 2025) | 0.2133±0.0249 | 0.2800±0.0283 | 38.36±0.97 | 38.35±0.93 | $5.56 \times 10^{-3}$ | $7.30 \times 10^{-3}$ |
| + GFPO (Shrivastava et al., 2025) | 0.2133±0.0340 | 0.2733±0.0094 | 64.51±1.06 | 64.59±1.22 | $3.31 \times 10^{-3}$ | $4.23 \times 10^{-3}$ |
| + GTPO (Tan et al., 2025) | 0.2000±0.0163 | 0.2533±0.0189 | 18.88±0.04 | 18.87±0.02 | $\mathbf{1.06 \times 10^{-2}}$ | $\mathbf{1.34 \times 10^{-2}}$ |
| + S-GRPO (Lee & Tong, 2025) | 0.2200±0.0327 | 0.2800±0.0163 | 60.36±0.75 | 60.11±0.52 | $3.64 \times 10^{-3}$ | $4.66 \times 10^{-3}$ |
| + IAPO (ours) | 0.2133±0.0411 | 0.3200±0.0000 | 41.39±0.57 | 41.46±0.56 | $5.15 \times 10^{-3}$ | $\underline{7.72 \times 10^{-3}}$ |
| Qwen2.5-7B-Instruct | 0.3267±0.0340 | 0.3800±0.0566 | 449.39±4.05 | 446.26±2.14 | $7.27 \times 10^{-4}$ | $8.52 \times 10^{-4}$ |
| + DAPO (Yu et al., 2025) | 0.3467±0.0249 | 0.4200±0.0327 | 445.48±1.03 | 444.91±0.79 | $7.78 \times 10^{-4}$ | $9.44 \times 10^{-4}$ |
| + GFPO (Shrivastava et al., 2025) | 0.3933±0.0189 | 0.4800±0.0327 | 424.36±0.55 | 424.10±0.49 | $\underline{9.27 \times 10^{-4}}$ | $\underline{1.13 \times 10^{-3}}$ |
| + GTPO (Tan et al., 2025) | 0.1667±0.0249 | 0.1933±0.0340 | 519.29±0.26 | 519.03±0.11 | $3.21 \times 10^{-4}$ | $3.72 \times 10^{-4}$ |
| + S-GRPO (Lee & Tong, 2025) | 0.4067±0.0189 | 0.4667±0.0094 | 398.34±0.13 | 398.02±0.28 | $\mathbf{1.02 \times 10^{-3}}$ | $\mathbf{1.17 \times 10^{-3}}$ |
| + IAPO (ours) | 0.3867±0.0189 | 0.4267±0.0249 | 430.06±1.68 | 429.62±0.94 | $8.99 \times 10^{-4}$ | $9.93 \times 10^{-4}$ |

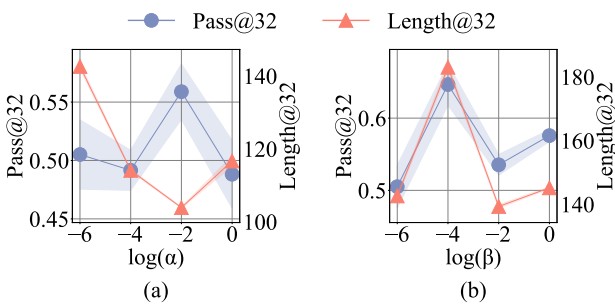

*Figure 6.* Parameter Analysis for IAPO.

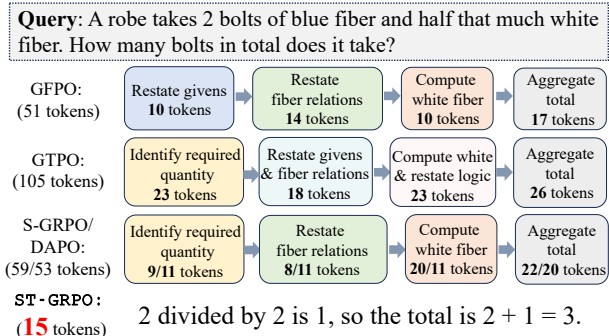

*Figure 7.* Case study: We show reasoning trajectories generated by Qwen2.5-7B-Instruct (Team, 2024) post-trained by IAPO and baselines for the same query from GSM8K (Cobbe et al., 2021) dataset. IAPO produces a substantially shorter yet sufficient completion by concentrating generation on high-informativeness tokens.

### 5.4. Parameter Analysis

We answer **RQ3** by examining the effects of two key hyperparameters in IAPO, i.e., the $\alpha$ and $\beta$ in Equ. 4. We report Pass@32 and Length@32 while varying $\alpha$ (or $\beta$) in $\{10^{-6}, 10^{-4}, 10^{-2}, 1\}$, with the other coefficient fixed at $10^{-6}$, in Fig. 6(a) and Fig. 6(b), respectively on Qwen2.5-0.5B-Instruct (Team, 2024) on MATH-500 dataset (Lightman et al., 2024). We observe that (1) As $\alpha$ increases, reasoning length decreases monotonically before $10^{-2}$, demonstrating the effectiveness of the informativeness level term in reducing verbosity. Although Pass@32 sometimes decreases, its degradation is substantially slower, indicating a favorable effectiveness-efficiency trade-off. (2) As $\beta$ increases, Pass@32 can significantly improve, but reasoning length also grows sharply. This suggests that a larger $\beta$ encourages exploration of reasoning, enhancing accuracy at the cost of substantially increased token consumption.

### 5.5. Case Study

We answer **RQ4** by examining reasoning completions generated by different methods on a query from the GSM8K dataset (Cobbe et al., 2021), shown in Fig. 7. We make two key observations: (1) IAPO produces *substantially shorter reasoning* by concentrating on informative tokens. For the query in Fig. 7, IAPO generates the correct answer with only 15 tokens, which is 3.4× to 7× shorter than baselines

(51–105 tokens). (2) *Baselines exhibit distinct verbosity patterns*. GFPO (Shrivastava et al., 2025) restates problem givens (10 tokens) and relations (14 tokens). GTPO (Tan et al., 2025) repeatedly restate givens or its high-level reasoning strategies. DAPO (Yu et al., 2025) and S-GRPO (Lee & Tong, 2025) share similar reasoning logic; they verbosely identify what quantities the problem requires.

## 6. Related Work

### 6.1. RL-based Post-training for LLMs

RL-based post-training methods improve LLM reasoning by optimizing sequence-level rewards, typically derived from correctness or preference signals, using policy-gradient algorithms such as PPO (Schulman et al., 2017) and GRPO (Shao et al., 2024). Variants including RLHF (Ouyang et al., 2022; Christiano et al., 2017), RLAIF (Lee et al., 2024;?), and DPO (Rafailov et al., 2023) simplify reward modeling or replace human feedback with AI-based evaluators, improving scalability and stability (Bai et al., 2022; Rafailov et al., 2023). More recent reasoning-focused approaches apply outcome-based or verifier-guided rewards, encouraging correct final answers or stepwise validity without explicit supervision of reasoning chains (Uesato et al., 2022; Lightman et al., 2024). Although these methods enhance reasoning, their reasoning processes are verbose, consuming large amounts of tokens and generation time.

### 6.2. Reasoning Efficiency of LLMs

Two main approaches improve LLM reasoning efficiency. The first (Cheng & Van Durme, 2024; Su et al., 2025; Xu et al., 2025; Shen et al., 2025) replaces verbose chain-of-thought with continuous embeddings—implicit reasoning tokens—bypassing explicit text generation and reducing reasoning length. While efficient, these methods sacrifice interpretability by using embeddings rather than explicit text, limiting their use in high-stakes scenarios that require transparent rationales. The second approach, including ours, reduces reasoning length via advantage shaping under GRPO (Shao et al., 2024). However, existing methods uniformly favor shorter completions (Shrivastava et al., 2025; Yu et al., 2025) or apply heuristic position-based decay (Dai et al., 2025; Lee & Tong, 2025), without distinguishing informative from redundant tokens during optimization. Consequently, they fail to align reasoning accuracy with token efficiency.

## 7. Conclusion

We present IAPO, a token-efficient post-training framework that assigns advantages based on information theoretic measurements, achieving up to 36% token reduction while preserving reasoning accuracy. IAPO provides a principled foundation for efficient LLM reasoning optimization.

## Acknowledgements

This work was supported in part by the National Science Foundation (NSF) under Grants IIS-2144209, IIS-2223769, BCS-2228534, and CMMI-2411248, and by the Office of Naval Research (ONR) under Grant N000142412636.

## Impact Statement

This paper advances machine learning by improving the efficiency of large language model reasoning. By reducing reasoning token consumption by up to 47% while maintaining accuracy, `IAPO` lowers the compute and energy required for LLM inference, supporting more sustainable AI deployment and reducing the carbon footprint of large-scale usage. The resulting cost reductions also make advanced reasoning capabilities more accessible to researchers, organizations, and individuals with limited computational budgets, helping to democratize AI and lower barriers to entry for educational, scientific, and startup applications. Beyond efficiency, `IAPO`'s information-theoretic framework offers principled insights into which reasoning steps contribute most to correct answers, supporting more interpretable AI systems and better oversight. Finally, the methodological contributions of information-aware advantage shaping and efficient conditional mutual information estimation are general and may benefit other areas of machine learning beyond language model reasoning.

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

# Appendix

**Organization.** This appendix is organized as follows: Section A presents supplementary related work on the application of information theory in reinforcement learning and natural language research, providing a broader context for our information-theoretic approach to token-efficient reasoning. Section B presents the theoretical foundations and proofs, including the complete derivation of Theorem 4.1, demonstrating how `IAPO` reduces expected completion length through covariance-based analysis, along with Corollary 4.2 establishing conditions under which `IAPO` achieves shorter completions than GRPO. Section C provides supplementary experimental results, including detailed implementation specifications (hardware configuration, distributed computing setup, hyperparameters), comprehensive evaluation results across additional metrics (Pass@2, Pass@4, Pass@8 with corresponding Length@$k$ and Ratio@$k$ measurements), training dynamics visualizations showing the evolution of correctness rewards, completion lengths, and efficiency ratios throughout the post-training process, and extensive case studies with token-level heatmaps demonstrating how our conditional mutual information estimator successfully identifies informative tokens while assigning low scores to redundant content such as meta-commentary and repetitive verifications. These supplementary materials provide complete transparency into our experimental methodology and offer deeper insights into `IAPO`'s mechanisms for achieving token-efficient reasoning.

## A. Supplementary Related Work

### A.1. Application of Information Theory in Reinforcement Learning

Information theory has been used to analyze fundamental questions in reinforcement learning. Chen & Jiang (2019) use information theory to prove lower bounds in batch RL, showing that certain assumptions about data coverage and function approximation are necessary: without them, no algorithm can learn efficiently even with unlimited data. Still & Precup (2012) shows that Boltzmann exploration naturally emerges when you minimize the information cost (mutual information) of a policy while maintaining a target reward level. Schmidhuber (2015) proposes that when a model and controller share algorithmic mutual information, the controller can learn more efficiently by querying the model. While these works apply information theory to understand what's possible in RL (lower bounds), how to explore efficiently, and how to leverage learned models, they focus on different problems than token efficiency in LLM reasoning. Our work takes a different direction: we use conditional mutual information to measure how much each token contributes to getting the correct answer, which lets us train models to generate shorter but still accurate reasoning chains.

### A.2. Token-Level Advantage Assignment in RL post-training for LLMs

Traditional RL-based post-training methods such as PPO (Schulman et al., 2017) and GRPO (Shao et al., 2024) face a fundamental challenge in credit assignment: determining which specific tokens contribute to the final outcome. While standard GRPO assigns uniform sequence-level advantages to all tokens (Shao et al., 2024), recent work has explored more sophisticated token-level strategies. PPO employs a learned value network for token-level advantages via GAE (Schulman et al., 2016), but this introduces complexity and overfitting risks. Tree-based approaches like TEMPO (Tran et al., 2025) and TreePO (Yang et al., 2025) exploit prefix structures in grouped rollouts to compute nonparametric values at branching points, while methods like T-SPMO (Lee & Tong, 2025) apply token-level prefix matching for fine-grained credit assignment. Another line of work focuses on entropy-based selection: Wang et al. (2025) identifies high-entropy "forking tokens" as critical decision points. Process Reward Models (PRMs) (Zhang et al., 2025) provide step-wise supervision by evaluating intermediate reasoning, with recent work like CAPO (Xie et al., 2025) leveraging LLMs as generative PRMs for token-level critique. In contrast to these approaches that use entropy, tree structure, or learned value functions as proxies for importance, our `IAPO` directly quantifies each token's contribution through conditional mutual information with the final answer, providing an explicit, principled mechanism that measures how much a token reduces uncertainty about correctness rather than merely reflecting generation uncertainty.

## B. Theory

**Theorem 4.1** (Change in expected length = covariance). *Let $L_{\text{GRPO}}$ and $L_{\text{IAPO}}$ denote the expected completion length of a query q under the GRPO and `IAPO` updated policies, respectively:*

$$L_{\text{GRPO}} \triangleq \mathbb{E}_{o \sim \pi_{\theta_{\text{GRPO}}}}[L(o)], \qquad L_{\text{IAPO}} \triangleq \mathbb{E}_{o \sim \pi_{\theta_{\text{IAPO}}}}[L(o)].$$

*Let $\theta_{\text{IAPO}} = \theta_{\text{GRPO}} + \eta\,\Delta\theta_s$, and define, for a trajectory o,*

$$g_t(q, o_{i,t}) \triangleq \nabla_\theta \log \pi_\theta(o_{i,t} \mid q)\Big|_{\theta=\theta_{\text{GRPO}}} \Delta\theta_s, \qquad S(o) \triangleq \sum_{t=1}^{L(o)} g_t(q, o_{i,t}).$$

*Then, for sufficiently small step size $\eta$, we have the first-order relation*

$$\tfrac{1}{\eta}\big(L_{\text{IAPO}} - L_{\text{GRPO}}\big) \;\approx\; \tfrac{d}{d\eta}\mathbb{E}_{o\sim p_\eta}[L(o)]\Big|_{\eta=0} \;=\; \text{Cov}_{o\sim p_{\text{GRPO}}}\big(L(o),\, S(o)\big), \tag{7}$$

*where $p_{\text{GRPO}}(o) = p_{\theta_{\text{GRPO}}}(o)$ is the trajectory distribution under GRPO and $p_\eta$ denotes the trajectory distribution induced by $\theta_\eta = \theta_{\text{GRPO}} + \eta\,\Delta\theta_s$. In particular, the sign of the change in expected completion length is determined by the covariance between $L(o)$ and $S(o)$ under the GRPO policy.*

*Proof.* We first analyze the parameters relationship between $\theta_{\text{IAPO}}$ and $\theta_{\text{GRPO}}$:

$$\begin{aligned}
\theta_{\text{IAPO}} &= \theta + \eta\, \tfrac{1}{G}\sum_{i=1}^{G}\tfrac{1}{|o_i|}\sum_{t=1}^{|o_i|} \hat{A}_{i,t}\,\nabla_\theta \log \pi_\theta(o_{i,t}\mid q, o_{i,<t}) \\
&= \theta + \eta\, \tfrac{1}{G}\sum_{i=1}^{G}\tfrac{1}{|o_i|}\sum_{t=1}^{|o_i|} \tfrac{r_i-\text{mean}(\mathbf{r})}{\text{std}(\mathbf{r})}\,\nabla_\theta \log \pi_\theta(o_{i,t}\mid q, o_{i,<t}) \\
&\quad + \eta\, \tfrac{1}{G}\sum_{i=1}^{G}\tfrac{1}{|o_i|}\sum_{t=1}^{|o_i|} \tfrac{s_{i,t}-\text{mean}(\mathbf{s}_i)}{\text{std}(\mathbf{s}_i)}\,\nabla_\theta \log \pi_\theta(o_{i,t}\mid q, o_{i,<t}) \\
&= \theta_{\text{GRPO}} + \eta\, \tfrac{1}{G}\sum_{i=1}^{G}\tfrac{1}{|o_i|}\sum_{t=1}^{|o_i|} \tfrac{s_{i,t}-\text{mean}(\mathbf{s}_i)}{\text{std}(\mathbf{s}_i)}\,\nabla_\theta \log \pi_\theta(o_{i,t}\mid q, o_{i,<t}).
\end{aligned} \tag{8}$$

The `IAPO` parameters can be written as

$$\theta_{\text{IAPO}} = \theta_{\text{GRPO}} + \eta\,\Delta\theta_s, \qquad \Delta\theta_s = \tfrac{1}{G}\sum_{i=1}^{G}\tfrac{1}{|o_i|}\sum_{t=1}^{|o_i|} \beta_{i,t}\,\nabla_\theta \log \pi_\theta(o_{i,t}\mid q, o_{i,<t})\Big|_{\theta=\theta_{\text{GRPO}}}, \tag{9}$$

where $\beta_{i,t} = \tfrac{s_{i,t}-\text{mean}(\mathbf{s}_i)}{\text{std}(\mathbf{s}_i)}$ is the normalized informativeness score.

We treat $\eta$ as a small scalar so that the `IAPO` update is a small perturbation of $\theta_{\text{GRPO}}$. For any $(q, o_{i,t})$, a first-order Taylor expansion of the log-policy around $\theta_{\text{GRPO}}$ gives

$$\log \pi_{\theta_{\text{IAPO}}}(o_{i,t}\mid q) \;\approx\; \log \pi_{\theta_{\text{GRPO}}}(o_{i,t}\mid q) + \eta\,\nabla_\theta \log \pi_\theta(o_{i,t}\mid q)\Big|_{\theta=\theta_{\text{GRPO}}}\Delta\theta_s. \tag{10}$$

Exponentiating and linearizing $e^x \approx 1 + x$ yields

$$\pi_{\theta_{\text{IAPO}}}(o_{i,t}\mid q) \;\approx\; \pi_{\theta_{\text{GRPO}}}(o_{i,t}\mid q)\big(1 + \eta\,g_t(q, o_{i,t})\big), \tag{11}$$

where we define

$$g_t(q, o_{i,t}) \;\triangleq\; \nabla_\theta \log \pi_\theta(o_{i,t}\mid q)\Big|_{\theta=\theta_{\text{GRPO}}}\Delta\theta_s. \tag{12}$$

For a trajectory $o = (o_1, \ldots, o_{L(o)})$ we have

$$p_\theta(o) = \prod_{t=1}^{L(o)} \pi_\theta(o_{i,t}\mid q), \tag{13}$$

so substituting (11) and linearizing the product gives

$$
\begin{aligned}
p_{\text{IAPO}}(o) = \prod_{t=1}^{L(o)} \pi_{\theta_{\text{IAPO}}}(o_{i,t} \mid q) &\approx \prod_{t=1}^{L(o)} \Big[ \pi_{\theta_{\text{GRPO}}}(o_{i,t} \mid q)\big(1 + \eta\, g_t(q, o_{i,t})\big)\Big] \\
&= \underbrace{\prod_{t=1}^{L(o)} \pi_{\theta_{\text{GRPO}}}(o_{i,t} \mid q)}_{=\, p_{\text{GRPO}}(o)} \prod_{t=1}^{L(o)} \big(1 + \eta\, g_t(q, o_{i,t})\big) \\
&\approx p_{\text{GRPO}}(o)\big(1 + \eta \textstyle\sum_{t=1}^{L(o)} g_t(q, o_{i,t})\big) \;=\; p_{\text{GRPO}}(o)\big(1 + \eta\, S(o)\big),
\end{aligned}
\tag{14}
$$

where we defined $S(o) \triangleq \sum_{t=1}^{L(o)} g_t(q, o_{i,t})$.

The expression in (14) is *unnormalized*. Let

$$
\tilde{p}_\eta(o) \triangleq p_{\text{GRPO}}(o)\big(1 + \eta\, S(o)\big), \qquad Z(\eta) \triangleq \sum_o \tilde{p}_\eta(o) = \sum_o p_{\text{GRPO}}(o)\big(1 + \eta\, S(o)\big).
$$

Then the properly normalized trajectory distribution is

$$
p_\eta(o) \;\triangleq\; \frac{\tilde{p}_\eta(o)}{Z(\eta)} \;=\; \frac{p_{\text{GRPO}}(o)\big(1 + \eta\, S(o)\big)}{1 + \eta\, \mathbb{E}_{o \sim p_{\text{GRPO}}}[S(o)]}.
\tag{15}
$$

For small $\eta$, we can again linearize the denominator to obtain

$$
\begin{aligned}
p_\eta(o) &\approx p_{\text{GRPO}}(o)\big(1 + \eta\, S(o)\big)\big(1 - \eta\, \mathbb{E}_{o' \sim p_{\text{GRPO}}}[S(o')]\big) \\
&\approx p_{\text{GRPO}}(o)\big(1 + \eta\big(S(o) - \mathbb{E}_{o' \sim p_{\text{GRPO}}}[S(o')]\big)\big).
\end{aligned}
\tag{16}
$$

Now consider the expected completion length under $p_\eta$:

$$
\mathbb{E}_{o \sim p_\eta}[L(o)] = \sum_o p_\eta(o)\, L(o).
$$

Using (16) and keeping only first-order terms in $\eta$, we obtain

$$
\begin{aligned}
\mathbb{E}_{o \sim p_\eta}[L(o)] &\approx \sum_o p_{\text{GRPO}}(o)\big(1 + \eta(S(o) - \mathbb{E}_{o' \sim p_{\text{GRPO}}}[S(o')])\big) L(o) \\
&= \mathbb{E}_{o \sim p_{\text{GRPO}}}[L(o)] + \eta\, \mathbb{E}_{o \sim p_{\text{GRPO}}}\Big[\big(S(o) - \mathbb{E}_{o' \sim p_{\text{GRPO}}}[S(o')]\big) L(o)\Big] \\
&= L_{\text{GRPO}} + \eta\Big(\mathbb{E}_{o \sim p_{\text{GRPO}}}[L(o)\, S(o)] - \mathbb{E}_{o \sim p_{\text{GRPO}}}[L(o)]\, \mathbb{E}_{o \sim p_{\text{GRPO}}}[S(o)]\Big) \\
&= L_{\text{GRPO}} + \eta\, \text{Cov}_{o \sim p_{\text{GRPO}}}\big(L(o),\, S(o)\big).
\end{aligned}
\tag{17}
$$

By definition of $p_\eta$, the IAPO expected length corresponds to $\eta$ evaluated at the small update step used in Eq. (9), so to first order in $\eta$, we have

$$
L_{\text{IAPO}} = \mathbb{E}_{o \sim p_\eta}[L(o)] \approx L_{\text{GRPO}} + \eta\, \text{Cov}_{o \sim p_{\text{GRPO}}}\big(L(o),\, S(o)\big).
$$

Rearranging yields

$$
\tfrac{1}{\eta}\big(L_{\text{IAPO}} - L_{\text{GRPO}}\big) \approx \text{Cov}_{o \sim p_{\text{GRPO}}}\big(L(o),\, S(o)\big),
$$

which is exactly Eq. (7). This completes the proof. $\qquad\square$

**Corollary 4.2** (Keyword-aligned updates shrink expected length). *Assume that, under the GRPO policy $\pi_{\theta_{\text{GRPO}}}$: (i) the token-wise scores $s_{i,t}$ satisfy that informative tokens have strictly larger scores than non-informative tokens, and (ii) for trajectories o with comparable semantic quality (same label Y), longer trajectories contain strictly more non-informative tokens. Then there exists $\eta_0 > 0$ such that for all $0 < \eta < \eta_0$, the IAPO update satisfies*

$$
L_{\text{IAPO}} < L_{\text{GRPO}}.
$$

*Proof.* By Theorem 4.1 we have, for small $\eta$,

$$L_{\texttt{IAPO}} = L_{\text{GRPO}} + \eta \, \text{Cov}_{o \sim p_{\text{GRPO}}} \big( L(o), \, S(o) \big) + o(\eta).$$

Conditions (i)–(ii), together with the within-trajectory normalization $\beta_{i,t} = \frac{s_{i,t} - \text{mean}(\mathbf{s}_i)}{\text{std}(\mathbf{s}_i)}$, imply that, at fixed semantic quality, trajectories with larger length $L(o)$ have strictly smaller total informativeness score mass $\sum_t \beta_{i,t}$, and hence strictly smaller $S(o)$. Therefore $L(o)$ and $S(o)$ are negatively correlated under $p_{\text{GRPO}}$, i.e., $\text{Cov}(L(o), S(o)) < 0$. For sufficiently small $\eta$, the first-order term dominates the higher-order terms, which yields $L_{\texttt{IAPO}} < L_{\text{GRPO}}$. $\qquad\square$

## C. Supplementary Experiments

### C.1. Implementation Details

**Remarks on Reasoning Length Measurement.** In our work, we employ different length measurement approaches depending on the context and purpose of each analysis. In Fig. 1, we compare model-generated reasoning with human-written solutions. Since human solutions are not processed through any LLM tokenizer, we use whitespace-delimited word counts for both human and model outputs to enable fair comparison. In Fig. 7, we present reasoning trajectories as natural text to illustrate qualitative differences in verbosity. For interpretability and alignment with the displayed text, we report whitespace-delimited word counts rather than subword tokens provided by the LLMs' tokenizers. In contrast, all quantitative results reported in Table 1 use the native tokenizer of the corresponding LLM, as this reflects actual computational cost and ensures fair comparison across methods applied to the same base model.

**Data Preprocessing.** We use the full GSM8K (Cobbe et al., 2021) and MATH-500 (Lightman et al., 2024) datasets. Due to the large scale of DAPO-Math-17k (Yu et al., 2025) (1.79M samples), we subsample the first 7,000 examples for training and validation. Specifically, the training sets contain 7,473, 400, and 5,600 samples for GSM8K, MATH-500, and DAPO-Math-17k, respectively, while the corresponding validation sets contain 1,319, 100, and 50 samples. For DAPO-Math-17k, the validation set is randomly sampled from the remaining 1,400 examples after the training split.

**Training Configurations & Implementation Details.** In addition to the key settings described in the main paper, we provide further details of our post-training procedure. All baselines and `IAPO` are trained with gradient accumulation steps of 6 for Qwen2.5-0.5B-Instruct and Qwen2.5-1.5B-Instruct, and 8 for Qwen2.5-7B-Instruct (Yang et al., 2024; Team, 2024). Gradient clipping is applied with a maximum norm of 1.0. We tune both the surprise and confidence coefficients over $\{10^{-6}, 10^{-4}, 10^{-2}, 1\}$ and report the best-performing configuration for each dataset–model combination. For Qwen2.5-0.5B-Instruct and Qwen2.5-1.5B-Instruct (Yang et al., 2024; Team, 2024), the per-GPU batch size is set to 32, corresponding to $32/8 = 4$ data items per device for group completion generation. For Qwen2.5-7B-Instruct (Yang et al., 2024; Team, 2024), the per-GPU batch size is reduced to 16, corresponding to $16/8 = 2$ data items per device. We utilize the $H(o_t|q, o_{<t})$ as substitution for the $\pi_\theta(o_t|q, o_{<t})$ for better practical significance. We train 10,000 steps for DAPO-Math-17k (Yu et al., 2025) dataset, 2 epochs for GSM8K (Cobbe et al., 2021) and 152 epochs for MATH-500 dataset (Lightman et al., 2024). We take the model checkpoints with the top validation correctness reward and conduct evaluation on them, and report the evaluation results of the checkpoint with the highest evaluation R@32 score.

**Hardware Information.** Post-training of Qwen2.5-0.5B-Instruct and Qwen2.5-1.5B-Instruct (Yang et al., 2024; Team, 2024) is conducted using 4 NVIDIA H100 GPUs (80GB HBM3), while Qwen2.5-7B-Instruct (Yang et al., 2024; Team, 2024) is trained using 8 H100 GPUs. All evaluations are performed using 2 H100 GPUs across all model sizes.

**Distributed Computing Configurations.** We adopt DeepSpeed ZeRO Stage 2 (Rasley et al., 2020) for post-training Qwen2.5-0.5B-Instruct and Qwen2.5-1.5B-Instruct (Yang et al., 2024; Team, 2024), and ZeRO Stage 3 (Rasley et al., 2020) for Qwen2.5-7B-Instruct (Yang et al., 2024; Team, 2024). For completion generation, we employ vLLM (Kwon et al., 2023) in colocation mode, with GPU memory utilization set to 0.4 and tensor parallelism set to 2.

### C.2. Effectiveness & Efficiency

#### C.2.1. TOKEN EFFICIENCY ON NON-MATH REASONING DATASETS

Although LLM post-training methods are primarily benchmarked on mathematical reasoning datasets, we also report the performance of our `IAPO` method on a commonsense reasoning dataset, namely CommonsenseQA (Talmor et al., 2019). In this dataset, each prompt presents a commonsense question followed by multiple-choice options. For example, a prompt may be: "The sanctions against the school were a punishing blow, and they seemed to what the efforts the school had made to

change? A. ignore, B. enforce, C. authoritarian, D. yell at, E. avoid." The LLM is expected to reason about the question and output the letter corresponding to its chosen answer. We report Pass@K, Length@K, and Ratio@K in Table 2. We observe that `IAPO` achieves the second-highest token-efficiency, as measured by Ratio@16 and Ratio@32. At the same time, `IAPO` delivers substantially higher effectiveness—approximately a 20% improvement in P@16 and P@32—demonstrating that our method achieves promising post-training performance.

*Table 2.* Comparison of `IAPO` with various baselines on CommonsenseQA (Talmor et al., 2019) and Qwen2.5-0.5B-Instruct (Team, 2024). P@$k$, L@$k$, and R@$k$ denote Pass@$k$, Length@$k$, and the ratio Pass@$k$/Length@$k$, respectively.

| Method | P@16 | P@32 | L@16 | L@32 | R@16 | R@32 |
|---|---|---|---|---|---|---|
| DAPO (Yu et al., 2025) | 0.7111±0.0314 | 0.7333±0.0000 | 25.51±0.12 | 25.28±0.17 | $2.79 \times 10^{-2}$ | $2.90 \times 10^{-2}$ |
| GFPO (Shrivastava et al., 2025) | 0.6667±0.0000 | 0.6667±0.0000 | 17.71±0.16 | 18.02±0.37 | $\mathbf{3.76 \times 10^{-2}}$ | $\mathbf{3.70 \times 10^{-2}}$ |
| GTPO (Tan et al., 2025) | 0.7111±0.0629 | 0.7333±0.0544 | 108.91±0.53 | 107.16±2.10 | $6.52 \times 10^{-3}$ | $6.84 \times 10^{-3}$ |
| S-GRPO (Lee & Tong, 2025) | 0.7556±0.0314 | 0.7778±0.0314 | 25.15±0.22 | 25.21±0.14 | $3.00 \times 10^{-2}$ | $3.09 \times 10^{-2}$ |
| + `IAPO` (ours) | 0.8000±0.0000 | 0.8000±0.0000 | 24.01±0.28 | 23.78±0.02 | $\underline{3.33 \times 10^{-2}}$ | $\underline{3.36 \times 10^{-2}}$ |

### C.2.2. EXTENDED EVALUATION ON PASS@K AND LENGTH@K

We evaluate our `IAPO` and the baselines on the three adopted datasets and LLMs with Pass@2, Length@2, Ratio@2, Pass@4, Length@4, Ratio@4, Pass@8, Length@8, Ratio@8 as well. The results of Pass@2, Length@2, and Ratio@2 are shown in Table 3, the results of Pass@4, Length@4, and Ratio@4 are shown in Table 4, the results of Pass@8, Length@8 and Ratio@8 are shown in Table 5. Our evaluation results align with those of the main paper that our proposed `IAPO` achieves a generally optimal effectiveness-efficiency trade-off when tested across LLMs and reasoning datasets.

*Table 3.* Effectiveness and efficiency of `IAPO` and baselines evaluated with P@2, L@2, and R@2.

| | | | DAPO | GFPO | GTPO | S-GRPO | IAPO |
|---|---|---|---|---|---|---|---|
| 0.5B | GSM8K | P@2 | 0.6515±0.0066 | 0.6027±0.0049 | 0.6138±0.0089 | 0.6113±0.0147 | 0.6247±0.0022 |
| | | L@2 | 172.40±2.59 | 217.02±1.56 | 241.73±1.55 | 159.72±0.97 | 151.00±1.78 |
| | | R@2 | $3.78 \times 10^{-3}$ | $2.78 \times 10^{-3}$ | $2.54 \times 10^{-3}$ | $3.83 \times 10^{-3}$ | $4.14 \times 10^{-3}$ |
| | Math-17k | P@2 | 0.0867±0.0094 | 0.0400±0.0163 | 0.0400±0.0163 | 0.0733±0.0249 | 0.1000±0.0163 |
| | | L@2 | 32.52±0.58 | 16.86±0.55 | 15.93±0.19 | 17.64±0.03 | 17.43±0.04 |
| | | R@2 | $2.67 \times 10^{-3}$ | $2.37 \times 10^{-3}$ | $2.51 \times 10^{-3}$ | $4.16 \times 10^{-3}$ | $5.74 \times 10^{-3}$ |
| | MATH-500 | P@2 | 0.2896±0.0126 | 0.2896±0.0126 | 0.2323±0.0359 | 0.2559±0.0126 | 0.2761±0.0172 |
| | | L@2 | 160.70±2.93 | 164.27±5.21 | 475.90±1.82 | 102.58±1.36 | 163.65±4.31 |
| | | R@2 | $1.80 \times 10^{-3}$ | $1.76 \times 10^{-3}$ | $4.88 \times 10^{-4}$ | $2.49 \times 10^{-3}$ | $1.69 \times 10^{-3}$ |
| 1.5B | GSM8K | P@2 | 0.8221±0.0053 | 0.8279±0.0032 | 0.8198±0.0034 | 0.8426±0.0085 | 0.8261±0.0014 |
| | | L@2 | 168.93±1.98 | 202.02±0.76 | 261.40±0.31 | 180.13±0.83 | 160.20±2.84 |
| | | R@2 | $4.87 \times 10^{-3}$ | $4.10 \times 10^{-3}$ | $3.14 \times 10^{-3}$ | $4.68 \times 10^{-3}$ | $5.16 \times 10^{-3}$ |
| | Math-17k | P@2 | 0.0400±0.0000 | 0.0600±0.0327 | 0.0800±0.0163 | 0.0933±0.0094 | 0.0733±0.0094 |
| | | L@2 | 38.61±1.33 | 64.59±4.98 | 18.94±0.13 | 62.01±2.61 | 41.08±0.76 |
| | | R@2 | $1.04 \times 10^{-3}$ | $9.29 \times 10^{-4}$ | $4.22 \times 10^{-3}$ | $1.50 \times 10^{-3}$ | $1.78 \times 10^{-3}$ |
| | MATH-500 | P@2 | 0.5522±0.0190 | 0.4949±0.0218 | 0.3502±0.0095 | 0.5421±0.0126 | 0.5488±0.0095 |
| | | L@2 | 332.77±3.31 | 374.32±8.07 | 495.30±3.07 | 276.59±3.55 | 264.53±5.22 |
| | | R@2 | $1.66 \times 10^{-3}$ | $1.32 \times 10^{-3}$ | $7.07 \times 10^{-4}$ | $1.96 \times 10^{-3}$ | $2.07 \times 10^{-3}$ |
| 7B | GSM8K | P@2 | 0.9431±0.0006 | 0.9449±0.0028 | 0.9338±0.0034 | 0.9419±0.0022 | 0.9179±0.0044 |
| | | L@2 | 156.33±3.20 | 156.16±1.96 | 193.27±0.60 | 144.39±0.14 | 84.14±0.61 |
| | | R@2 | $6.03 \times 10^{-3}$ | $6.05 \times 10^{-3}$ | $4.83 \times 10^{-3}$ | $6.52 \times 10^{-3}$ | $1.09 \times 10^{-2}$ |
| | Math-17k | P@2 | 0.2400±0.0283 | 0.2667±0.0094 | 0.1200±0.0163 | 0.2600±0.0000 | 0.2600±0.0163 |
| | | L@2 | 444.34±2.66 | 420.55±3.39 | 518.77±1.87 | 400.40±9.52 | 425.78±2.39 |
| | | R@2 | $5.40 \times 10^{-4}$ | $6.34 \times 10^{-4}$ | $2.31 \times 10^{-4}$ | $6.49 \times 10^{-4}$ | $6.11 \times 10^{-4}$ |
| | MATH-500 | P@2 | 0.6431±0.0126 | 0.6094±0.0095 | 0.5185±0.0252 | 0.6700±0.0238 | 0.6801±0.0048 |
| | | L@2 | 327.31±3.79 | 341.66±2.71 | 452.15±1.19 | 334.14±2.25 | 299.38±1.99 |
| | | R@2 | $1.96 \times 10^{-3}$ | $1.78 \times 10^{-3}$ | $1.15 \times 10^{-3}$ | $2.00 \times 10^{-3}$ | $2.27 \times 10^{-3}$ |

Table 4. Effectiveness and efficiency of IAPO and baselines evaluated with P@4, L@4, and R@4.

| | | | DAPO | GFPO | GTPO | S-GRPO | IAPO |
|---|---|---|---|---|---|---|---|
| 0.5B | GSM8K | P@4 | 0.7374±0.0056 | 0.7028±0.0125 | 0.7159±0.0095 | 0.7111±0.0059 | 0.7165±0.0049 |
| | | L@4 | 172.31±1.65 | 215.23±1.60 | 242.17±0.15 | 159.34±0.88 | 150.32±0.86 |
| | | R@4 | $4.28 \times 10^{-3}$ | $3.27 \times 10^{-3}$ | $2.96 \times 10^{-3}$ | $4.46 \times 10^{-3}$ | $4.77 \times 10^{-3}$ |
| | Math-17k | P@4 | 0.1467±0.0189 | 0.0533±0.0094 | 0.0667±0.0189 | 0.1133±0.0189 | 0.1200±0.0163 |
| | | L@4 | 33.10±0.45 | 16.60±0.28 | 15.86±0.11 | 17.60±0.13 | 17.39±0.07 |
| | | R@4 | $4.43 \times 10^{-3}$ | $3.21 \times 10^{-3}$ | $4.21 \times 10^{-3}$ | $6.44 \times 10^{-3}$ | $6.90 \times 10^{-3}$ |
| | MATH-500 | P@4 | 0.3333±0.0082 | 0.3535±0.0082 | 0.2997±0.0390 | 0.3232±0.0165 | 0.3704±0.0172 |
| | | L@4 | 161.22±1.16 | 162.88±2.78 | 474.21±0.53 | 101.98±1.77 | 162.34±2.18 |
| | | R@4 | $2.07 \times 10^{-3}$ | $2.17 \times 10^{-3}$ | $6.32 \times 10^{-4}$ | $3.17 \times 10^{-3}$ | $2.28 \times 10^{-3}$ |
| 1.5B | GSM8K | P@4 | 0.8835±0.0086 | 0.8855±0.0022 | 0.8812±0.0029 | 0.8964±0.0004 | 0.8888±0.0036 |
| | | L@4 | 169.43±1.09 | 202.70±1.28 | 262.67±0.88 | 180.57±0.71 | 161.83±1.04 |
| | | R@4 | $5.21 \times 10^{-3}$ | $4.37 \times 10^{-3}$ | $3.36 \times 10^{-3}$ | $4.96 \times 10^{-3}$ | $5.49 \times 10^{-3}$ |
| | Math-17k | P@4 | 0.1200±0.0163 | 0.0933±0.0189 | 0.1133±0.0094 | 0.1200±0.0283 | 0.1067±0.0340 |
| | | L@4 | 39.72±1.06 | 65.52±0.54 | 18.91±0.08 | 60.56±1.11 | 41.84±0.47 |
| | | R@4 | $3.02 \times 10^{-3}$ | $1.42 \times 10^{-3}$ | $5.99 \times 10^{-3}$ | $1.98 \times 10^{-3}$ | $2.55 \times 10^{-3}$ |
| | MATH-500 | P@4 | 0.6027±0.0252 | 0.5657±0.0218 | 0.4343±0.0218 | 0.6128±0.0126 | 0.5960±0.0218 |
| | | L@4 | 329.57±3.74 | 372.52±7.55 | 495.57±0.41 | 278.09±3.86 | 264.57±2.82 |
| | | R@4 | $1.83 \times 10^{-3}$ | $1.52 \times 10^{-3}$ | $8.76 \times 10^{-4}$ | $2.20 \times 10^{-3}$ | $2.25 \times 10^{-3}$ |
| 7B | GSM8K | P@4 | 0.9616±0.0019 | 0.9636±0.0022 | 0.9530±0.0011 | 0.9618±0.0014 | 0.9441±0.0032 |
| | | L@4 | 158.14±0.93 | 158.05±2.04 | 193.05±0.13 | 145.81±0.88 | 84.04±0.47 |
| | | R@4 | $6.08 \times 10^{-3}$ | $6.10 \times 10^{-3}$ | $4.94 \times 10^{-3}$ | $6.60 \times 10^{-3}$ | $1.12 \times 10^{-2}$ |
| | Math-17k | P@4 | 0.2667±0.0249 | 0.3133±0.0094 | 0.1333±0.0249 | 0.3000±0.0163 | 0.2867±0.0094 |
| | | L@4 | 446.21±3.16 | 424.78±0.74 | 518.73±0.86 | 401.46±6.58 | 427.52±5.75 |
| | | R@4 | $5.98 \times 10^{-4}$ | $7.38 \times 10^{-4}$ | $2.57 \times 10^{-4}$ | $7.47 \times 10^{-4}$ | $6.71 \times 10^{-4}$ |
| | MATH-500 | P@4 | 0.7037±0.0095 | 0.6566±0.0165 | 0.5623±0.0172 | 0.7172±0.0165 | 0.7138±0.0208 |
| | | L@4 | 330.62±2.13 | 342.70±2.41 | 453.33±2.31 | 334.08±1.80 | 300.63±2.18 |
| | | R@4 | $2.13 \times 10^{-3}$ | $1.92 \times 10^{-3}$ | $1.24 \times 10^{-3}$ | $2.15 \times 10^{-3}$ | $2.37 \times 10^{-3}$ |

*Table 5.* Effectiveness and efficiency of `IAPO` and baselines evaluated with P@8, L@8, and R@8.

| | | | DAPO | GFPO | GTPO | S-GRPO | IAPO |
|---|---|---|---|---|---|---|---|
| **0.5B** | GSM8K | P@8 | 0.8067±0.0019 | 0.7923±0.0087 | 0.7930±0.0059 | 0.7902±0.0053 | 0.7928±0.0054 |
| | | L@8 | 172.94±0.21 | 216.56±1.86 | 242.81±0.22 | 159.81±0.35 | 150.55±0.48 |
| | | R@8 | $4.66 \times 10^{-3}$ | $3.66 \times 10^{-3}$ | $3.27 \times 10^{-3}$ | $4.94 \times 10^{-3}$ | $5.27 \times 10^{-3}$ |
| | Math-17k | P@8 | 0.1867±0.0094 | 0.1600±0.0283 | 0.1133±0.0340 | 0.1400±0.0327 | 0.1800±0.0000 |
| | | L@8 | 32.95±0.22 | 16.56±0.01 | 15.82±0.04 | 17.49±0.12 | 17.36±0.04 |
| | | R@8 | $5.67 \times 10^{-3}$ | $9.66 \times 10^{-3}$ | $7.16 \times 10^{-3}$ | $8.00 \times 10^{-3}$ | $1.04 \times 10^{-2}$ |
| | MATH-500 | P@8 | 0.3973±0.0172 | 0.4478±0.0238 | 0.4108±0.0372 | 0.3872±0.0126 | 0.4276±0.0390 |
| | | L@8 | 162.97±1.64 | 162.86±0.74 | 473.62±1.17 | 103.93±0.66 | 165.15±0.21 |
| | | R@8 | $2.44 \times 10^{-3}$ | $2.75 \times 10^{-3}$ | $8.67 \times 10^{-4}$ | $3.73 \times 10^{-3}$ | $2.59 \times 10^{-3}$ |
| **1.5B** | GSM8K | P@8 | 0.9222±0.0041 | 0.9232±0.0009 | 0.9242±0.0033 | 0.9348±0.0016 | 0.9282±0.0035 |
| | | L@8 | 169.26±0.59 | 203.80±0.61 | 262.51±0.38 | 180.73±0.16 | 162.77±0.78 |
| | | R@8 | $5.45 \times 10^{-3}$ | $4.53 \times 10^{-3}$ | $3.52 \times 10^{-3}$ | $5.17 \times 10^{-3}$ | $5.70 \times 10^{-3}$ |
| | Math-17k | P@8 | 0.1600±0.0283 | 0.1533±0.0094 | 0.1600±0.0163 | 0.1733±0.0340 | 0.1600±0.0432 |
| | | L@8 | 38.95±0.67 | 65.56±1.20 | 18.88±0.03 | 60.20±0.58 | 41.76±0.82 |
| | | R@8 | $4.11 \times 10^{-3}$ | $2.34 \times 10^{-3}$ | $8.47 \times 10^{-3}$ | $2.88 \times 10^{-3}$ | $3.83 \times 10^{-3}$ |
| | MATH-500 | P@8 | 0.6566±0.0000 | 0.6162±0.0218 | 0.4714±0.0172 | 0.6835±0.0126 | 0.6599±0.0126 |
| | | L@8 | 327.52±2.35 | 375.96±6.42 | 495.66±0.84 | 277.13±2.27 | 265.36±3.18 |
| | | R@8 | $2.00 \times 10^{-3}$ | $1.64 \times 10^{-3}$ | $9.51 \times 10^{-4}$ | $2.47 \times 10^{-3}$ | $2.49 \times 10^{-3}$ |
| **7B** | GSM8K | P@8 | 0.9704±0.0012 | 0.9742±0.0012 | 0.9669±0.0009 | 0.9725±0.0025 | 0.9623±0.0014 |
| | | L@8 | 159.53±0.55 | 160.06±1.45 | 193.23±0.02 | 147.33±0.69 | 83.95±0.25 |
| | | R@8 | $6.08 \times 10^{-3}$ | $6.09 \times 10^{-3}$ | $5.00 \times 10^{-3}$ | $6.60 \times 10^{-3}$ | $1.15 \times 10^{-2}$ |
| | Math-17k | P@8 | 0.3067±0.0094 | 0.3467±0.0249 | 0.1533±0.0249 | 0.3533±0.0340 | 0.3333±0.0189 |
| | | L@8 | 445.73±1.63 | 423.35±0.95 | 518.86±0.35 | 398.63±1.65 | 429.34±0.35 |
| | | R@8 | $6.88 \times 10^{-4}$ | $8.19 \times 10^{-4}$ | $2.95 \times 10^{-4}$ | $8.86 \times 10^{-4}$ | $7.76 \times 10^{-4}$ |
| | MATH-500 | P@8 | 0.7441±0.0126 | 0.7205±0.0126 | 0.5892±0.0048 | 0.7677±0.0082 | 0.7542±0.0126 |
| | | L@8 | 331.99±1.89 | 344.19±0.51 | 453.86±1.12 | 335.72±1.30 | 301.11±1.70 |
| | | R@8 | $2.24 \times 10^{-3}$ | $2.09 \times 10^{-3}$ | $1.30 \times 10^{-3}$ | $2.29 \times 10^{-3}$ | $2.50 \times 10^{-3}$ |

### C.2.3. WALL-CLOCK INFERENCE TIME

We further report the average wall-clock inference time required to generate 32 completions per query on Qwen2.5-7B-Instruct (Team, 2024) across all three datasets (Fig. 8). Notably, the model post-trained with IAPO achieves a substantially lower inference time—up to a 17.7% reduction—compared to the original model.

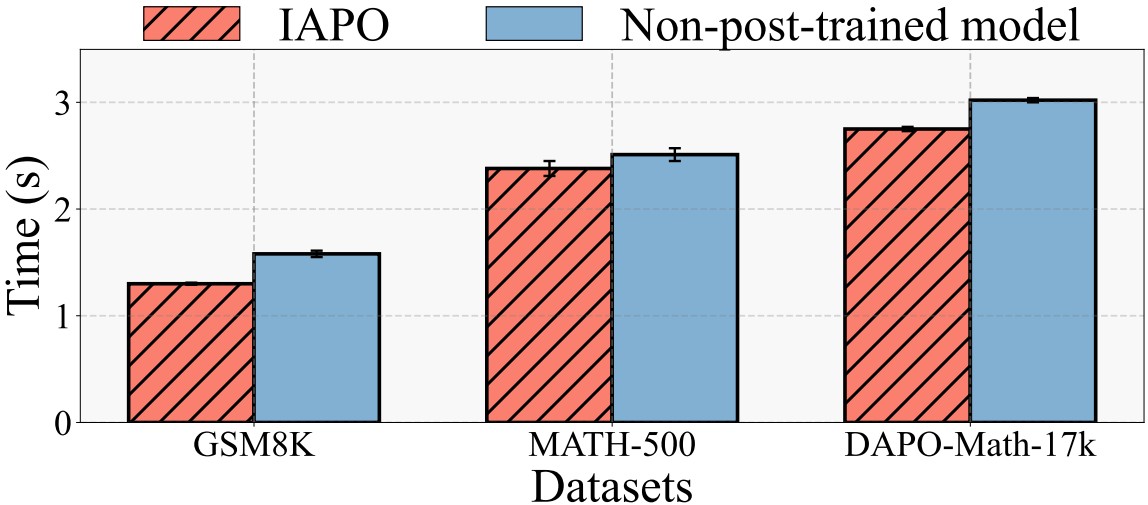

*Figure 8.* Wall-clock LLM inference time before and after IAPO training.

## C.3. Training Dynamics

Here we present the training dynamics of (i) the mean correctness reward of model completions (with a reward of 1 for correct completions and -1 for incorrect ones), (ii) the mean completion length, and (iii) the ratio between correctness reward and completion length. These dynamics indicate that our method consistently achieves optimal token efficiency throughout the post-training process compared with the state-of-the-art baselines.

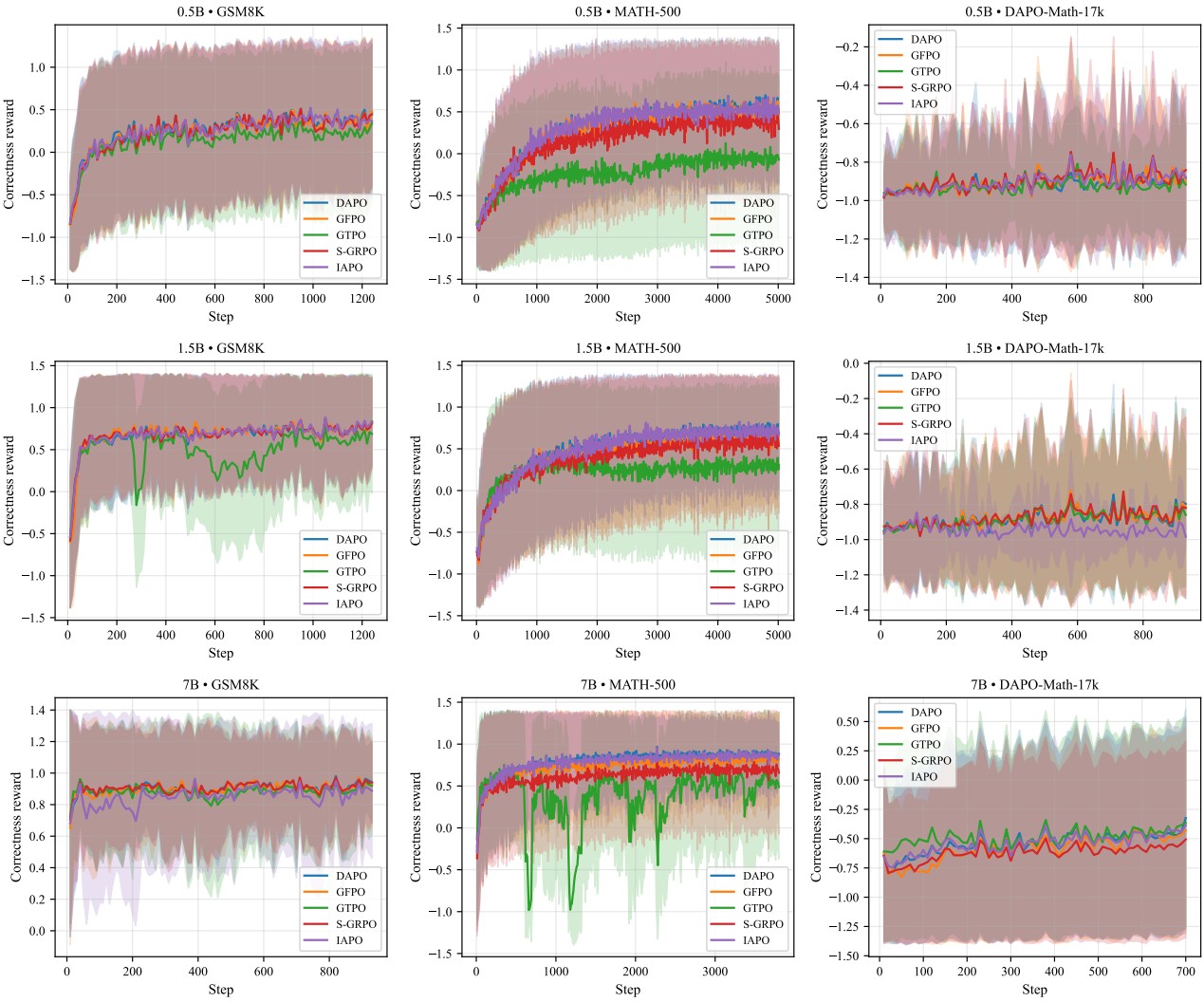

*Figure 9.* The dynamics of correctness reward during the LLM post-training process.

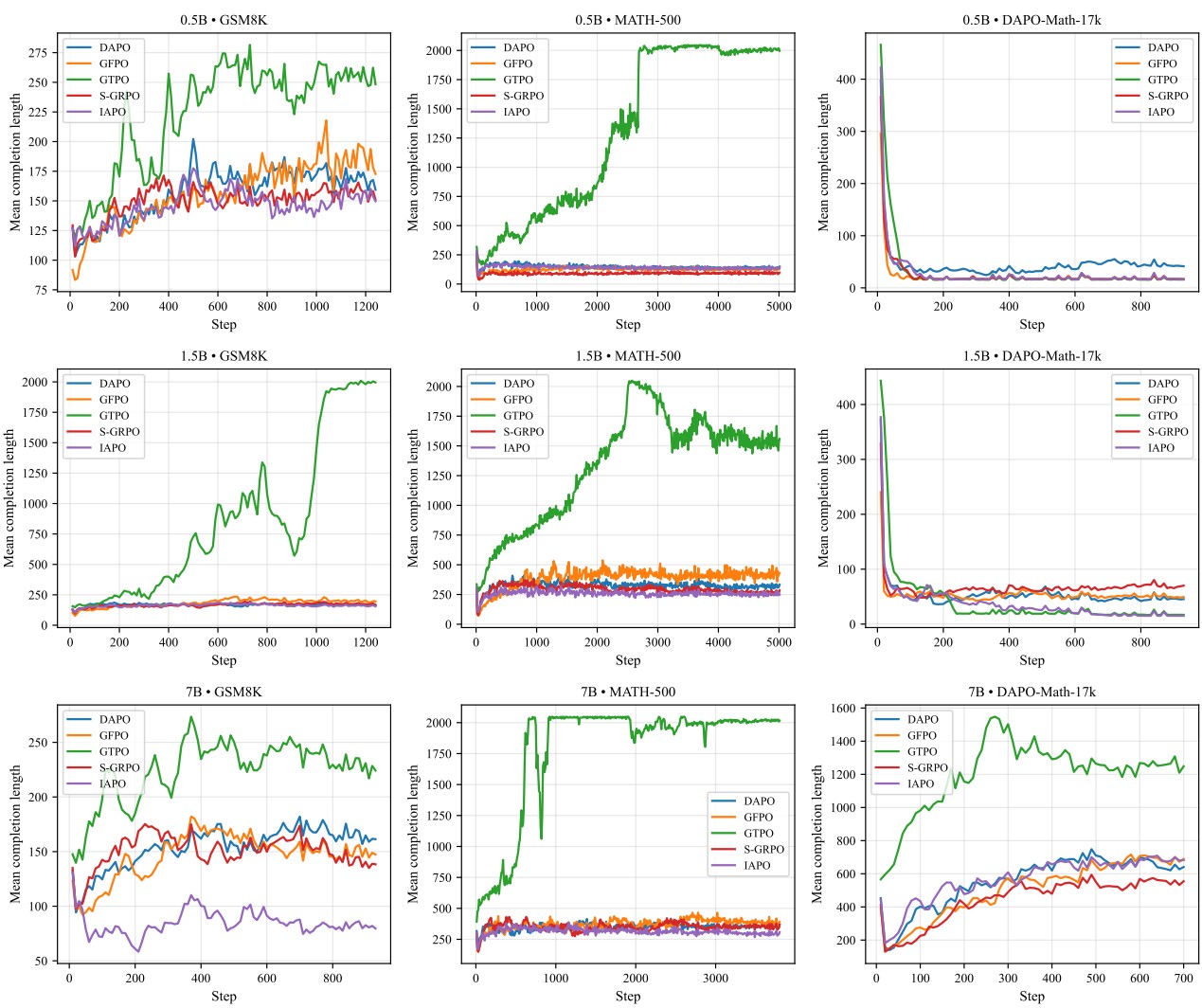

*Figure 10.* The dynamics of completion lengths during the LLM post-training process.

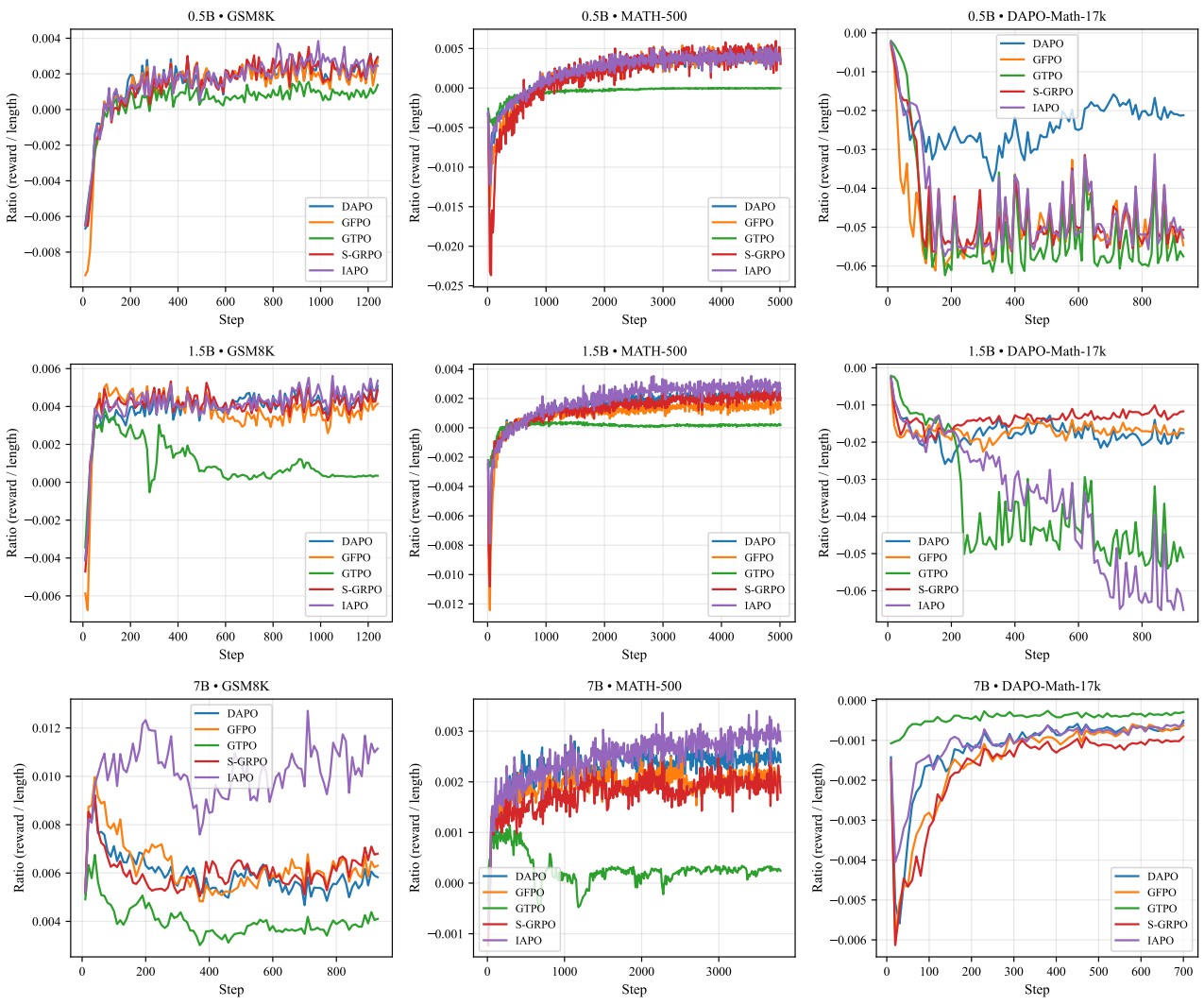

*Figure 11.* The dynamics of the ratio between the mean correctness reward and the mean completion lengths during the LLM post-training process.

### C.4. Case Study

We examine two research questions in this subsection: (1) **Generalizability of Conciseness**: We present additional case studies demonstrating that our IAPO achieves optimal token-efficiency compared with state-of-the-art baselines (see Table 6). The samples show that our method exhibits minimal self-commentary, virtually no failed intermediate derivations, and no redundant verifications, indicating that IAPO achieves token-efficiency improvements by addressing all three factors contributing to verbosity in current RL post-trained LLMs. (2) **Informative Token Identification via Conditional MI**: We examine whether the conditional MI estimated by our early-exit estimator effectively captures informative tokens for producing correct answers. Specifically, we plot (in Fig. 12) token-wise heatmaps for completions generated by Qwen2.5-0.5B-Instruct (Team, 2024) on the GSM8K dataset (Cobbe et al., 2021), where color intensity represents the conditional MI value of each token. The conditional MI successfully identifies informative tokens such as key numbers in reasoning and logical conjunctions (e.g., "since", "proceed"), while assigning low values to redundant tokens such as restatements of givens and meta-commentary.

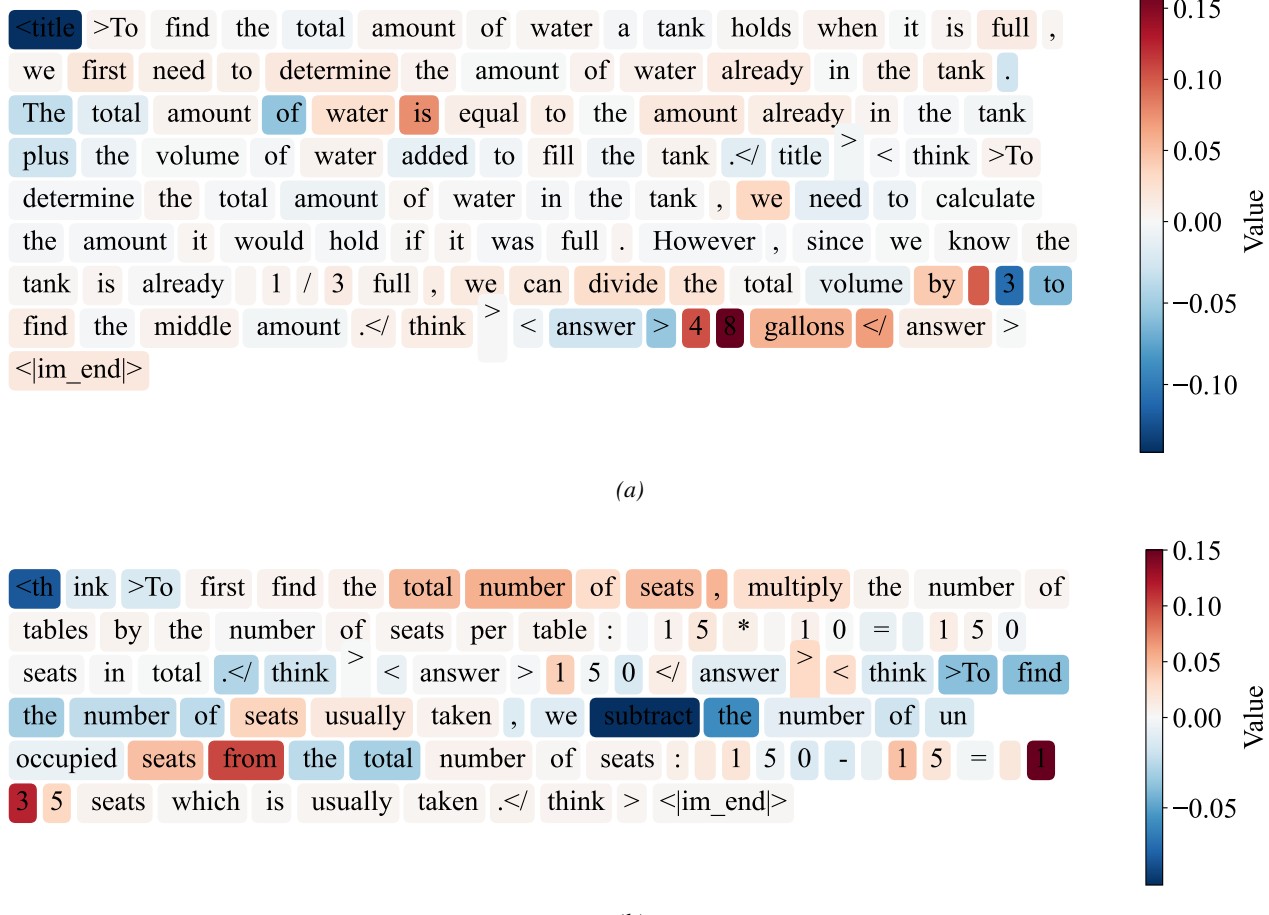

*(a)*

*(b)*

*Figure 12.* Case study on the effectiveness of the conditional MI in identifying informative tokens. Each subplot is a reasoning completion generated by Qwen2.5-0.5B-Instruct on a question from the GSM8K dataset. (Part 1 of 13)

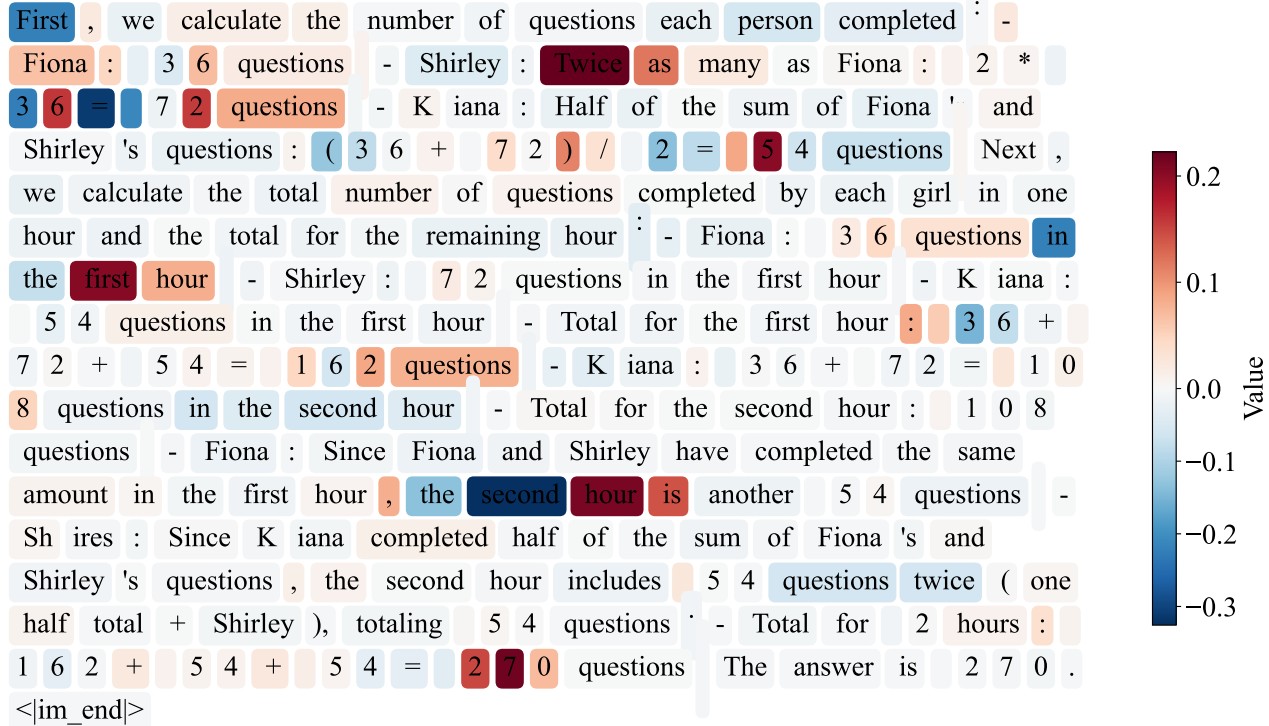

*(c)*

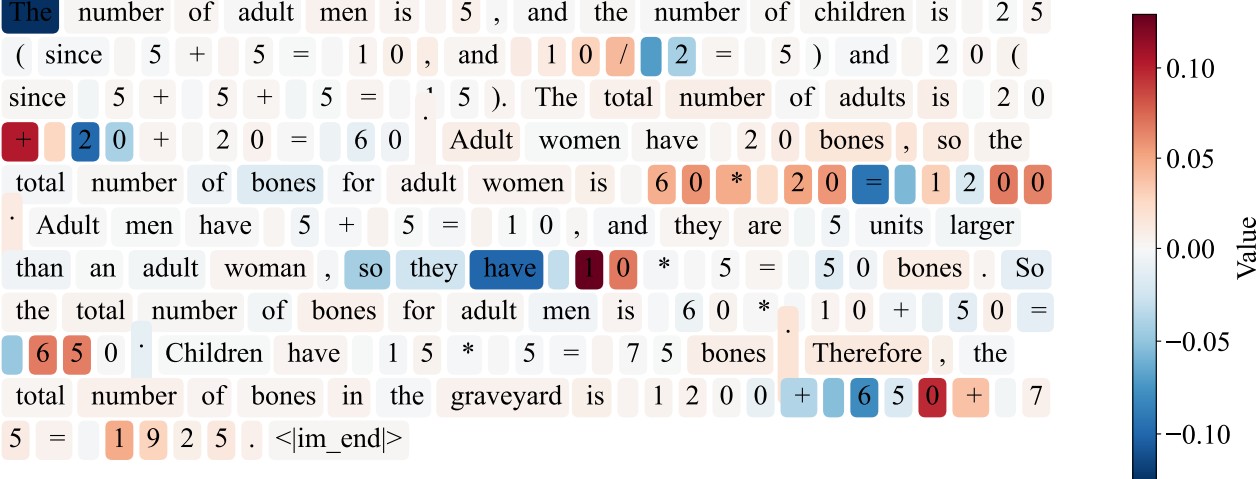

*(d)*

*Figure 12.* Case study on the effectiveness of the conditional MI in identifying informative tokens. Each subplot is a reasoning completion generated by Qwen2.5-0.5B-Instruct on a question from the GSM8K dataset. (Part 2 of 13)

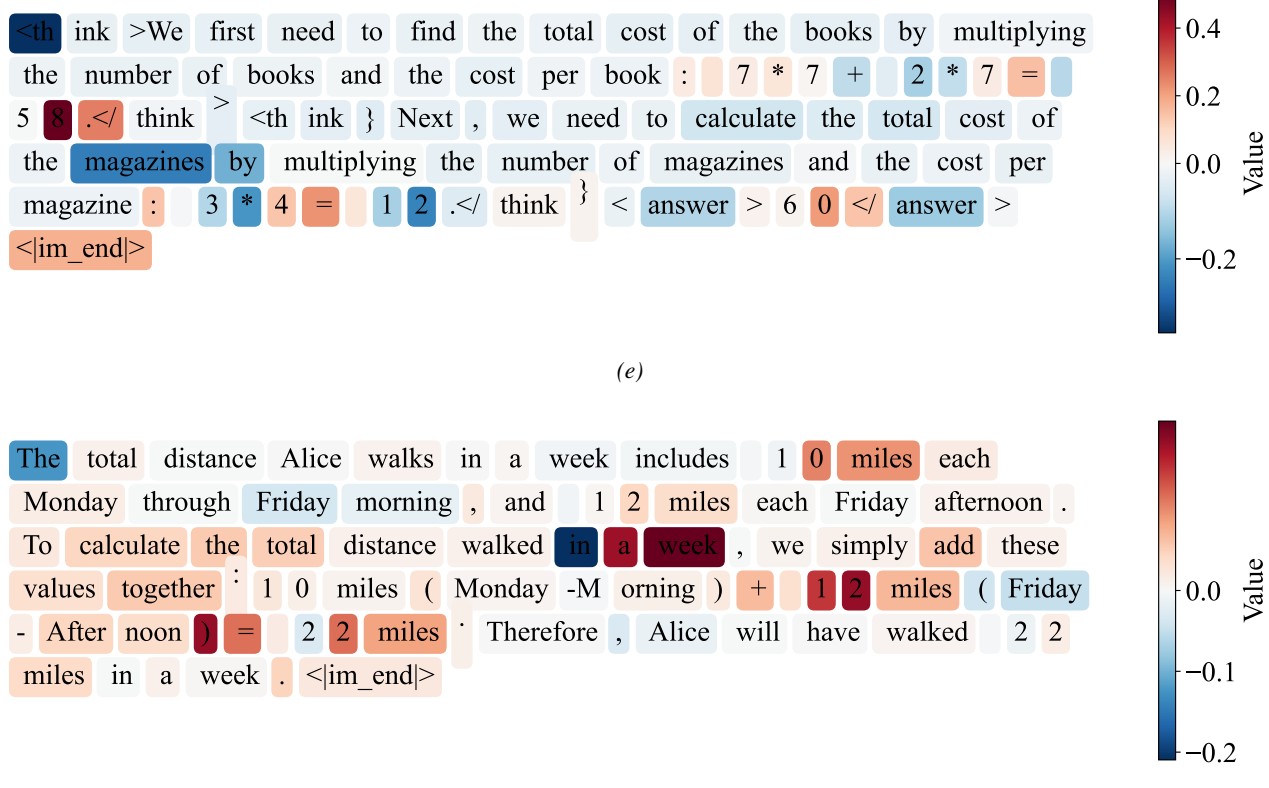

*(e)*

*(f)*

*Figure 12.* Case study on the effectiveness of the conditional MI in identifying informative tokens. Each subplot is a reasoning completion generated by Qwen2.5-0.5B-Instruct on a question from the GSM8K dataset. (Part 3 of 13)

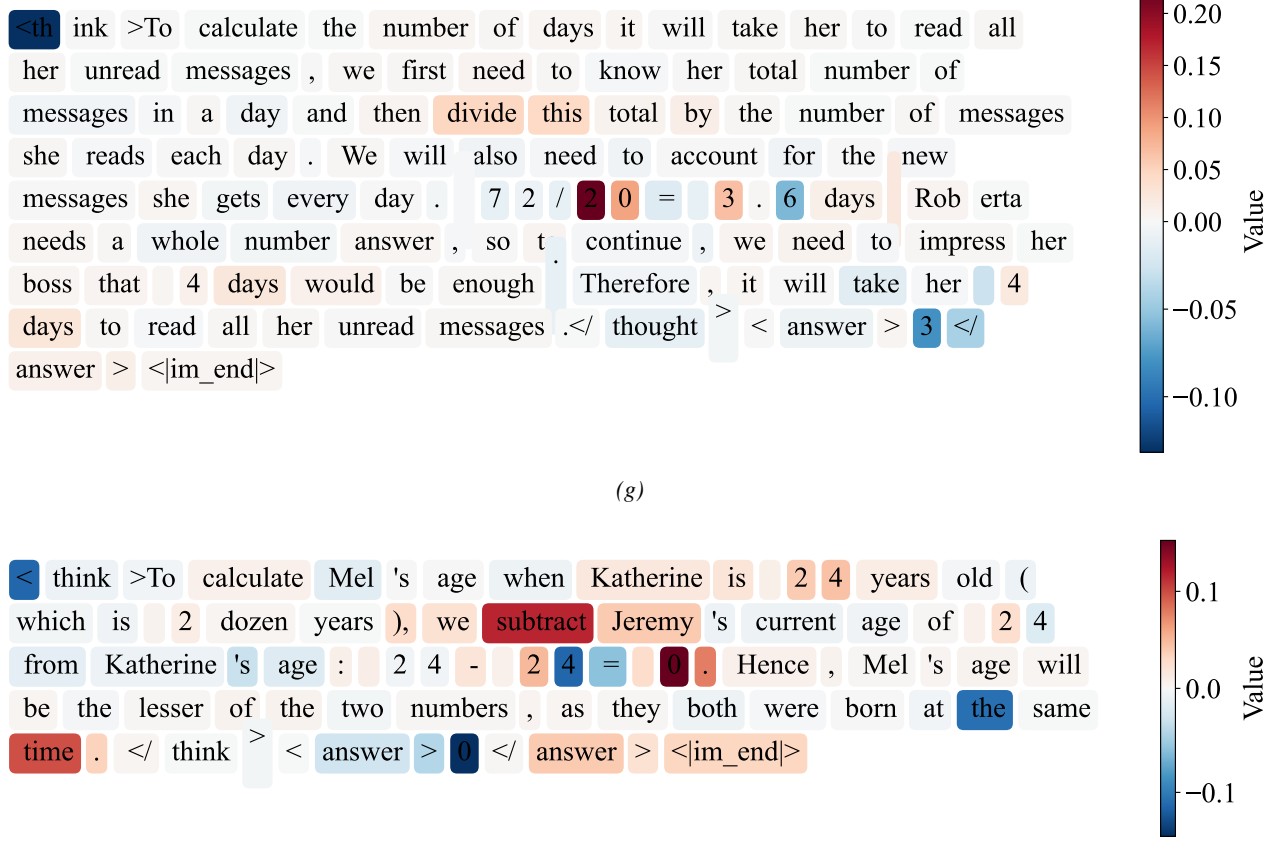

*(g)*

*(h)*

*Figure 12.* Case study on the effectiveness of the conditional MI in identifying informative tokens. Each subplot is a reasoning completion generated by Qwen2.5-0.5B-Instruct on a question from the GSM8K dataset. (Part 4 of 13)

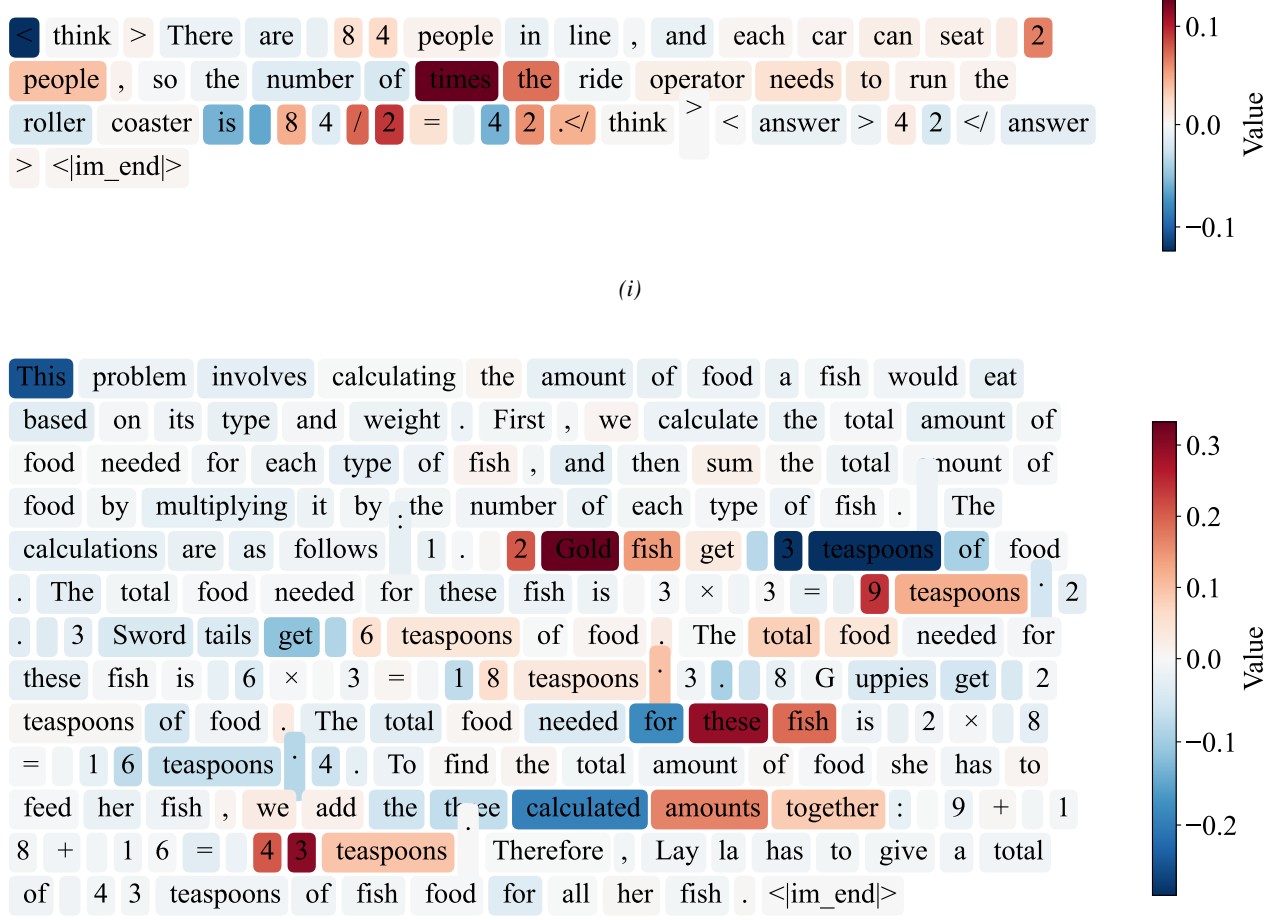

*(i)*

*(j)*

*Figure 12.* Case study on the effectiveness of the conditional MI in identifying informative tokens. Each subplot is a reasoning completion generated by Qwen2.5-0.5B-Instruct on a question from the GSM8K dataset. (Part 5 of 13)

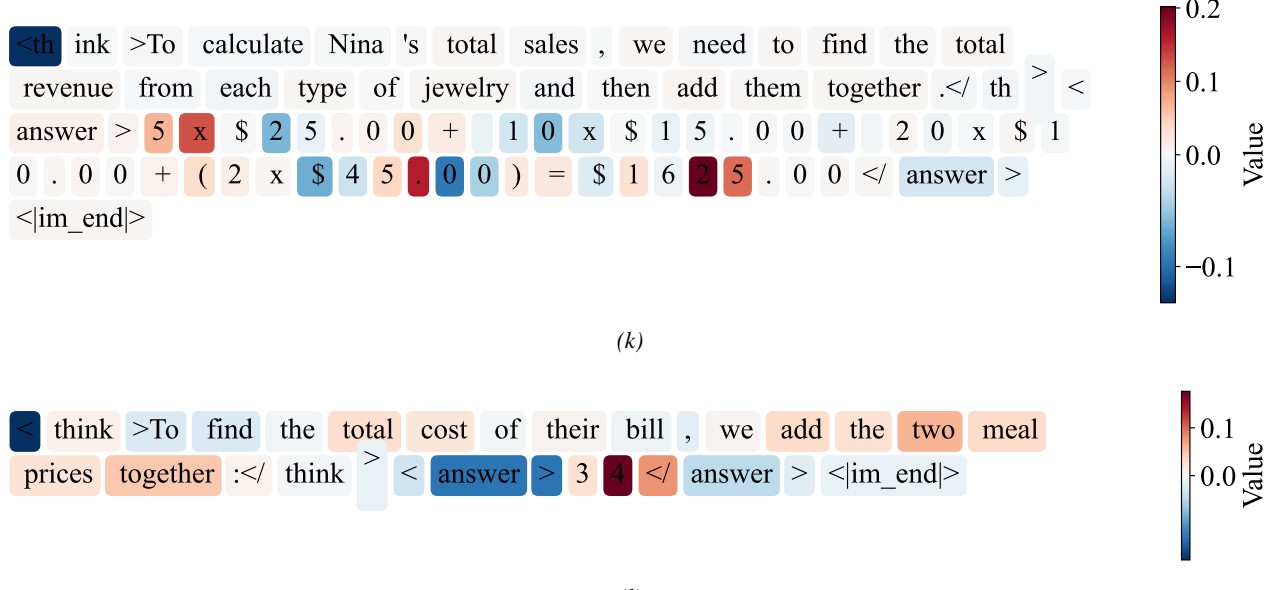

*(k)*

*(l)*

*Figure 12.* Case study on the effectiveness of the conditional MI in identifying informative tokens. Each subplot is a reasoning completion generated by Qwen2.5-0.5B-Instruct on a question from the GSM8K dataset. (Part 6 of 13)

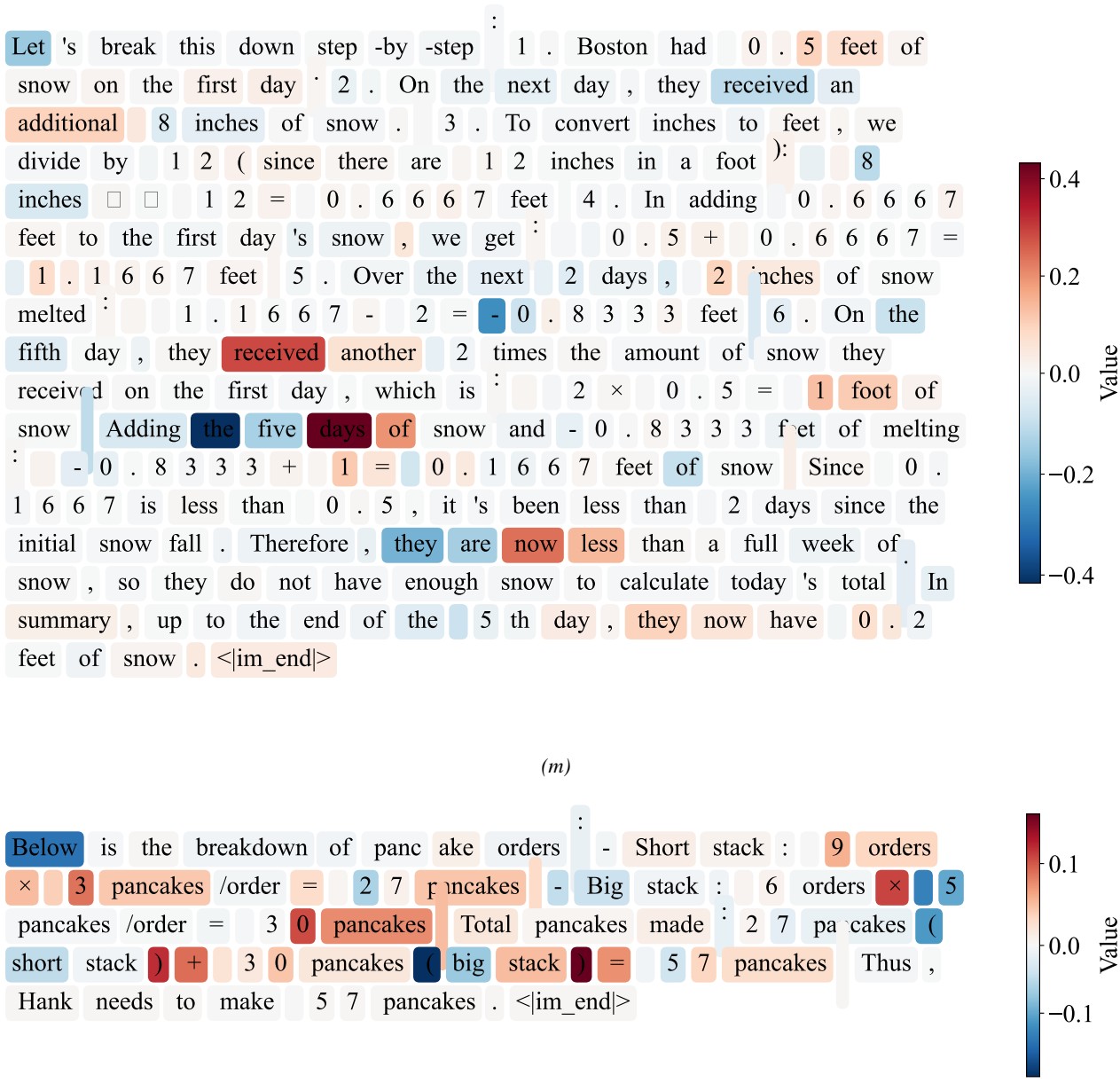

*(m)*

*(n)*

*Figure 12.* Case study on the effectiveness of the conditional MI in identifying informative tokens. Each subplot is a reasoning completion generated by Qwen2.5-0.5B-Instruct on a question from the GSM8K dataset. (Part 7 of 13)

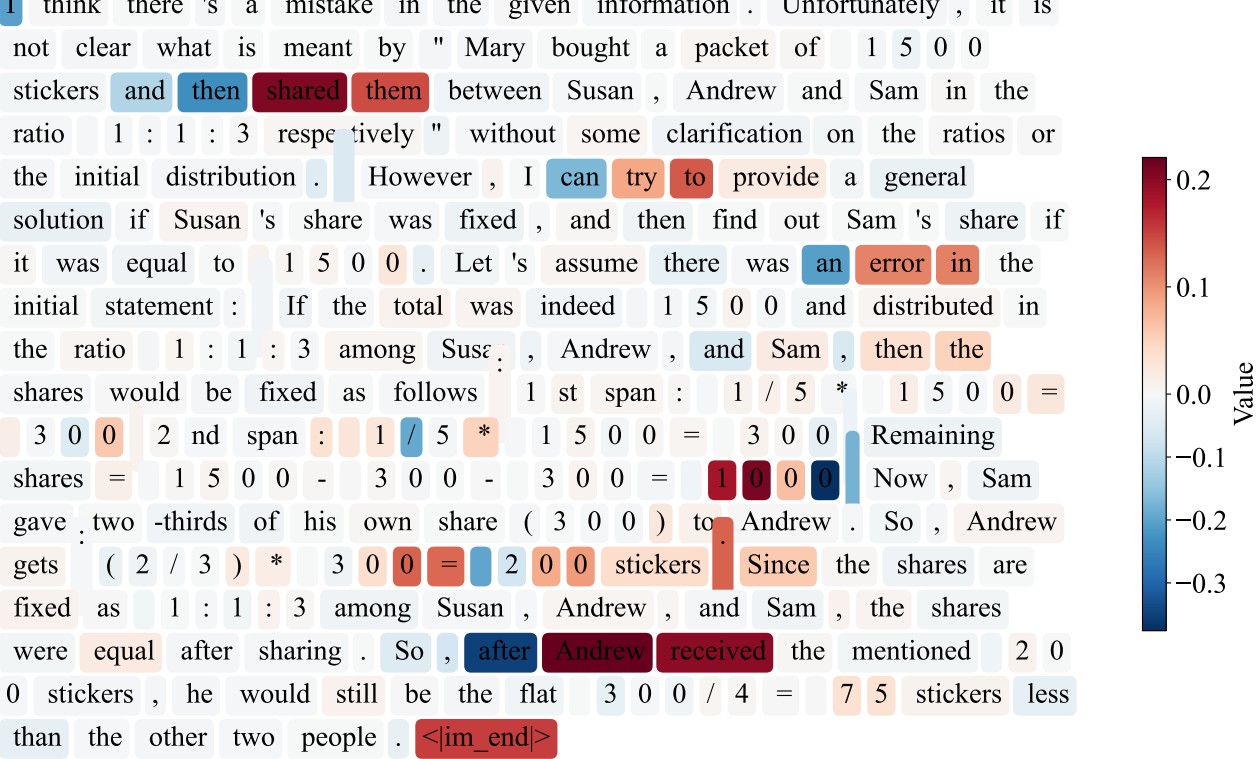

*(o)*

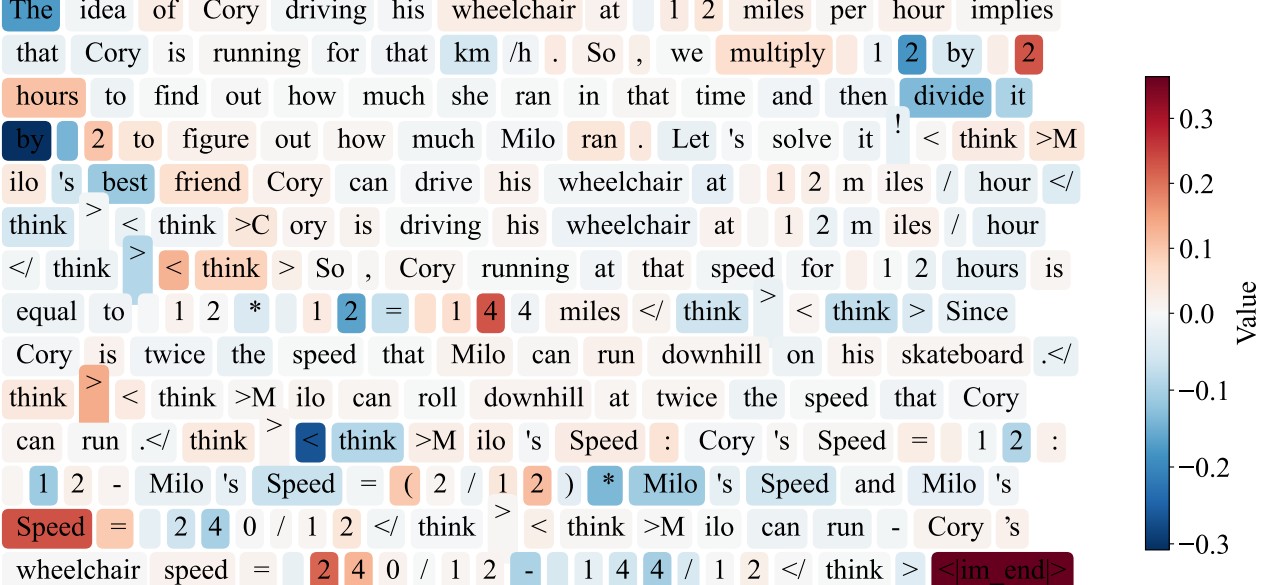

*(p)*

*Figure 12.* Case study on the effectiveness of the conditional MI in identifying informative tokens. Each subplot is a reasoning completion generated by Qwen2.5-0.5B-Instruct on a question from the GSM8K dataset. (Part 8 of 13)

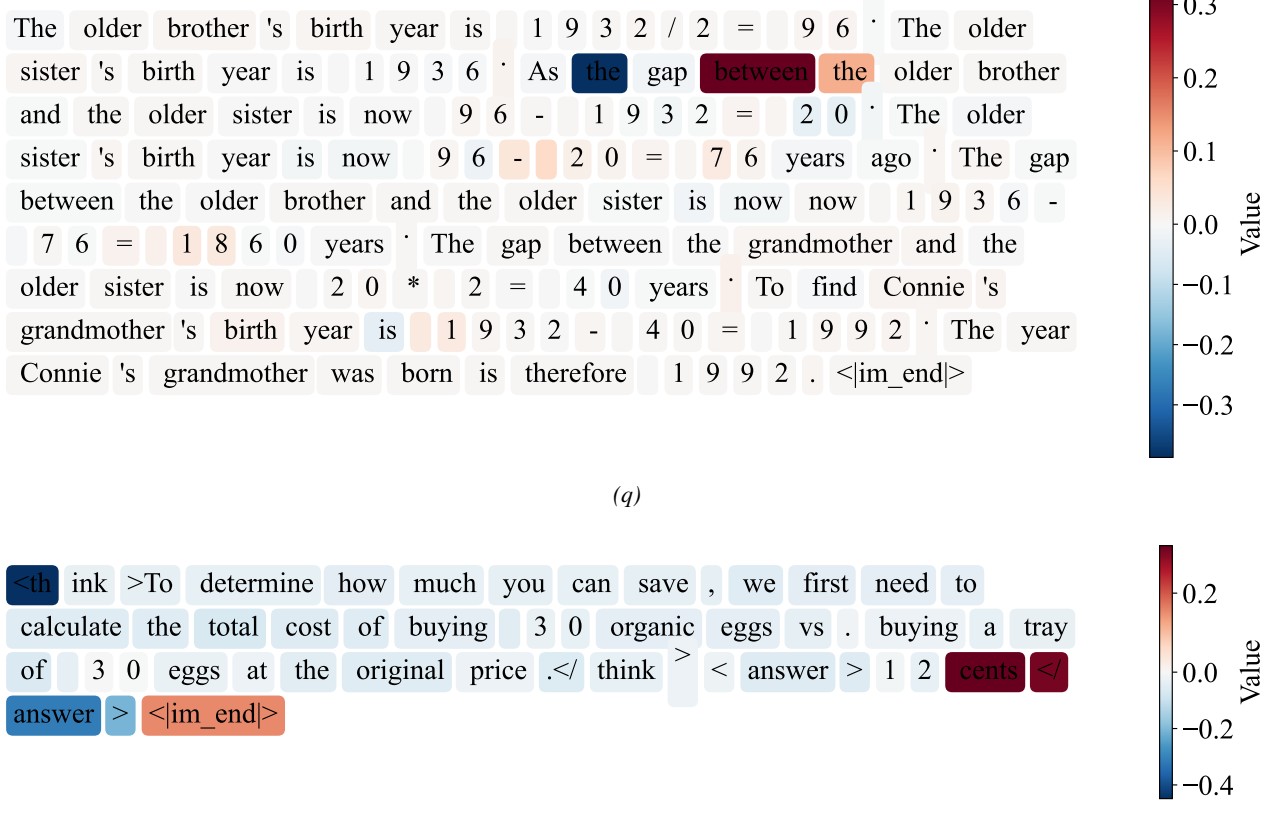

*(q)*

*(r)*

*Figure 12.* Case study on the effectiveness of the conditional MI in identifying informative tokens. Each subplot is a reasoning completion generated by Qwen2.5-0.5B-Instruct on a question from the GSM8K dataset. (Part 9 of 13)

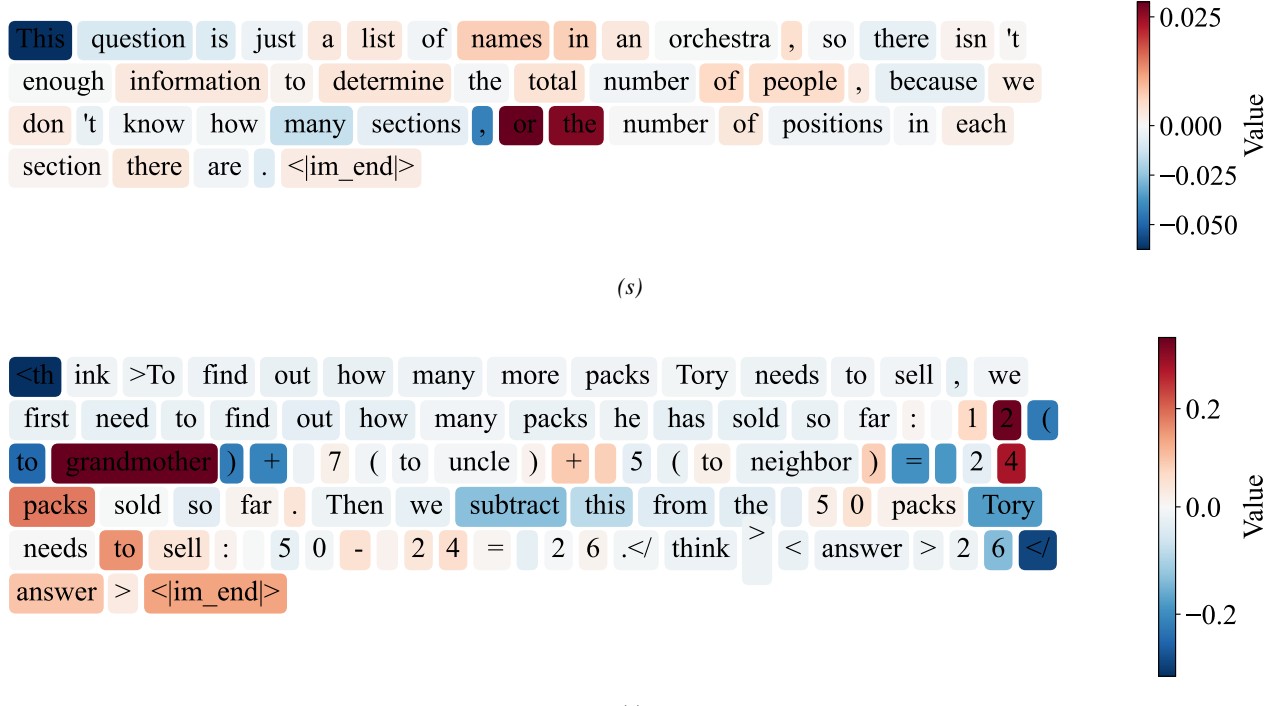

*Figure 12.* Case study on the effectiveness of the conditional MI in identifying informative tokens. Each subplot is a reasoning completion generated by Qwen2.5-0.5B-Instruct on a question from the GSM8K dataset. (Part 10 of 13)

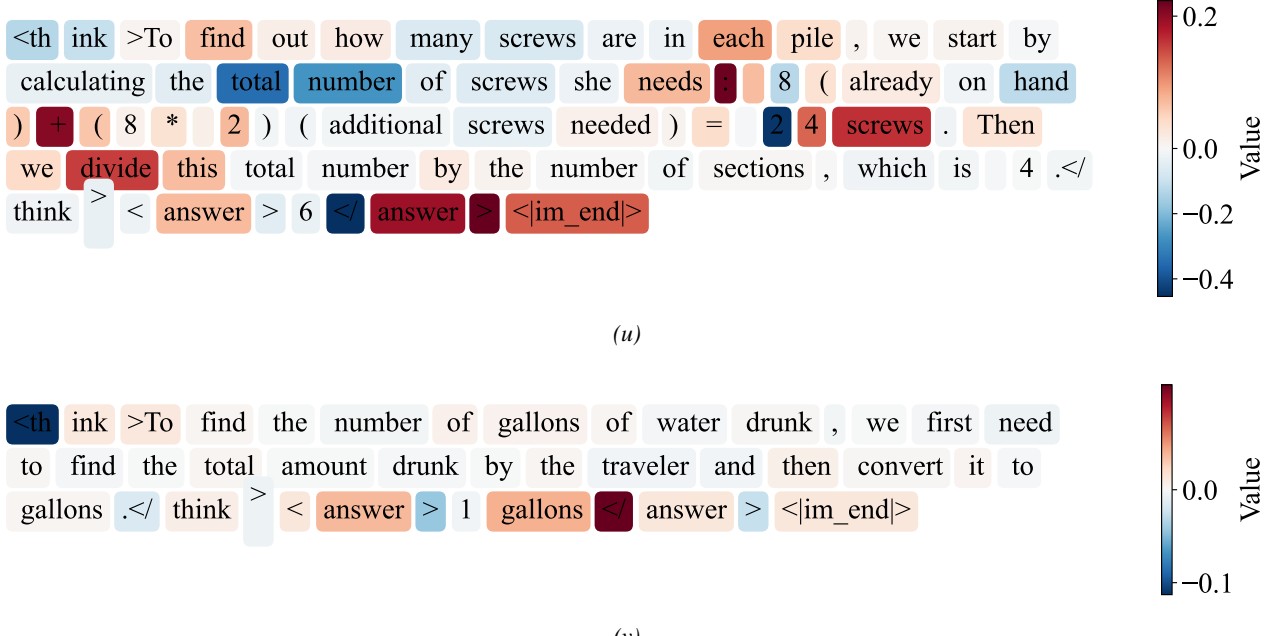

*Figure 12.* Case study on the effectiveness of the conditional MI in identifying informative tokens. Each subplot is a reasoning completion generated by Qwen2.5-0.5B-Instruct on a question from the GSM8K dataset. (Part 11 of 13)

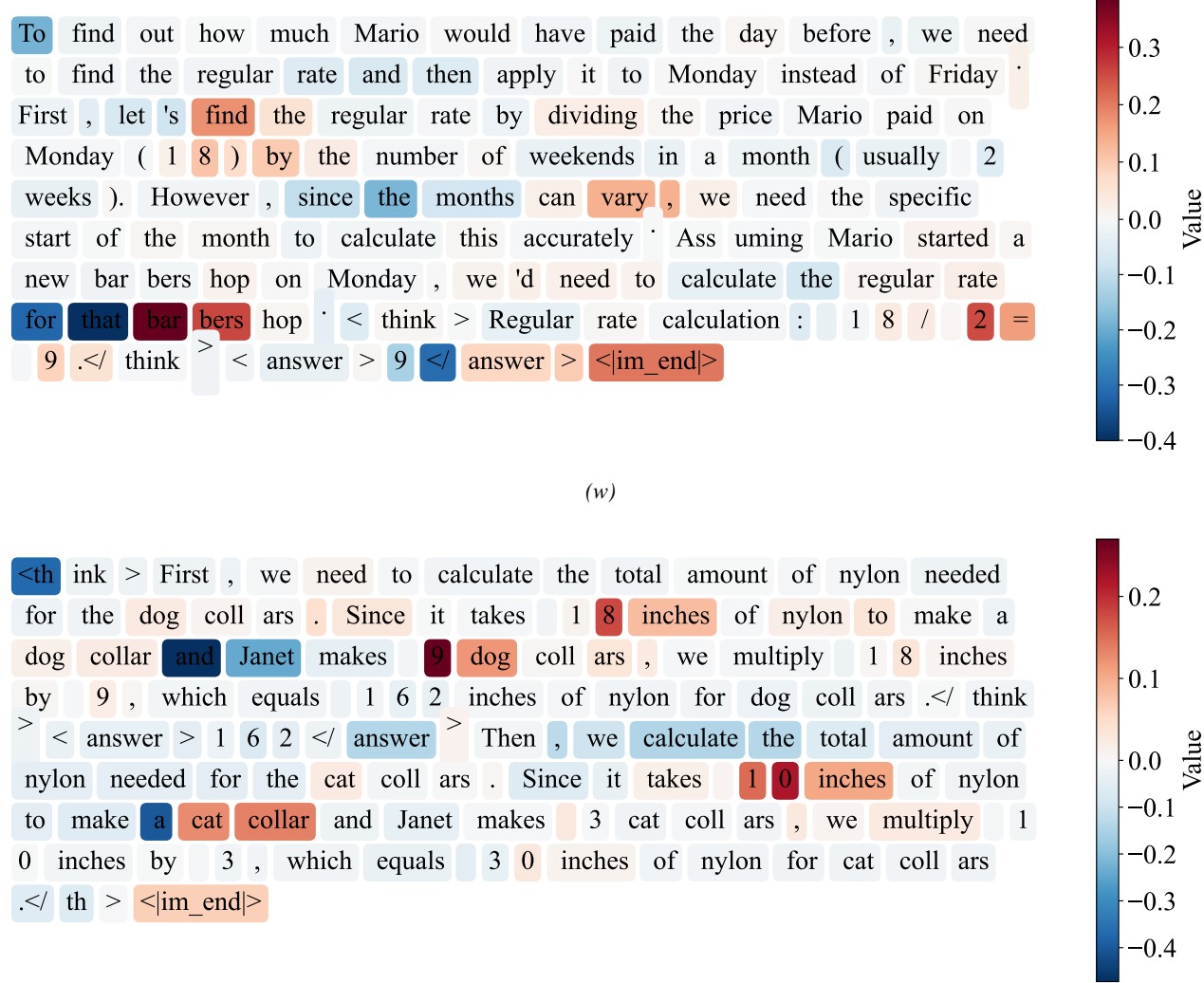

*(w)*

*(x)*

*Figure 12.* Case study on the effectiveness of the conditional MI in identifying informative tokens. Each subplot is a reasoning completion generated by Qwen2.5-0.5B-Instruct on a question from the GSM8K dataset. (Part 12 of 13)

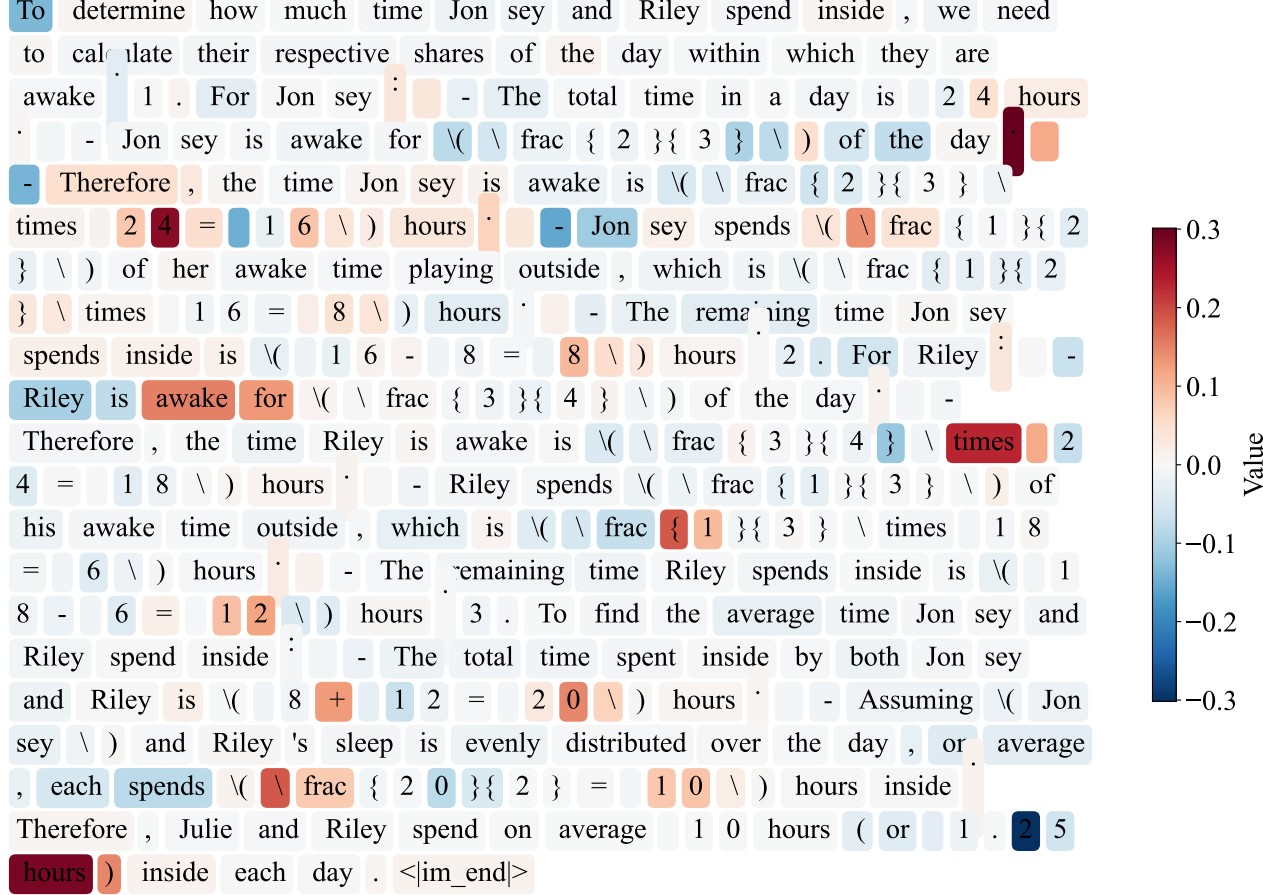

*(y)*

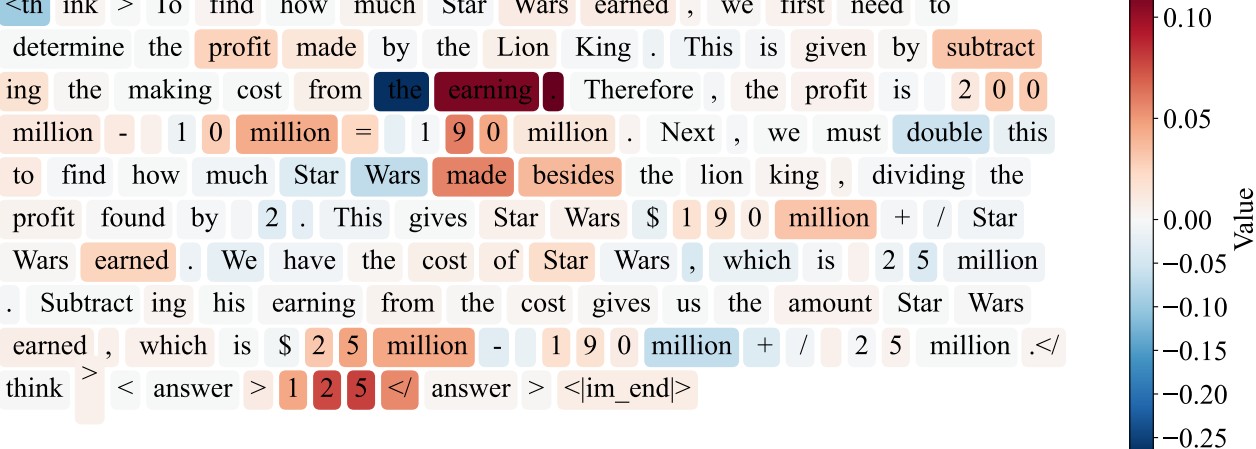

*(z)*

*Figure 12.* Case study on the effectiveness of the conditional MI in identifying informative tokens. Each subplot is a reasoning completion generated by Qwen2.5-0.5B-Instruct on a question from the GSM8K dataset. (Part 13 of 13)

*Table 6.* Comparison of completions across different methods

| Problem | GFPO | GTPO | SGRPOLee | DAPO | IAPO |
|---|---|---|---|---|---|
| Robe problem | For the robe, it requires 2 bolts of blue fiber. Since the amount of white fiber needed is half that of the blue fiber, we calculate half of 2 bolts, which is 1 bolt. Adding the blue and white fibers together, we get 2 + 1 = 3 bolts in total. Answer: 3 | To determine the total number of bolts needed for the robe, we first look at the amount of each type of fiber required. The problem states that a robe requires 2 bolts of blue fiber and half that much white fiber. Since half of 2 is 1, the robe will require 1 bolt of white fiber. This means the total number of bolts needed is the sum of the bolts of blue fiber and the bolts of white fiber. So, 2 (blue fiber) + 1 (white fiber) = 3 bolts in total. The robe will need a total of 3 bolts of fiber. Answer: 3 | First, we calculate how much white fiber is needed, which is half the amount of blue fiber. Since the robe takes 2 bolts of blue fiber, the amount of white fiber needed is 2 / 2 = 1 bolt. The total amount of fiber needed is the sum of the blue and white fibers, so 2 + 1 = 3 bolts. Answer: 3 | First, we need to determine how much white fiber is required. Since the amount of white fiber is half of the blue fiber, we have 2/2 = 1 bolt of white fiber. Then, we add the bolts of blue fiber and white fiber together: 2 + 1 = 3 bolts in total. Answer: 3 | 2 divided by 2 is 1, so the total is 2 + 1 = 3. Answer: 3 |
| Duck eggs problem | First, we calculate the total number of eggs Janet has per day which is 16. Janet consumes 3 eggs for breakfast every day, leaving 16 - 3 = 13 eggs. Then she uses 4 eggs to bake muffins for her friends, leaving 13 - 4 = 9 eggs. Since she sells the remaining eggs at the farmers' market for $2 each, she makes 9 * $2 = $18 per day from selling eggs. Answer: 18 | To find out how much money Janet makes every day at the farmers' market, let's first determine how many eggs she has left after consuming some for her personal use. Janet's ducks lay 16 eggs per day. She eats 3 eggs for breakfast, so the number of eggs left after breakfast is 16 - 3 = 13 eggs. She also uses 4 eggs to bake muffins for her friends, so the number of eggs left after baking is 13 - 4 = 9 eggs. These 9 eggs are what she sells at the farmers' market. Since each egg is sold for $2, the amount of money she makes from selling 9 eggs is 9 * 2 = 18 dollars. Therefore, Janet makes $18 every day at the farmers' market. Answer: 18 | Janet's ducks lay 16 eggs per day. She eats 3 eggs every morning for breakfast, so the number of eggs used for breakfast is 3. She also uses 4 eggs per day to bake muffins for her friends, so the number of eggs used for baking is 4. The total number of eggs used every day is 3 + 4 = 7 eggs. The number of eggs left to sell at the farmers' market is 16 - 7 = 9 eggs. She sells the remaining eggs at $2 per egg, so the amount she makes from selling the eggs is 9 * 2 = 18 dollars. Answer: 18 | Janet's ducks lay 16 eggs per day. She eats 3 eggs every morning for breakfast and uses 4 eggs every day to bake muffins for her friends. In total, she uses 3 + 4 = 7 eggs every day. The number of eggs left to sell at the farmers' market is 16 - 7 = 9 eggs. She sells each egg for $2, so the amount she makes every day at the farmers' market is 9 * $2 = $18. Answer: 18 | Janet gets 16 - 3 - 4 = 9 eggs per day. At $2 per egg, she makes 9 * 2 = 18 dollars. Answer: 18 |

Table 6 – *Continued from previous page*

| Problem | GFPO | GTPO | SGRPOLee | DAPO | IAPO |
|---------|------|------|----------|------|------|
| Sprints problem | To find out how many total meters James runs in a week, we first determine how many meters he runs in one session. Since he runs 3 sprints each session and each sprint is 60 meters, he runs 3 * 60 = 180 meters per session. Since he does this 3 times a week, the total meters he runs in a week is 180 * 3 = 540 meters. Answer: 540 | Let's break down the problem step by step. First, we need to find out how many meters James runs in a single sprint session. Since each sprint is 60 meters and he does 3 sprints per session, the total distance he runs in one session is 3 * 60 = 180 meters. Next, we need to find out how many meters he runs in a week. Since he does these sprint sessions 3 times a week, the total distance he runs in a week is 180 * 3 = 540 meters. Therefore, James runs 540 meters in a week. Answer: 540 | First, we calculate the number of meters James runs in one session. He runs 3 sprints and each sprint is 60 meters, so in one session, he runs 3 * 60 = 180 meters. Then we calculate the total for the week. He runs 3 times a week, so the total meters he runs in a week is 180 * 3 = 540 meters. Answer: 540 | First, we calculate the total distance James runs in one sprint session. Since he runs 60 meters each sprint and does 3 sprints per session, the distance covered in one session is 60 * 3 = 180 meters. Next, since he does 3 sessions a week, the total distance he runs in a week is 180 * 3 = 540 meters. Answer: 540 | Each sprint is 60 meters, and he does 3 sprints 3 times a week, so 3 * 60 * 3 = 540 meters. Answer: 540 |
| Glasses problem | To solve this, we need to calculate the total cost of 16 glasses, where every second glass is at 60% of the price. First, let's determine the cost of one pair of glasses: the first glass is $5 and the second is 60% of $5, which is $3. The total for one pair is $5 + $3 = $8. Since Kylar is buying 16 glasses, which form 8 pairs, we multiply the cost of one pair by 8: $8 * 8 = $64. Answer: 64 | To determine the total cost of 16 glasses, we need to consider the pricing scheme. The first glass is priced at $5, and every second glass is priced at 60% of $5, which is $5 * 0.6 = $3. Therefore, the cost of a pair (1 + 1) of glasses is $5 + $3 = $8. Since Kylar is buying 16 glasses, we can imagine the sequence in pairs, and since 16 is an even number, there are exactly 8 pairs (because 16 / 2 = 8). So, for 8 pairs, the total cost is 8 * 8 = 64. Hence, Kylar needs to pay $64 for 16 glasses. Answer: 64 | First, we need to determine the cost for one pair of glasses (two glasses). The first glass costs $5 and the second glass costs 60% of $5, which is $5 * 0.6 = $3. So the total cost for one pair of glasses is $5 + $3 = $8. Kylar wants to buy 16 glasses, which is 8 pairs of glasses since 16 / 2 = 8. The total cost for 8 pairs of glasses at $8 each is 8 * $8 = $64. Answer: 64 | First, let's calculate the cost of one pair of glasses, where the second glass costs 60% of $5. So, the cost of the second glass is 0.60 * $5 = $3. Therefore, the cost of one pair of glasses is $5 + $3 = $8. Since Kylar wants to buy 16 glasses, and he gets a discounted second glass for every pair, he will have 8 pairs of glasses. The total cost will be 8 * $8 = $64. Answer: 64 | Every second pair of glasses costs 0.6 * 5 = $3. For 16 glasses, we can group them into 8 pairs, so 8 * (5 + 3) = 8 * 8 = 64. Answer: 64 |

*Continued on next page*

Table 6 – *Continued from previous page*

| Problem | GFPO | GTPO | SGRPOLee | DAPO | IAPO |
|---|---|---|---|---|---|
| Chicken feed problem | First, we need to calculate how many cups of feed Wendi provides to each chicken in the morning and afternoon. In the morning, she gives 15 cups, and in the afternoon, she gives 25 cups. Since she feeds each chicken 15/20 = 0.75 cups in the morning, and 25/20 = 1.25 cups in the afternoon, the total amount each chicken gets in these two meals is 0.75 + 1.25 = 2 cups. Since each chicken should get 3 cups of feed per day, the final meal must make up the difference: 3 - 2 = 1 cup of feed per chicken. With 20 chickens, Wendi needs to provide 20 * 1 = 20 cups of feed in the final meal. Answer: 20 | Wendi gives her chickens 15 cups of feed in the morning and 25 cups of feed in the afternoon. First, we need to determine how many cups of feed each chicken gets per day. Since there are three feedings a day and each chicken should get 3 cups of feed per day, the total feed required for all 20 chickens per day is 20 * 3 = 60 cups. In the first two feedings (morning and afternoon), Wendi has already provided 15 + 25 = 40 cups of feed. To find out how much feed is needed for the final feeding, we subtract the amount already given from the total daily requirement: 60 - 40 = 20 cups. Therefore, Wendi needs to give her chickens 20 cups of feed in the final feeding of the day. Answer: 20 | First, we calculate the total amount of feed given to the chickens in a day. Wendi gives 15 cups in the morning and 25 cups in the afternoon, so the total for these two meals is 15 + 25 = 40 cups. Each chicken is given 3 cups of feed per day. Since there are three meals, the total daily feed needed is 3 * 20 * 3 = 180 cups. We already know the morning and afternoon feed totals, so the feed for the final meal is 180 - 40 = 140 cups. Answer: 140 | First, we need to calculate the total amount of feed that each chicken consumes daily. Since each chicken gets 3 cups of feed and there are 20 chickens, the daily requirement is 20 * 3 = 60 cups of feed. Wendi gives her chickens 15 cups in the morning and 25 cups in the afternoon, so the total feed given in these two meals is 15 + 25 = 40 cups. To find out how much feed is needed in the final meal, we subtract: 60 - 40 = 20 cups. Answer: 20 | First, calculate the total daily feed required: 20 * 3 = 60 cups. Then, the remaining feed is 60 - (15 + 25) = 60 - 40 = 20 cups. Answer: 20 |

*Continued on next page*

Table 6 – *Continued from previous page*

| Problem | GFPO | GTPO | SGRPOLee | DAPO | IAPO |
|---------|------|------|----------|------|------|
| Lemon tree problem | To determine how many years it will take for Carlos to start earning money from his lemon tree, we first need to calculate the net income each year. Each year, the tree grows 7 lemons, and he sells them for $1.5 each, resulting in a revenue of $7 \times 1.5 = 10.5$ dollars. The cost to water and feed the tree each year is $3. Therefore, the net income each year is $10.5 - 3 = 7.5$ dollars. The initial cost to plant the tree is $90. To find out how many years it will take to recover this cost, we divide: $90/7.5 = 12$. Thus, it will take 12 years. Answer: 12 | Let's break down the problem step by step. First, the initial cost of planting the lemon tree is $90. Next, each year, the tree will grow 7 lemons, which Carlos can sell at $1.5 per lemon. The cost to water and feed the tree each year is $3. To determine how many years it will take before Carlos starts earning money, we need to find the point where his total earnings from selling lemons exceed the initial planting cost plus the yearly costs. The profit each year from selling lemons is 7 * 1.5 = 10.5 dollars. Each year, the net profit (after subtracting the yearly cost) is 10.5 - 3 = 7.5 dollars. Now, we need to find out how many years it will take for the total net profit to exceed the initial planting cost of $90. We can set up the equation: 7.5 * n = 90, where n is the number of years. Solving for n, we get n = 90 / 7.5 = 12. Therefore, it will take 12 years for Carlos to start earning money on the lemon tree. Answer: 12 | The tree costs $90 to plant. Each year the tree produces 7 lemons at $1.5 each, so revenue is $7 \times 1.5 = 10.5$ dollars. Annual cost is $3. Net profit per year is $10.5 - 3 = 7.5$ dollars. Years to break even: $90/7.5 = 12$ years. Answer: 12 | To determine how many years it will take for Carlos to start earning money on the lemon tree, we need to calculate the costs and income over time. Let's denote the number of years by $n$. Annual income from lemons: $7 \times 1.5 = 10.5$ dollars. Annual cost: $3. Net annual profit: $7.5n - 90$. We want the smallest $n$ such that net profit is positive: $7.5n - 90 > 0$. Solving: $7.5n > 90$, thus $n > 12$. Since $n$ must be a whole number, we round up to 13. Therefore, it will take 13 years. Answer: 13 | First, we calculate the net profit: 7 * 1.5 - 3 = 10.5 - 3 = 7.5. To break even, we need 90 / 7.5 = 12 years. Answer: 12 |

*Continued on next page*

Table 6 – *Continued from previous page*

| Problem | GFPO | GTPO | SGRPOLee | DAPO | IAPO |
|---|---|---|---|---|---|
| Sheep problem | To solve this problem, we start by determining the number of sheep in each city. If Seattle has 20 sheep, then Charleston has $4 \times 20 = 80$ sheep. Toulouse has $2 \times 80 = 160$ sheep. Total: $20 + 80 + 160 = 260$ sheep. Answer: 260 | Let's start by figuring out how many sheep each city has based on the given relationships. We know that Seattle has 20 sheep. The problem states that Charleston has 4 times as many sheep as Seattle. Since Seattle has 20 sheep, we can calculate the number of sheep in Charleston as 4 * 20 = 80 sheep. Next, we are told that Toulouse has twice as many sheep as Charleston. Since Charleston has 80 sheep, we can calculate the number of sheep in Toulouse as 2 * 80 = 160 sheep. Now, to find the total number of sheep in Toulouse, Charleston, and Seattle together, we add up the sheep from each city: 160 (Toulouse) + 80 (Charleston) + 20 (Seattle) = 260 sheep. Therefore, the total number of sheep in all three cities is 260. Answer: 260 | First, calculate Charleston: $4 \times 20 = 80$ sheep. Next, Toulouse: $2 \times 80 = 160$ sheep. Finally, sum: $160 + 80 + 20 = 260$ sheep. Answer: 260 | First, Charleston has $4 \times 20 = 80$ sheep. Next, Toulouse has $2 \times 80 = 160$ sheep. Now, we add: $160 + 80 + 20 = 260$ sheep. Answer: 260 | Seattle has 20 sheep. Charleston has $4 * 20 = 80$ sheep. Toulouse has $2 * 80 = 160$ sheep. Therefore, $20 + 80 + 160 = 260$. Answer: 260 |
| Eliza's earnings | Eliza earns $40 \times 10 = 400$ dollars for first 40 hours. For 5 overtime hours at $10 \times 1.2 = 12$ dollars/hour, she earns $5 \times 12 = 60$ dollars. Total: $400 + 60 = 460$ dollars. Answer: 460 | Let's break down the problem step by step to calculate Eliza's total earnings. First, we need to determine her pay for the regular hours. Eliza's regular pay rate is $10 per hour. She worked 45 hours this week, and the first 40 hours are considered regular hours, so for the first 40 hours, her pay is 40 * 10 = 400 dollars. Next, we calculate her overtime pay. Any hours worked beyond 40 are considered overtime, and the overtime pay rate is 1.2 times the regular rate. The overtime rate is therefore 10 * 1.2 = 12 dollars per hour. She worked 45 - 40 = 5 overtime hours. The pay for these 5 overtime hours is 5 * 12 = 60 dollars. Finally, we add her regular pay and overtime pay to find her total earnings: 400 + 60 = 460 dollars. Therefore, Eliza's total earnings for this week are 460 dollars. Answer: 460 | Eliza worked 45 hours. Regular pay for first 40 hours: $40 \times 10 = 400$ dollars. Remaining 5 hours are overtime at $1.2 \times 10 = 12$ dollars/hour. Overtime earnings: $5 \times 12 = 60$ dollars. Total: $400 + 60 = 460$ dollars. Answer: 460 | Eliza's regular pay: $40 \times 10 = 400$ dollars. Overtime hours: $45 - 40 = 5$ hours. Overtime rate: $1.2 \times 10 = 12$ dollars/hour. Overtime pay: $5 \times 12 = 60$ dollars. Total: $400 + 60 = 460$ dollars. Answer: 460 | Eliza's regular pay is 40 * \$10 = \$400. For the 5 hours of overtime, her pay is 5 * (10 * 1.2) = \$60. Thus, her total earnings are \$400 + \$60 = \$460. Answer: 460 |

*Continued on next page*

Table 6 – *Continued from previous page*

| Problem | GFPO | GTPO | SGRPOLee | DAPO | IAPO |
|---|---|---|---|---|---|
| Pastries | Cost of 3 dozen donuts: $3 \times 68 = 204$ dollars. Cost of 2 dozen mini cupcakes: $2 \times 80 = 160$ dollars. Cost of 6 dozen mini cheesecakes: $6 \times 55 = 330$ dollars. Total: $204 + 160 + 330 = 694$ dollars. Answer: 694 | To find the total cost of the items Toula bought, we need to calculate the cost for each type of pastry and then sum these costs. First, we calculate the cost of the donuts. Toula bought 3 dozen donuts, and each dozen costs \$68. Therefore, the cost of the donuts is 3 dozen * \$68 per dozen = 3 * 68 = \$204. Next, we calculate the cost of the mini cupcakes. Toula bought 2 dozen mini cupcakes, and each dozen costs \$80. Thus, the cost of the mini cupcakes is 2 dozen * \$80 per dozen = 2 * 80 = \$160. Finally, we calculate the cost of the mini cheesecakes. Toula bought 6 dozen mini cheesecakes, and each dozen costs \$55. Therefore, the cost of the mini cheesecakes is 6 dozen * \$55 per dozen = 6 * 55 = \$330. To find the total cost, we add these amounts together: \$204 + \$160 + \$330 = \$694. Therefore, the total cost of the pastries is \$694. Answer: 694 | First, donuts: $3 \times 68 = 204$ dollars. Next, mini cupcakes: $2 \times 80 = 160$ dollars. Then, mini cheesecakes: $6 \times 55 = 330$ dollars. Total: $204 + 160 + 330 = 694$ dollars. Answer: 694 | First, donuts: $3 \times 68 = 204$ dollars. Next, mini cupcakes: $2 \times 80 = 160$ dollars. Then, mini cheesecakes: $6 \times 55 = 330$ dollars. Total: $204 + 160 + 330 = 694$ dollars. Answer: 694 | Calculating the cost for each item: 3 * 68 = 204, 2 * 80 = 160, and 6 * 55 = 330. The total cost is 204 + 160 + 330 = 694. Answer: 694 |
| Download | First month: 60 downloads. Second month: $3 \times 60 = 180$ downloads. Third month reduced by 30%: $0.30 \times 180 = 54$, so $180 - 54 = 126$ downloads. Total: $60 + 180 + 126 = 366$ downloads. Answer: 366 | Let's break down the problem step by step. First, we know the number of downloads in the first month is 60. In the second month, the number of downloads is three times the first month's downloads. So, we calculate the downloads for the second month as 3 * 60 = 180. In the third month, the downloads are reduced by 30%, meaning 70% of the second month's downloads. To find 70% of 180, we calculate 0.70 * 180 = 126. Therefore, the third month's downloads are 126. To find the total downloads over the three months, we sum the downloads from each month: 60 + 180 + 126. Adding these together, we get 60 + 180 + 126 = 366. Therefore, the total downloads over the three months are 366. Answer: 366 | First month: 60. Second month: $3 \times 60 = 180$. Third month: $180 - (30/100 \times 180) = 180 - 54 = 126$. Total: $60 + 180 + 126 = 366$. Answer: 366 | First month: 60 downloads. Second month: $3 \times 60 = 180$ downloads. Third month 30% reduction: $0.30 \times 180 = 54$ downloads, so $180 - 54 = 126$ downloads. Total: $60 + 180 + 126 = 366$ downloads. Answer: 366 | In the second month, 60 * 3 = 180. In the third month, 180 - (180 * 0.3) = 180 - 54 = 126. Total downloads = 60 + 180 + 126 = 366. Answer: 366 |

*Continued on next page*

Table 6 – *Continued from previous page*

| Problem | GFPO | GTPO | SGRPOLee | DAPO | IAPO |
|---|---|---|---|---|---|
| Dance class | Contemporary dance: $0.20 \times 20 = 4$ students. Remaining: $20 - 4 = 16$ students. Jazz dance: $0.25 \times 16 = 4$ students. Hip-hop: $16 - 4 = 12$ students. Percentage: $(12/20) \times 100 = 60\%$. Answer: 60 | To find the percentage of students in the hip-hop dance class, let's start by determining how many students are in each type of dance class. First, 20% of the 20 students signed up for contemporary dance. To calculate this, we multiply 20 by 20%, which is 20 * 0.20 = 4 students. This means 4 students are in the contemporary dance class. After signing up 4 students for contemporary dance, the number of students remaining is 20 - 4 = 16 students. Next, 25% of the remaining 16 students signed up for jazz dance. To calculate this, we multiply 16 by 25%, which is 16 * 0.25 = 4 students. This means 4 students are in the jazz dance class. After signing up 4 students for jazz dance, the number of students remaining for hip-hop is 16 - 4 = 12 students. To find the percentage of the total students who signed up for hip-hop, we divide the number of hip-hop students by the total number of students and multiply by 100. This is (12 / 20) * 100 = 60%. Therefore, 60% of the students signed up for hip-hop dance. Answer: 60 | Contemporary dance: $20 \times 0.2 = 4$ students. Remaining: $20 - 4 = 16$ students. Jazz dance: $16 \times 0.25 = 4$ students. Remaining: $16 - 4 = 12$ students (hip-hop). Percentage: $(12/20) \times 100 = 60\%$. Answer: 60 | Contemporary dance: $(20/100) \times 20 = 4$ students. Remaining: $20 - 4 = 16$ students. Jazz dance: $(25/100) \times 16 = 4$ students. Hip-hop: $16 - 4 = 12$ students. Percentage: $(12/20) \times 100 = 60\%$. Answer: 60 | First, 20% of 20 students = 4, so 20 - 4 = 16. Then, 25% of 16 = 4, so 16 - 4 = 12. Thus, 12 out of 20 students are in hip-hop, which is 12/20 = 60%. Answer: 60 |

*Continued on next page*

Table 6 – *Continued from previous page*

| Problem | GFPO | GTPO | SGRPOLee | DAPO | IAPO |
|---------|------|------|----------|------|------|
| File download | 40% of 200 GB is 80 GB. At 2 GB/minute, this takes $80/2 = 40$ minutes. After restart (20 minutes), she downloads full 200 GB: $200/2 = 100$ minutes. Total: $40 + 20 + 100 = 160$ minutes. Answer: 160 | Carla is downloading a 200 GB file at a rate of 2 GB per minute. This means that the total download time without interruptions would be 200 GB / 2 GB/minute = 100 minutes. However, she encounters an interruption 40% into the download. To find out how much data she has downloaded by 40%, we calculate 40% of 200 GB: 200 * 0.4 = 80 GB. Since she is downloading at a rate of 2 GB per minute, it would take her 80 GB / 2 GB/minute = 40 minutes to download 80 GB. At this point, Windows forces a restart, and the download process restarts from the beginning. After the restart, the total data to be downloaded is 200 GB again, and the download process has to start from the beginning. During the restart, it takes 20 minutes to download 80 GB. But since she is already at 40 minutes and the Windows restart takes 20 minutes, the download restarts and then needs to complete the remaining 120 GB (200 GB - 80 GB). The rate of download is still 2 GB/minute, so it takes her 120 GB / 2 GB/minute = 60 minutes to download the remaining 120 GB. Therefore, the total time taken to download the file is the sum of the initial 40 minutes, the restart time of 20 minutes, and the restart download time of 60 minutes, which is 40 + 20 + 60 = 120 minutes. So, the total time taken to download the 200 GB file is 120 minutes. Answer: 120 | File size: 200 GB. 40% is $200 \times 0.4 = 80$ GB. Time to download 80 GB: $80/2 = 40$ minutes. Restart takes 20 minutes. Total until restart: $40 + 20 = 60$ minutes. Then she restarts from beginning, downloading full 200 GB: $200/2 = 100$ minutes. Total: $40 + 20 + 100 = 160$ minutes. Answer: 160 | 40% of 200 GB is $0.4 \times 200 = 80$ GB. Time: $80/2 = 40$ minutes. After restart, she downloads full 200 GB: $200/2 = 100$ minutes. Plus 20 minutes for restart. Total: $40 + 20 + 100 = 160$ minutes. Answer: 160 | 40% of 200 GB is $200 * 0.4 = 80$ GB. 80 GB / 2 = 40 minutes. The remaining 200 - 80 = 120 GB takes 120 / 2 = 60 minutes. The total time is 40 + 20 + 60 = 120 minutes. Answer: 120 |

Table 6 – *Continued from previous page*

| Problem | GFPO | GTPO | SGRPOLee | DAPO | IAPO |
|---------|------|------|----------|------|------|
| Driving | John drives $3 \times 60 = 180$ miles away. Returning: 2 hours in traffic (0 miles), 0.5 hours at 30 mph ($0.5 \times 30 = 15$ miles), remaining time $4 - 2 - 0.5 = 1.5$ hours at 80 mph ($1.5 \times 80 = 120$ miles). Total return: $15 + 120 = 135$ miles. Distance from home: $180 - 135 = 45$ miles. Answer: 45 | John drives at a speed of 60 mph for 3 hours, so the distance he drives away from home is 60 * 3 = 180 miles. On the return trip, John spends 2 hours in standstill traffic, which means he doesn't cover any distance during this time. After the traffic, he drives for the remaining time at different speeds. The total time for the return trip is 4 hours. Since he spent 2 hours in traffic, he has 4 - 2 = 2 hours left. During the first part of the remaining time, John drives at 30 mph for half an hour (0.5 hours). The distance covered during this time is 30 * 0.5 = 15 miles. For the rest of the return trip, he drives at 80 mph. The remaining time after driving at 30 mph is 2 - 0.5 = 1.5 hours. The distance covered during this time is 80 * 1.5 = 120 miles. Therefore, the total distance John covers on the return trip is 0 (from the 2 hours in traffic) + 15 (from driving at 30 mph) + 120 (from driving at 80 mph) = 135 miles. Since John drove 180 miles away from home, and he covered 135 miles on the return trip, the distance he is from home is 180 - 135 = 45 miles. Answer: 45 | Initial distance: $3 \times 60 = 180$ miles. After traffic and 0.5 hours at 30 mph ($0.5 \times 30 = 15$ miles), he's $180 - 15 = 165$ miles away. Remaining time: $4 - 2 - 0.5 = 1.5$ hours at 80 mph: $80 \times 1.5 = 120$ miles. Total return: $15 + 120 = 135$ miles. Distance from home: $180 - 135 = 45$ miles. Answer: 45 | Initial distance: $3 \times 60 = 180$ miles. Return: 2 hours traffic (0 miles), 0.5 hours at 30 mph ($0.5 \times 30 = 15$ miles), 1.5 hours at 80 mph ($1.5 \times 80 = 120$ miles). Total return: $15 + 120 = 135$ miles. Distance from home: $180 - 135 = 45$ miles. Answer: 45 | First, John drives 3 * 60 = 180 miles. Turning around, he spends 2 * 0 = 0 miles (in standstill traffic), then 0.5 * 30 = 15 miles, and finally 3.5 * 80 = 280 miles. So, the total distance from home is 180 + 15 + 280 = 475 miles. Answer: 475 |

Table 6 – *Continued from previous page*

| Problem | GFPO | GTPO | SGRPOLee | DAPO | IAPO |
|---|---|---|---|---|---|
| House flip-ping | Purchase: $80,000. Repairs: $50,000. Total investment: $130,000. Value increase: $80,000 \times 1.5 = 120,000$ dollars. New value: $80,000 + 120,000 = 200,000$ dollars. Profit: $200,000 - 130,000 = 70,000$ dollars. Answer: 70000 | Josh initially buys a house for $80,000 and then spends an additional $50,000 on repairs. The total cost for Josh is the sum of the house price and the repair cost. So, let's calculate the total cost: 80,000 + 50,000 = 130,000 dollars. Next, we need to find the gain in value from the repairs. The repair cost of $50,000 increased the value of the house by 150%. To find out the new value of the house after the 150% increase in value, we first understand that a 150% increase means the value is tripled (since 100% + 150% = 250% of the repair cost, or 2.5 times the repair cost). Thus, the increase in value due to repairs is 50,000 * 1.5 = 75,000 dollars. Adding this increase to the original house purchase price of $80,000, the new value of the house is 80,000 + 75,000 = 155,000 dollars. Josh needs to sell the house to make a profit. The profit is calculated by subtracting the total cost from the new value of the house: 155,000 - 130,000 = 25,000 dollars. Therefore, Josh made a profit of $25,000. Answer: 25000 | Total spent: $80,000 + 50,000 = 130,000$ dollars. Value increase: $80,000 \times 1.5 = 120,000$ dollars. New value: $80,000 + 120,000 = 200,000$ dollars. Profit: $200,000 - 130,000 = 70,000$ dollars. Answer: 70000 | Initial cost before repairs: $80,000. Repairs: $50,000. Total cost: $130,000. Increase of 150% means value goes up by $1.5 \times 130,000 = 195,000$ dollars. New value: $130,000 + 195,000 = 325,000$ dollars. Profit: $325,000 - 130,000 = 195,000$ dollars. Answer: 195000 | First, calculate the increased value: $80,000 * 1.5 = $120,000. The new value of the house is $80,000 + $50,000 = $130,000. The profit is $130,000 - ($80,000 + $50,000) = $130,000 - $130,000 = $0. Answer: 0 |

*Continued on next page*

Table 6 – *Continued from previous page*

| Problem | GFPO | GTPO | SGRPOLee | DAPO | IAPO |
|---|---|---|---|---|---|
| Vacuum cleaners | Let $x$ be total vacuum cleaners. Sold $x/3$ at green house. Remaining: $2x/3$. Sold 2 more at red house: $2x/3 - 2$ left. Sold half at orange house, leaving: $(1/2)(2x/3 - 2) = 5$. Solving: $2x/3 - 2 = 10$, $2x/3 = 12$, $x = 18$. Answer: 18 | Melanie sold some of her vacuum cleaners and has 5 left. Let's break down her sales process step-by-step. Suppose Melanie started with x vacuum cleaners. First, she sold a third of the vacuum cleaners at the green house, which means she sold x/3 vacuum cleaners there. After selling at the green house, she had x - x/3 = 2x/3 vacuum cleaners left. Next, she sold 2 more to the red house, so she then had 2x/3 - 2 vacuum cleaners left. Finally, she sold half of what was left at the orange house, which means she sold (1/2)(2x/3 - 2) vacuum cleaners there. Therefore, the number of vacuum cleaners she has left after the orange house sale is (1/2)(2x/3 - 2). We know that after the sales at the green, red, and orange houses, she has 5 vacuum cleaners left. So, we can set up the equation: (1/2)(2x/3 - 2) = 5. To solve for x, we can multiply both sides by 2 to clear the fraction: 2x/3 - 2 = 10. Next, we add 2 to both sides to isolate the term with x: 2x/3 = 12. Then, we multiply both sides by 3/2 to solve for x: x = 12 * 3/2 = 18. So, Melanie started with 18 vacuum cleaners. We can verify this: if she started with 18, after selling a third at the green house, she sold 18/3 = 6 vacuum cleaners, leaving her with 12. Then, she sold 2 more to the red house, leaving her with 12 - 2 = 10. Lastly, she sold half of the 10 left at the orange house, which is 5, leaving her with 5. This confirms our solution. Therefore, Melanie started with 18 vacuum cleaners. Answer: 18 | Let $x$ be total. Sold $x/3$ at green house. After green house: $2x/3$ left. After red house: $2x/3 - 2$ left. Sold half at orange house: $(1/2)(2x/3 - 2)$. After orange house: $2x/3 - 2 - (1/2)(2x/3 - 2) = 5$. Simplifying: $2x/3 - 2 - (2x/6 - 1) = 5$, $2x/3 - 2 - x/3 + 1 = 5$, $x/3 - 1 = 5$, $x/3 = 6$, $x = 18$. Answer: 18 | Let $x$ be total. Sold $x/3$ at green house. After green: $2x/3$ left. After red: $2x/3 - 2$ left. Sold half at orange: $(1/2)(2x/3 - 2) = x/3 - 1$. After orange: $2x/3 - 2 - (x/3 - 1) = x/3 - 1 = 5$. Solving: $x/3 = 6$, $x = 18$. Answer: 18 | Let's denote the total number of vacuum cleaners by $x$. She sold $\frac{x}{3}$ at the green house, and then 2 more to the red house, leaving $\frac{2x}{3} - 2$. The remaining $\frac{1}{2}\left(\frac{2x}{3} - 2\right) = 5$, simplifying to $\frac{x}{3} - 1 = 5$. So, $\frac{x}{3} = 6$ and $x = 18$. Answer: 18 |

