# OpenReview forum: "IAPO: Information-Aware Policy Optimization for Token-Efficient Reasoning"
_ICML.cc/2026/Conference — ICML 2026 regular_

### Official Review · Reviewer_gPX7 · 2026-02-14

**Soundness:** 2
**Presentation:** 2
**Significance:** 3
**Originality:** 2
**Overall Recommendation:** 4
**Confidence:** 2

**Summary:**

This paper mainly focuses on how to make large language models more "concise" during reasoning, avoiding the situation where they ramble on and write a lot of unnecessary steps. The author proposed the IAPO method, and the core idea is quite interesting: using the conditional mutual information in information theory to determine how important each token is to the final answer. The more important it is, the more "reward" it will receive; the less useful it is, the less reward or even punishment it will get.

**Compliance With Llm Reviewing Policy:**

Affirmed.

**Final Justification:**

I will keep my initial rates.

**Key Questions For Authors:**

However, there are a few aspects that cause some concern.

1. Theoretical part demonstrated that the length would decrease. But does the assumption that "average information quantity monotonically decreases with length" always hold true in actual complex reasoning? For instance, some difficult problems might require taking a detour before getting back on track. The seemingly redundant steps in the middle can actually be helpful for achieving the final correctness.

2. The main focus of the experiment was on mathematical reasoning. Although the appendix mentioned common sense reasoning, no validation was seen in tasks that require a large amount of background knowledge or complex logical chains (such as legal reasoning and scientific question answering). The generalization ability remains to be observed.

3. Finally, the tuning of hyperparameters α and β seems quite sensitive. The paper does not clearly explain how to automatically select them, and in practical applications, it might take a considerable amount of time to fine-tune these parameters. In addition, there is a lack of experiments on the latest model, such as Qwen3.

**Limitations:**

See Questions

**Strengths And Weaknesses:**

The highlights of this paper are quite obvious. Firstly, using conditional mutual information to quantify the information content of tokens from this perspective is very novel. It is much more reasonable than previous methods that simply decay by position or only consider the overall length. It can indeed distinguish between "key steps" and "boring stuff". The experimental part is also very solid. It has been verified on mainstream datasets such as GSM8K and MATH, and the effectiveness of the method has been demonstrated. Especially, the reduction of 36% in inference length on Qwen2.5-7B is quite impressive. Additionally, the early-exit MI estimator they designed, along with KV cache preloading, is also very practical. Otherwise, even with the most beautiful theory, it would be useless.

---

> ### Author Rebuttal · Authors · 2026-03-31
>
> We sincerely appreciate the reviewer's thoughtful, comprehensive, and detailed feedback. Here, we are dedicated to addressing your concerns point-by-point as follows:
>
> > Concern 1: Does the "average information density decreases with length" assumption always hold for complex reasoning? Some seemingly redundant steps may be necessary.
>
> Thank you for the thoughtful comment. First, we empirically examined the correlation between the information density (i.e., average conditional mutual information (MI) across the reasoning) and the reasoning length, please refer to Fig. 3 in [the anonymous link]( https://anonymous.4open.science/r/agent_rl-107E/). From the figure, we can observe a strong negative correlation between the conditional MI and reasoning length. Second, we believe this is a misunderstanding; we view the conditional MI as the amount of information a token (step) contributes to the final answer. That said, it is possible when the LLM considers a reasoning step important (in the sense of high conditional MI) while it seems non-important to humans. The LLM, in this case, still correctly assigns high reward to the seemingly unnecessary step.
>
>
> > Concern 2: Generalization to tasks requiring background knowledge or complex logical chains is unverified.
>
> We add evaluation in one additional reasoning domain, i.e., ARC-Challenge, a scientific reasoning dataset that requires an LLM to leverage a large amount of its background knowledge beyond mathematics. The results are shown below:
> | Method   | Pass@2         | Pass@4         | Pass@8         | Pass@16        | Pass@32        | Length@2     | Length@4     | Length@8     | Length@16    | Length@32    |
> |----------|-----------------|----------------|----------------|----------------|----------------|--------------|--------------|--------------|--------------|--------------|
> | **IAPO** (ours)     | 0.5964±0.0057  | 0.6388±0.0125  | 0.6800±0.0096  | 0.7213±0.0088  | 0.7514±0.0088  | 79.78±0.49   | 79.49±0.72   | 79.89±0.85   | 79.97±0.57   | 79.81±0.27   |
> | GTPO     | 0.6076±0.0042  | 0.6644±0.0057  | 0.7135±0.0103  | 0.7480±0.0069  | 0.7770±0.0057  | 187.87±4.85  | 185.58±2.35  | 186.98±1.07  | 187.48±0.88  | 186.24±0.87  |
> | DAPO     | 0.5975±0.0016  | 0.6366±0.0042  | 0.6990±0.0055  | 0.7492±0.0098  | 0.7848±0.0126  | 111.51±1.64  | 111.93±0.60  | 112.60±0.46  | 112.57±0.46  | 112.43±0.35  |
> | GFPO     | 0.5563±0.0079  | 0.5831±0.0103  | 0.6054±0.0072  | 0.6187±0.0055  | 0.6399±0.0079  | 131.71±1.89  | 131.87±1.59  | 132.37±0.82  | 132.01±0.46  | 131.39±0.32  |
> | SGRPOLee | 0.5663±0.0096  | 0.6009±0.0155  | 0.6310±0.0123  | 0.6633±0.0150  | 0.7046±0.0057  | 109.92±1.54  | 109.05±2.09  | 110.29±0.61  | 110.28±0.36  | 109.97±0.30  |
>
> We observe that (1) our IAPO achieves the highest effectiveness-efficiency tradeoff; (2) our IAPO achieves effectiveness comparable to the optimal baseline, but reduces 30% in reasoning length. Additionally, we note that the MI-based approach is domain-agnostic: it measures information contribution to the answer regardless of task type. The early-exit estimator requires only that the model can produce some answer distribution from a partial completion, which is broadly true for any generation task.
>
> > Concern 3: Hyperparameter sensitivity of $\alpha$ and $\beta$; lack of automatic selection; no experiments on latest models (e.g., Qwen3).
>
> On hyperparameter sensitivity: Figure 6 and the extended Figure 13 in the paper show that performance is relatively robust across a 4-order-of-magnitude range for $\alpha$ and $\beta$, and that the tradeoff between accuracy and brevity is monotone and predictable (larger $\alpha$ → shorter outputs; larger $\beta$ → higher accuracy at cost of length). This makes manual selection tractable. We will add guidance on default settings (e.g., $\alpha=10^{-4}, \beta=10^{-6}$) in the paper.
>
> On automatic selection: Thank you for the thoughtful comment. A principled approach would be to use multi-objective optimization or to gradually increase $\alpha$ to achieve a target length reduction (e.g., 30%). We will discuss this in the revision.
>
> On Qwen3: Due to time and resource constraints, we are still performing experiments on Qwen3, and the results are not yet available. We are currently testing on Qwen3-0.6B to demonstrate that IAPO's gains generalize to the latest architectures. We will update the results to the [the anonymous link]( https://anonymous.4open.science/r/agent_rl-107E/) as soon as possible.

---

> > ### Author Rebuttal · Reviewer_gPX7 · 2026-04-03
> >
> > Thanks! I keep my initial rates.

---

> > > ### Author Response · Authors · 2026-04-07
> > >
> > > Dear Reviewer gPX7,
> > >
> > >
> > >
> > > Thank you so much for your acknowledgment of our rebuttal. Here are our experiment results for our IAPO and all the baselines on Qwen3-0.6B:
> > >
> > >
> > >
> > > | Method | Pass@2 | Pass@4 | Pass@8 | Pass@16 | Pass@32 | Length@2 | Length@4 | Length@8 | Length@16 | Length@32 |
> > > |-----------|---------------|---------------|---------------|---------------|---------------|----------------|----------------|----------------|-----------------|----------------|
> > > | GFPO | 0.6567±0.0170 | 0.7067±0.0205 | 0.7633±0.0094 | 0.7900±0.0216 | 0.8367±0.0125 | 1085.69±5.17 | 1091.81±5.73 | 1084.66±7.67 | 1088.63±12.52 | 1088.37±6.15 |
> > > | GTPO | 0.5200±0.0216 | 0.6000±0.0216 | 0.6267±0.0205 | 0.6867±0.0205 | 0.7100±0.0082 | 1593.26±7.80 | 1598.92±8.49 | 1594.32±2.74 | 1592.81±6.83 | 1593.09±3.40 |
> > > | DAPO | 0.6333±0.0309 | 0.7000±0.0294 | 0.7500±0.0082 | 0.7967±0.0047 | 0.8433±0.0170 | 1285.02±20.53 | 1278.00±9.23 | 1282.97±1.75 | 1283.79±2.41 | 1282.12±0.58 |
> > > | S-GRPO | 0.5433±0.0249 | 0.5967±0.0249 | 0.6600±0.0000 | 0.7133±0.0094 | 0.7333±0.0170 | 1168.40±21.11 | 1173.37±9.21 | 1153.86±9.09 | 1146.25±2.27 | 1151.25±2.60 |
> > > | IAPO | 0.6100±0.0216 | 0.6833±0.0170 | 0.7533±0.0125 | 0.8000±0.0216 | 0.8400±0.0163 | 1071.48±13.99 | 1080.52±7.32 | 1082.44±5.09 | 1085.80±3.83 | 1083.99±2.46 |
> > >
> > >
> > >
> > > From the table, we can observe that our IAPO achieves the optimal effectiveness-efficiency trade-off: it generates shortest completions while maintaining the highest Pass@k.
> > >
> > >
> > >
> > >
> > >
> > > Sincerely,
> > >
> > > Authors from Submission 21159

---

### Official Review · Reviewer_G4dt · 2026-03-10

**Soundness:** 3
**Presentation:** 3
**Significance:** 2
**Originality:** 3
**Overall Recommendation:** 4
**Confidence:** 4

**Summary:**

This paper proposes IAPO, an RL post-training method for token-efficient inference. It uses the conditional MI between each token and the final answer for token-wise advantage estimation, encouraging highly informative reasoning steps, suppressing redundant output, and adding an exploration adjustment term to avoid over-compression. The authors also approximate conditional MI via early-exit entropy differences, and combine KV-cache with chunked forward passes to reduce computational overhead. Experiments on multiple mathematical reasoning datasets show that IAPO can shorten inference length while maintaining/improving accuracy, and outperform existing baselines on efficiency metrics.

**Compliance With Llm Reviewing Policy:**

Affirmed.

**Final Justification:**

The responses have resolved my concerns and issues. I will raise my score to weak accept.

**Key Questions For Authors:**

Please refer to the weaknesses above.

**Limitations:**

yes

**Strengths And Weaknesses:**

Strengths:
1. Using conditional mutual information to measure each token’s contribution to the final answer is more reasonable than methods that handle it based on length or position.
2. The authors add an exploration adjustment in addition to MI: suppressing entropy when predictions are correct and increasing entropy when they are wrong, reducing the risk of collapse after excessive compression. The paper also analyzes the relevant theory and verifies the effectiveness of this term through experiments.
3. The authors considered the efficiency of real-world deployment and proposed an implementation based on KV-cache preloading and chunk-wise forwarding, demonstrating speedups in experiments.

Weaknesses:
1. This paper relies on the drop in “answer entropy” at early exit as a proxy for conditional MI, but it's currently unclear what the answer distribution actually is (e.g., token-level vs. string-level). And how are multi-token answers handled? How are equivalent strings normalized (e.g., 0.5 vs 1/2)? Whether the probabilities are normalized over the candidate set, and whether any truncation is used (e.g., top-k). These implementation choices could substantially change each token’s score, making the gains brought by the method harder to reproduce.
2. Even with KV-cache and chunk-wise forwarding, IAPO introduces extra forward passes for MI estimation. Without clear GPU compute usage, it's difficult to determine how much of the improvement comes from the algorithm itself versus additional training computation. The paper should report the final gains measured in terms of GPU hours or num of forward passes.
3. The argument for shortening length relies on the claim that “as the model’s output length increases, the average information density decreases.” This seems reasonable, but the paper doesn't provide direct evidence. For example, it could provide plots of the relationship between length and the average $\(s_{i,t}\)$ by dataset, difficulty, and correctness. If this assumption does not hold for harder problems or non-math tasks, the method may overcompress and discard necessary reasoning steps. However, the paper does not report when and why this would happen.
4. Nearly all the experiments in the paper are conducted on pure-text math tasks. I would like to see at least one multimodal and non-math benchmark (e.g., chart/diagram QA or document VQA). In such settings, the early-exit entropy signal may be driven more by visual uncertainty than by linguistic reasoning, which could change what the method considers “informative” (grounding tokens vs. reasoning tokens). Without this, the method’s generalization may be insufficient.

---

> ### Author Rebuttal · Authors · 2026-03-31
>
> We sincerely appreciate the reviewer's comprehensive feedback. Moreover, we are grateful for your acknowledgement of our motivation, methodology, theory, and the performance of our results. We address the reviewer's concerns point by point as follows.
>
> > Concern 1: The answer distribution used for entropy estimation is underspecified (token-level vs. string-level, normalization, multi-token answers, equivalent string handling).
>
> Thank you for the insightful feedback. In our experiment, we use a *token-level* distribution over the first generated answer token given the prompt + postfix. The entropy $H(y | q, o_{\leq t})$ is computed as the Shannon entropy of the LLM's next-token distribution at the final position of the postfix prompt.
> - **Multiple tokens in ground-truth answer**: For multi-token answers (e.g., "142857"), we use only the first answer token's distribution as a proxy. Since the number of tokens in the ground-truth answers of our adopted benchmarks is very small (mostly no more than 3-5), we deem the first token's entropy to be representative of the generation entropy of the entire answer. To fully address the author's concern, we examine the relationship between conditional MI computed with the first generated token's entropy and the one computed with the average entropy of the generated tokens of the same length as the ground-truth answer. From Fig. 2 in [the anonymous link](https://anonymous.4open.science/r/agent_rl-107E/), we see that the two values are strongly correlated. Therefore, we can use the entropy of the first generated token as a good proxy for the entire answer's entropy.
> - **Equivalent string normalization**: For mathematical tasks, we apply canonical normalization (e.g., "0.5" and "1/2" are mapped to the same form before evaluation) following standard practice in math reasoning benchmarks.
> - **Probability normalization**: We use the full vocabulary softmax distribution (no top-k truncation) rather than a normalized distribution across candidates for efficiency.
>
> > Concern 2: Without GPU compute usage, it is hard to determine how much improvement comes from the algorithm vs. additional training computation.
>
> Thank you for the constructive comment. Here, we point the reviewer to a comprehensive training cost table in our rebuttal to Concern 3 of reviewer uFAL, where we could observe that our additional training cost is modest.
>
> > Concern 3: No direct evidence for the claim that "average information density decreases with length."
>
> Thank you for the thoughtful comment. Here, we conduct experiments and add plots in [the anonymous link]( https://anonymous.4open.science/r/agent_rl-107E/) showing the empirical relationship between completion length and average token-wise conditional MI score ($\sum_t s_{i,t} / |o_i|$). Specifically, we generate 16 reasonings for each sample drawed from GSM8K and compute the lengths of these reasonings and the average token-wise conditional MI. As shown in Fig. 2, this holds clearly for the datasets studied, where the reasoning length and average token-wise conditional MI are strongly negatively correlated.
>
> > Concern 4: Nearly all experiments are on pure-text math tasks; no multimodal or non-math benchmarks.
>
> We thank the reviewer for pointing this out. We note that Appendix C.2.1 already evaluates IAPO on CommonsenseQA (Table 2), where IAPO achieves the highest effectiveness (~20% improvement in Pass@k) and strong token efficiency.
> For broader non-math evaluation, we run IAPO on one additional reasoning task (i.e., **ARC-Challenge**).
> | Method   | Pass@16        | Pass@32        | Length@16    | Length@32    |
> |----------|-----------------|----------------|--------------|--------------|
> | **IAPO** (ours)     | 0.7213±0.0088  | 0.7514±0.0088  | 79.97±0.57   | 79.81±0.27   |
> | GTPO     | 0.7480±0.0069  | 0.7770±0.0057  | 187.48±0.88  | 186.24±0.87  |
> | DAPO     | 0.7492±0.0098  | 0.7848±0.0126  | 112.57±0.46  | 112.43±0.35  |
> | GFPO     | 0.6187±0.0055  | 0.6399±0.0079  | 132.01±0.46  | 131.39±0.32  |
> | S-GRPO | 0.6633±0.0150  | 0.7046±0.0057  | 110.28±0.36  | 109.97±0.30  |
>
> Additionally, we examine our method on the **multimodal benchmark ChartQA**, please see the results as follows:
>
> | Method | Pass@16 | Pass@32 | Length@16 | Length@32 |
> |--------|---------|---------|-----------|-----------|
> | **IAPO** (ours) | 0.6800±0.0216 | 0.7233±0.0094 | 42.95±0.44 | 42.56±0.55 |
> | GFPO | 0.7033±0.0125 | 0.7333±0.0047 | 55.05±0.09 | 55.08±0.29 |
> | DAPO | 0.6867±0.0047 | 0.7267±0.0125 | 68.72±0.96 | 68.82±0.64 |
> | S-GRPO | 0.6867±0.0287 | 0.7233±0.0205 | 52.94±0.12 | 52.86±0.07 |
> | GTPO | 0.6867±0.0094 | 0.7167±0.0047 | 115.85±1.48 | 115.58±0.88 |
>
> We observe that we achieve the best token-efficiency in multimodal reasoning as well.

---

> > ### Author Rebuttal · Reviewer_G4dt · 2026-04-02
> >
> > Thank you for the clear and thoughtful response. Your explanations have fully resolved my concerns. I will raise my score to weak accept.

---

> > > ### Author Response · Authors · 2026-04-02
> > >
> > > Dear Reviewer G4dt,
> > >
> > > We appreciate your acknowledgement that our response is clear and thoughtful, addressing all your concerns. Thank you for raising your score.
> > >
> > > We are grateful for your time and efforts in reviewing our submission.
> > >
> > > Sincerely,
> > > Authors from Submission 21159

---

### Official Review · Reviewer_VUKm · 2026-03-12

**Soundness:** 4
**Presentation:** 3
**Significance:** 3
**Originality:** 4
**Overall Recommendation:** 4
**Confidence:** 4

**Summary:**

The paper proposes IAPO, a post training method for token efficient reasoning: the core idea is not simply to penalize long inference, but to allocate training signals based on the "information contribution" of each token to the final answer, allowing the model to retain truly useful inference steps and compress redundant content, thereby reducing inference length as much as possible without compromising accuracy. The author also designed an efficient estimation method for conditional mutual information and explored adjustment mechanisms to avoid the model being too short and answering incorrectly.

**Compliance With Llm Reviewing Policy:**

Affirmed.

**Final Justification:**

This paper is meaningful. Using entropy and outcome as signals to supervise the sequence process is a very interesting idea and quite inspiring. I maintain my positive attitude.

**Key Questions For Authors:**

The key quantity of IAPO is the approximate estimation of conditional mutual information between tokens and the final answer, but this estimation relies on early exit design, short postfix prompting, and several engineering approximations. Can the author provide a more systematic explanation of how much deviation this estimate may have from the true 'information contribution', and whether this deviation will significantly vary with task type, answer format, or model size?

**Limitations:**

No. Although the paper includes an impact statement, overall it mainly emphasizes the positive effects of reducing inference tokens, reducing computational costs, and improving accessibility. However, the limitations and potential negative social impacts of the method itself have not been fully discussed. For example, the experiments in this article mainly focus on mathematical reasoning tasks, so the method's generalization ability to other types of complex reasoning tasks is still unclear; Meanwhile, IAPO relies on additional token level information estimation, which incurs additional training costs, and its actual efficiency benefits need to be more comprehensively balanced.

**Strengths And Weaknesses:**

**Strengths**

1 The creative use of information gain as a measure of token importance provides an important dense signal for RL, which is a great idea

2 Compared to regular GRPO, IAPO tends to make the model generate shorter completion times; And this' shortening 'direction can be characterized by a very clear statistic, not just by empirical observation. More specifically, Theorem 4.1 of the paper proves that under small step update conditions, IAPO has a more advantageous expected output length change compared to GRPO.

3 The paper is written very clearly and the structure is very clear

**Weakness**

1 The experiment is not strong enough and should be tested on more complex benchmarks, such as AIME 2025.

2 Author can establish a more in-depth analysis and link the relationship between reward and information gain

---

> ### Author Rebuttal · Authors · 2026-03-31
>
> We sincerely appreciate the reviewer's detailed, constructive, and insightful feedback. We are grateful for your acknowledging our idea as "**creative**" and "**great**", our method as theoretically principled and empirically performant, our writing and structure as "**very clear**." Here, we address the reviewer's concerns point-by-point as follows:
>
> > Concern 1: Experiments should be tested on more complex benchmarks, such as AIME.
>
> Thank you for the insightful suggestion. Evaluating on the AIME dataset would demonstrate whether IAPO's information-aware compression generalizes to olympiad-level problems where long reasoning chains may be genuinely necessary. We ran IAPO and the adopted baselines on AIME 2026 (we noticed that AIME released its newest version, AIME 2026, so we conducted the experiment directly on this new version of the data). Due to the computing resource and time limit, we are unable to conduct experiments on Qwen2.5-7B-Instruct, and the Qwen2.5-0.5B-Instruct model could not successfully answer AIME questions due to their high complexity. Therefore, we only report the results for Qwen2.5-1.5B-Instruct, please see the following table:
>
> | Method | Pass@2 | Pass@4 | Pass@8 | Pass@16 | Pass@32 | Length@2 | Length@4 | Length@8 | Length@16 | Length@32 |
> | :----: | :---: | :---: | :---: | :---: | :---: | :---: | :---: | :---: | :---: | :---: |
> | **IAPO** (ours) | 0.0556±0.0786 | 0.1111±0.0786 | 0.1111±0.0786 | 0.1667±0.0000 | 0.1667±0.0000 | 30.97±4.97 | 32.97±5.29 | 32.18±2.44 | 31.36±1.33 | 31.63±0.96 |
> | S-GRPO | 0.0000±0.0000 | 0.0000±0.0000 | 0.0000±0.0000 | 0.0000±0.0000 | 0.0000±0.0000 | 83.89±9.75 | 78.64±6.21 | 81.04±4.34 | 79.99±3.49 | 80.61±3.17 |
> | GTPO | 0.0000±0.0000 | 0.0000±0.0000 | 0.0000±0.0000 | 0.0556±0.0786 | 0.0556±0.0786 | 72.25±24.19 | 65.60±11.01 | 59.06±8.11 | 67.05±3.65 | 60.48±4.58 |
> | DAPO | 0.0000±0.0000 | 0.0000±0.0000 | 0.0000±0.0000 | 0.1111±0.0786 | 0.1111±0.0786 | 67.33±10.33 | 67.51±2.10 | 64.65±2.97 | 65.76±3.48 | 63.43±1.15 |
> | GFPO | 0.0000±0.0000 | 0.0000±0.0000 | 0.0556±0.0786 | 0.0556±0.0786 | 0.0556±0.0786 | 47.86±1.68 | 50.17±0.49 | 50.44±1.64 | 53.97±1.75 | 52.04±0.56 |
>
> We observe from the table that our IAPO indeed achieves the best effectiveness-efficiency trade-off across baselines on AIME2026, with the highest Pass@k and the smallest Length@k.
>
> > Concern 2: More analysis linking reward and information gain.
>
> We thank the reviewer for this question. The binary reward $r_i = \mathbf{1}[o_i \text{ is correct}]$ and the mutual information $I(y_i; o_i \mid q)$ both measure how well the completion $o_i$ accomplishes the task. However, the binary reward has a fundamental limitation for token-level training: it is rigid and non-decomposable. It assigns a single scalar to the entire completion, providing no principled way to determine which tokens were
> responsible for correctness. Mutual information resolves this via the chain rule:
> $I(y_i; o_i \mid q) = \sum_t I(y_i; o_{i,t} \mid q, o_{i,<t}) = \sum_t s_{i,t}.$
> Unlike the binary reward, $I(y_i; o_i \mid q)$ can be exactly and principally decomposed into per-token contributions. We therefore can view information gain as a **soft, token-distributable surrogate for reward**: it preserves the same notion of task completion as the binary reward, while enabling fine-grained credit assignment across tokens that the binary reward cannot provide. IAPO exploits this property to redistribute gradient mass toward tokens that genuinely contribute to the correct sequence-level reward signal.
>
> > Concern 3: How much deviation does the early-exit MI estimator have from the true information contribution, and does this vary with task type, answer format, or model size?
>
> This is a key question about the reliability of our approximation. The approximation quality depends on: (1) how well the model can produce a calibrated answer distribution from a partial reasoning chain, and (2) the length of the postfix prompt relative to the full completion. For (1), we note that the estimator is only used during *training* to shape advantages, not at inference time. Even a noisy but directionally correct signal is sufficient to provide a useful training signal, as is the case with noisy reward signals in standard RL. For (2), since K $\ll$ L in practice, and our KV-cache preloading approach guarantees the estimator sees the full preceding context, it provides a reasonable conditional distribution. To fully address the reviewer's concern on MI estimation, we conduct an experiment (results displayed in Fig. 1  of [the anonymous link]( https://anonymous.4open.science/r/agent_rl-107E/)) quantifying the correlation between our estimated MI values and a simulated ground-truth conditional MI computed via Monte Carlo sampling (using multiple rollouts from each prefix position), across different task types and model sizes. From the Figure, we can observe that the estimated MI is strongly correlated with the real conditional MI.

---

> > ### Author Rebuttal · Reviewer_VUKm · 2026-04-01
> >
> > Thanks for your response. I will keep my positive score. Good luck.

---

> > > ### Author Response · Authors · 2026-04-02
> > >
> > > Dear Reviewer VUKm,
> > >
> > > We sincerely thank you for the acknowledgement that your concerns are adequately addressed. We thank you again for your time and effort in reviewing our submission.
> > >
> > > Sincerely,
> > > Authors from Submission 21159

---

### Official Review · Reviewer_uFAL · 2026-03-13

**Soundness:** 3
**Presentation:** 3
**Significance:** 3
**Originality:** 3
**Overall Recommendation:** 4
**Confidence:** 2

**Summary:**

The paper consists of token-efficient post-training of LLM via RL. The authors claim the tendency of RL to produce excessively verbose, redundant, and computationally expensive reasoning chains. To mitigate this, the authors propose IAPO, which shapes token-level advantages based on an approximation of Conditional MI. They introduce an early-exit strategy to estimate this MI by appending a postfix prompt to partial completions and measuring the drop in answer entropy. To circumvent the massive computational cost of this token-level evaluation, they employ KV-cache preloading and chunk-wise forwarding.

**Compliance With Llm Reviewing Policy:**

Affirmed.

**Final Justification:**

The authors have Basically addressed my concerns, I will raise my score.

**Key Questions For Authors:**

Can author provide a table detailing the exacttraining time and peak GPU memory consumption for standard GRPO, DAPO, and your IAPO?

**Limitations:**

The authors completely fail to discuss the  limitations regarding the actual computational overhead introduced during the training phase.  See question and Weakness

**Strengths And Weaknesses:**

The problem of reasoning verbosity is a highly rpractical issue in the deployment of reasoning models.
The implementation of KV-cache preloading and chunk-wise forwarding is a solid piece of systems-level engineering.

Weakness:
Ratio@k appears to be a somewhat questionable metric. More importantly, the Pareto frontier between Pass@k and Length@k would be a more informative evaluation, rather than relying on such a single metric.

Moreover, the proposed method does not seem to demonstrate significantly superior performance on some baselines. In addition, since the authors employ KV cache and chunking techniques to accelerate training, the actual training cost remains unclear.

---

> ### Author Rebuttal · Authors · 2026-03-31
>
> We sincerely appreciate the efforts and time you have dedicated to reviewing and providing invaluable feedback to enhance the quality of this paper. We also thank you for acknowledging our research problem as **highly practical** and our methodology as a **solid piece of systems-level engineering**. Here is a point-by-point reply to your concerns and questions.
>
> > Concern 1: Pareto frontier between Pass@k and Length@k would be more informative than Ratio@k (Pass@k / Length@k).
>
> We appreciate this thoughtful suggestion and have added the Pareto frontiers figures in `iapo_pareto_frontier.html` in [the anonymous link]( https://anonymous.4open.science/r/agent_rl-107E/). We design Ratio@k (Pass@k / Length@k) as a direct single-number summary of the accuracy-efficiency trade-off, which follows from our problem definition (see Problem 1). We did not report the Pareto frontier because Table 1 provides the same information as the Pareto frontier (i.e., Pass@k and Length@k) with higher space efficiency. Specifically, we could place all token-efficiency-related performance information on one page in Table 1, while Pareto frontiers consume more space. We agree with the reviewer that the Pareto frontier is informative and will incorporate it into the revision.
>
>
> > Concern 2: IAPO does not show significantly superior performance on some baselines.
>
> We note that IAPO's advantage is most pronounced on the token-efficiency (specifically R@k), where it achieves the best or runner-up result across most of the experimental settings (8 out of 9 dataset-model combinations). Although IAPO is not always the shortest (some baselines like GFPO can produce extremely short but often incorrect outputs) or not always the most accurate, this is **by design**: IAPO aims to optimize its *ratio*, which is the objective stated in Problem 1. The fact that IAPO produces correct answers with the fewest tokens, rather than blindly short or maximally accurate, is the central contribution. Besides, we also tested our IAPO on the scientific reasoning dataset (ARC-Challenge) and the multimodal reasoning dataset (ChartQA), and it performs superior in terms of token efficiency; please refer to our rebuttal to *concern 4 of reviewer G4dt*.
>
> > Concern 3: Actual training cost (GPU hours, peak memory) is unclear.
>
> We thank the reviewer for the insightful comment. Here, we perform 30 steps of reinforcement learning and compare wall-clock training time and peak GPU memory usage on Qwen2.5-1.5B-Instruct with DAPO, GFPO, GTPO, S-GRPO, and IAPO under identical hardware conditions (8 NVIDIA H100 80GB) as follows  (GPU hours are the wall-clock hours multiplied by 8):
>
>
> | Model | Method | Wall-clock Hours (mean ± std) | Peak GPU Memory (GB) |
> |-------|--------|-----------------------------|----------------------|
> | Qwen2.5-1.5B-Instruct | DAPO | 0.000801 ± 0.000289 | 41.39 |
> |  | GFPO | 0.000796 ± 0.000156 | 41.39 |
> |  | GTPO | 0.000962 ± 0.000167 | 41.39 |
> |  | IAPO | 0.001846 ± 0.000550 | 42.42 |
> |  | SGRPOLee | 0.000736 ± 0.000131 | 41.39 |
>
> We note that, with our proposed KV-cache preloading and chunk-wise forwarding techniques, the computation overhead of our IAPO relative to baselines is moderate. This is because our two techniques can effectively bring the computation from naive MI estimation as high as $O(L^3d)$ down to only $O((K^3 + L^2)d)$, where $K$ (postfix length, ~5–10 tokens) $\ll$ L (completion length, ~100–500 tokens). Moreover, we also display **inference time cost** in Fig.8 in our paper, where IAPO performs visibly faster than non-post-trained models.

---

> > ### Author Rebuttal · Reviewer_uFAL · 2026-04-02
> >
> > Thank you for the clear  response. Your explanations have fully resolved my concerns.

---

> > > ### Author Response · Authors · 2026-04-02
> > >
> > > Dear Reviewer uFAL,
> > >
> > > We are grateful for your acknowledgment that our responses are clear and that your concerns have been fully resolved. We would be sincerely thankful if you could consider *raising your evaluation score* in light of this.
> > >
> > > Thank you so much for your time and consideration!
> > >
> > > Sincerely,
> > > Authors from Submission 21159

---

### Decision · Program_Chairs · 2026-04-30

**Decision:**

Accept (regular)

**Comment:**

This paper proposes IAPO, an information-theoretic post-training method that shapes token-level advantages using an estimate of each token's conditional mutual information with the final answer. The goal is to compress reasoning chains by rewarding informative tokens and suppressing low-utility exploration, with an early-exit MI estimator accelerated by KV-cache preloading and chunk-wise forwarding.

All four reviewers converge to Weak Accept after rebuttal.

Consensus strengths: the problem of verbose CoT is timely and practically important. Using conditional MI as a dense credit signal is a novel and principled idea, distinct from simple length penalties. The paper provides theoretical motivation (monotonic verbosity reduction under small updates), a clean implementation, and thorough initial evaluation showing up to 36% length reduction while maintaining accuracy on math reasoning.

Main weaknesses raised and addressed:

Evaluation scope and metrics. Reviewers asked for Pareto frontiers rather than Ratio@k alone, and for results beyond standard math sets. Authors added Pareto plots, and new results on AIME 2026, ARC-Challenge, and ChartQA showing IAPO achieves best or competitive accuracy-efficiency trade-offs, including strong performance on the multimodal ChartQA task.
Training cost. Initial submission lacked GPU-time analysis. Authors provided a detailed table showing IAPO adds modest overhead versus DAPO/GFPO/GTPO (e.g., ∼1.85 vs ∼0.8 wall-clock hours for 30 steps on 8×H100, peak memory 42.4 vs 41.4 GB), thanks to their KV-cache and chunking optimizations.

MI estimation validity. Concerns about using first-token entropy as proxy, normalization, and approximation error were addressed with ablations showing strong correlation between first-token and full-answer entropy, and between estimated MI and Monte-Carlo ground truth across tasks and model sizes.

Core assumption. The claim that information density decreases with length was supported with empirical plots on GSM8K showing strong negative correlation, addressing concerns about over-compression on hard problems.

Hyperparameters and generalization. Authors showed robustness across a wide range of α and β, and provided initial results on Qwen3-0.6B showing maintained gains, partially addressing concerns about sensitivity and recency of models.